# Metabolic rewiring in skin epidermis drives tolerance to oncogenic mutations

Anupama Hemalatha ®[1], Zongyu Li[2], David G. Gonzalez[1], Catherine Matte-Martone ®[1], Karen Tai[1], Elizabeth Lathrop[1], Daniel Gil[3,4], Smirthy Ganesan[1], Lauren E. Gonzalez ®[1], Melissa Skala ®[3,4], Rachel J. Perry ®[2] ✉ & Valentina Greco ®[1,5,6] ✉

Skin epithelial stem cells correct aberrancies induced by oncogenic mutations. Oncogenes invoke different strategies of epithelial tolerance; while wild-type cells outcompete β-catenin-gain-of-function (βcatGOF) cells, Hras[G12V] cells outcompete wild-type cells. Here we ask how metabolic states change as wild-type stem cells interface with mutant cells and drive different cell-competition outcomes. By tracking the endogenous redox ratio (NAD(P)H/FAD) with single-cell resolution in the same mouse over time, we discover that βcatGOF and Hras[G12V] mutations, when interfaced with wild-type epidermal stem cells, lead to a rapid drop in redox ratios, indicating more oxidized cellular redox. However, the resultant redox differential persists through time in βcatGOF, whereas it is flattened rapidly in the Hras[G12V] model. Using [13]C liquid chromatography–tandem mass spectrometry, we find that the βcatGOF and Hras[G12V] mutant epidermis increase the fractional contribution of glucose through the oxidative tricarboxylic acid cycle. Treatment with metformin, a modifier of cytosolic redox, inhibits downstream mutant phenotypes and reverses cell-competition outcomes of both mutant models.

Phenotypically normal human and mouse tissues contain oncogenic mutations, sometimes in frequencies similar to cancer[1–4]. It is critical to elucidate the cellular mechanisms that maintain tissue function and homoeostasis in the presence of oncogenic mutant cells to understand how those mechanisms are lost in disease and could be restored therapeutically. Using our ability to track oncogenic mutant cells within tissues over time, we have previously shown that genetically mosaic mouse skin cells can use different strategies to correct phenotypic aberrancies caused by oncogenic mutations and to restore function[5–7]. Specifically, while activated β-catenin (βcatGOF) mutant cells are eliminated from the tissue through selective differentiation[5], constitutively active Hras (Hras[G12V]) mutant cells outcompete wild-type

(WT) cells and are integrated into normal tissue architecture and function[5,7]. These two mutants and their behaviours in the adult mouse skin invokes cell-competition models, with activated βcatGOF cells taking on a 'loser' fate and Hras[G12V] taking on a 'winner' fate in comparison with WT cells. Understanding the early responses to either mutation would help uncover the sequence of events that culminate in a cell type being integrated in or being eliminated from the stem cell compartment.

Cell competition, the phenomenon by which unfit cells are eliminated from the tissue by their neighbours, is essential during development, maintenance of organ size and tumour surveillance[8,9]. Several signalling pathways, including the Wnt and Ras pathways (related to

[1]Department of Genetics, Yale School of Medicine, New Haven, CT, USA. [2]Departments of Cellular & Molecular Physiology and Internal Medicine (Endocrinology), Yale School of Medicine, New Haven, CT, USA. [3]Department of Biomedical Engineering, University of Wisconsin, Madison, WI, USA. [4]Morgridge Institute for Research, Madison, WI, USA. [5]Departments of Cell Biology and Dermatology, Yale Stem Cell Center, Yale Cancer Center, Yale School of Medicine, New Haven, CT, USA. [6]Howard Hughes Medical Institute (HHMI), Chevy Chase, MD, USA. ✉e-mail: rachel.perry@yale.edu; valentina.greco@yale.edu

mutations used in this study) have been implicated[9]. Cell competition is thought to act through modulation of mechanical forces[10], apicobasal polarity[10,11] and metabolism[12,13]. Among these, cellular metabolism is sensitive to other sensors of cell competition, such as growth factor signalling[14] and mechanical forces[15], and could potentially integrate upstream cues to directly modulate behaviours involved in tissue growth, such as proliferation[14,16] and increase in cell size[17,18].

Proliferative cells, including stem cells, disproportionately upregulate glycolysis over mitochondrial oxidation, even in the presence of oxygen, despite glycolysis being far less efficient in generating ATP; this is historically described as the Warburg effect, initially observed and most commonly studied in tumours[19,20]. Hence, understanding the dynamics of metabolic state in stem cell behaviour of the highly regenerative skin epidermis in the presence of oncogenic mutations is especially relevant.

However, it remains difficult to measure metabolic state dynamically throughout time and with intracellular resolution while preserving the morphology and integrity of the observed tissue. Optical redox imaging captures the endogenous fluorescence of the reduced NAD(P)H and oxidized FAD such that their relative ratios reflect the balance between reduced and oxidized reactions within the cell[21–26]. Their relative ratio also represents the balance of glycolytic rate (measured by the reduced NADH levels) and mitochondrial metabolism (measured by oxidized FAD levels) within the cell[27,28]. Optical redox imaging has also been shown to detect metabolic responses to deviations from homoeostasis in squamous epithelia, including wound healing[29–31].

To interrogate the cellular properties that determine cell-competition outcomes, we combine live imaging of the mutant mosaic epidermis with optical redox imaging, so as to determine intracellular metabolic states as the tissue adapts to oncogenic mutations. Revisiting the same epidermal cells in the live animal over long periods of time has previously enabled us to uncover intercellular coordination and dynamic behaviours within the epidermal stem cell compartment[5,32,33]. Tracking the metabolic state of epidermal stem cells allows us to bridge the gap between cell behaviours and intracellular pathways that define the ability of cells to persist within the tissue. To understand why redox ratios change, we measured alterations in glucose metabolism tailoring an established liquid chromatography–tandem mass spectrometry (LC–MS/MS)-based stable isotope glucose tracer technique[34] in awake mice.

Collectively, we find that the drop in redox ratio, observed as one of the earliest changes in both βcatGOF and Hras[G12V] mutant epidermis, is explained by enhanced fractional utilization of glucose in mitochondrial oxidation in epidermal stem cells, contrary to the Warburg effect. This metabolic rewiring is necessary to drive downstream cell and tissue morphological changes that uniquely support each mutation to have opposite cell-competition phenotypes. Thus, we uncover dynamic metabolic adaptations in vivo that drive oncogenic phenotypes and cell-competition outcomes, and show the different metabolic trajectories used to re-establish tissue homoeostasis.

## Results

### Optical redox imaging in epidermal stem cells through time

The mouse epidermis is constantly renewed by stem cells located in the basal layer (Fig. 1a(i),(ii)), all of which are capable of proliferating and replenishing the differentiated layers[35]. To date, the interplay between the metabolic state and cellular homoeostasis in vivo in epithelial cells is not understood. To address this question, we adapted our previously described two-photon microscopy deep-tissue imaging[35–37] to incorporate optical redox imaging[22,23] and visualize label-free fluorescence of the endogenous metabolites in the hairy (ear) and nonhairy (paw) skin of live mice (Fig. 1a(i)–(iii)). The intensities captured for redox imaging at specific two-photon excitation wavelengths from intravital imaging are collected at wavelength ranges that correspond to NAD(P) H and FAD and can be represented post-imaging as a ratio NAD(P) H/ FAD (Fig. 1a and Supplementary Videos 1 and 2). To overcome the challenge posed by the undulating three-dimensional (3D) tissue in our measurements of NAD(P)H and FAD fluorescence in hundreds of stem cells in the live animal, we isolated the fluorescence signal from the basal layer using distance from the interface of the epidermis and collagen (captured via the second harmonic signal) (Extended Data Fig. 1e,f).

Given the unique regenerative ability of epidermal stem cells, we asked whether they had a distinct metabolic signature compared with other cell types in the dermis. We found that the cells in the epidermis have a high NAD(P)H intensity in comparison with FAD (Fig. 1a(iv)). This is unlike many cells in the dermis (Fig. 1a(iv) and Supplementary Videos 1 and 2). To test whether our method accurately captured NAD(P)H and FAD intensities, we first treated 293T cells in culture with cyanide, which blocks complex IV of the electron transport chain and causes the accumulation of NADH in the cells. We observed an immediate (2–5 min) increase in NAD(P)H intensities and NAD(P)H/ FAD ratio (Extended Data Fig. 1a). We then proceeded to test the redox ratios we measure in vivo by injecting sodium cyanide intradermally into live mice and immediately (5–10 min post injection) recording the accumulation of NAD(P)H and increased NAD(P)H/FAD from the same mice (Fig. 1b). This extreme and irreversible treatment represents the upper range of redox change that is physiologically possible (28–37% increase), underlining the importance of maintaining redox ratios in a narrow range. Thus, in vivo, upon inhibition of mitochondrial oxidation, NADH accumulates leading to increased NAD(P)H/FAD, indicative of a more 'reduced' redox state.

Epidermal stem cells are constantly cycling through cell cycle, leading us to ask whether their metabolic signature is dynamic or stable. We revisited the same tissue region over hours to days to track its metabolic state (described in Methods and Extended Data Fig. 1b). In the basal stem cell layer, there were large clusters (~25–50) of cells sharing similar NAD(P)H intensities in a pattern that remained stable over 2–3 h of imaging (Extended Data Fig. 1c). However, these clusters had different relative NAD(P)H intensities when revisited after 24 h (Fig. 1c(i) and Extended Data Fig. 1d). To understand how the cell cycle may influence NAD(P)H intensities, we used a reporter

**Fig. 1 | Epidermal stem cells have a unique and constrained metabolic signature at homoeostasis. a**, (i) Label-free imaging of NAD(P)H and FAD in the skin of live mice reveals (ii) the dynamic metabolic signature of (iii) the stem cell compartment. Only the reduced and oxidized counterparts NADH, NADPH and FAD are fluorescent and together they report on the redox ratio of the cell. (iv) Two-photon images of various epidermal layers and cell types in skin epithelia capture the endogenous autofluorescence of the metabolic cofactors NAD(P)H and FAD. NAD(P)H/FAD levels in the epidermal stem cell compartment (basal layer) are high compared to cells in dermis. Redox ratio (NAD(P)H/FAD) is represented as an intensity plot (scale below). (iii) Schematic showing how changes in glycolysis or mitochondrial metabolism rates influence the redox ratio. **b**, NAD(P)H/FAD ratios from the basal layer before and 5–10 min after sodium cyanide injection locally. Rapid increase in NAD(P)H/FAD is plotted on the right. The coloured dots represent average redox ratios per mouse from three mice, revisited before and after injection. **P = 0.0082 (paired t-test two-sided). Violin plot, n = 222 (before) and n = 187 (after) cells from three mice. **c**, (i) Basal layer when revisited after 24 h shows altered distribution of NAD(P)H intensity patterns. The underlying second harmonic signal patterns used to identify the same regions are shown in Extended Data Fig. 1b. (ii) Fucci (cell cycle reporter)-labelled stem cells. Green-labelled nuclei represent Geminin-containing nuclei (S/G2 cell cycle stage) and red-labelled nuclei represent Cdt1-containing nuclei (G1 stage). (iii) NAD(P)H intensities per cell measured from G1 nuclei show an increase in mean intensity compared with S/G2 nuclei. Violin plot, n = 120 (S/G2) and n = 490 (G1) cells from three mice. The average value from each mouse is labelled with a different colour. *P = 0.0063 (two-sided nested t-test). Scale bar, 50 μm. Panel **a**(i) created with BioRender.com.

that distinguishes G1 cells (marked by mCherry-Ctd1) and S/G2 cells (marked by mVenus-Geminin-Fucci2 mice) (Fig. 1c(ii)). We observed a small but significant difference between the mean NAD(P)H intensities of S/G2 cells and G1 cells in the basal layer, with S/G2 cells displaying a relatively lower NAD(P)H intensity than G1 cells (Fig. 1c(iii)).

Altogether, we show that epidermal stem cells have a unique metabolic signature, with small variations conferred by changing cell cycle statuses in individual cells contributing to the spread of the redox values in homoeostasis.

## Redox changes precede βcatGOF-induced aberrancies

We next asked how epidermal stem cell redox states change upon expression of the oncogenic mutation βcatGOF (*β-catenin*[flox(Ex3)] (refs. 5,6)). Mice carrying *βcatGOF; K14CreER* were treated with

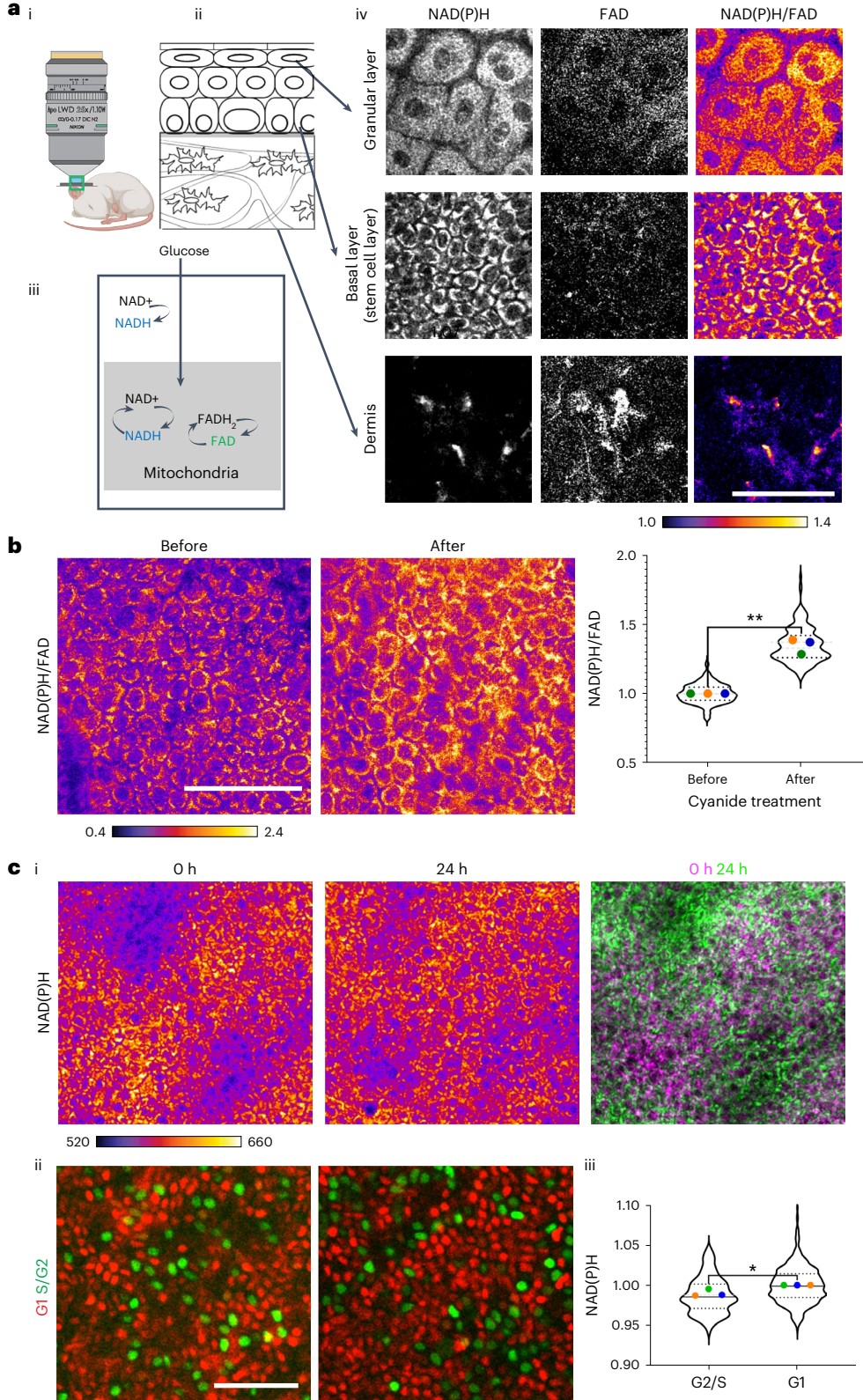

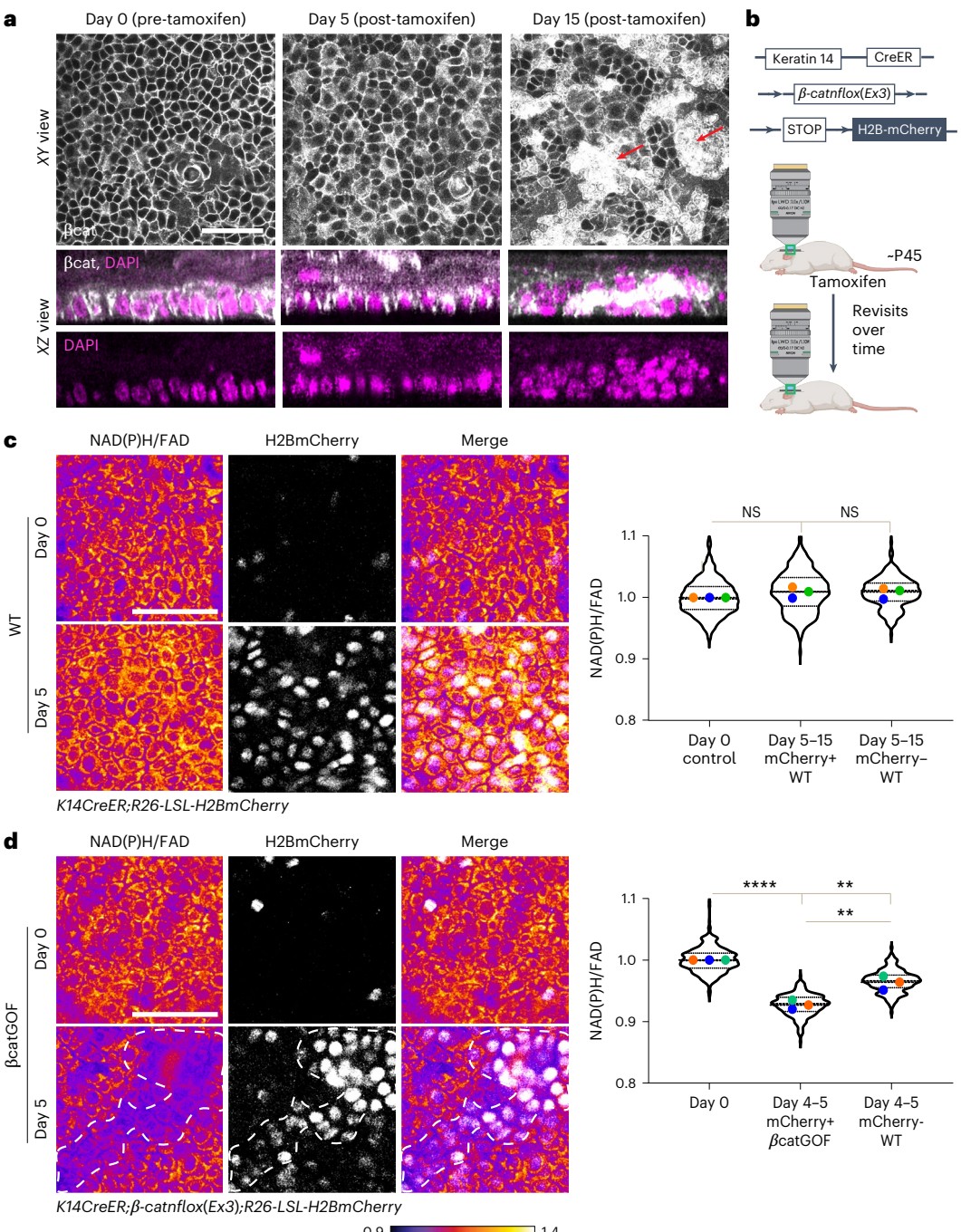

**Fig. 2 | βcatGOF-induced stem cells shows rapid changes in redox (NAD(P) H/FAD) ratio before the emergence of other morphological aberrancies.**
**a**, β-catenin immunofluorescence staining in basal layer of epidermis from *K14CreER; βcatGOF* mice 0, 5 and 15 days post-tamoxifen-induced recombination and expression of βcatGOF. The *xy* (top) and *xz* (bottom) sections show that the 3D structure of the basal layer is unperturbed at 5 days, unlike the multiple rows of nuclei found in aberrant placodes at 15 days (red arrows; z-stack for day 0, 5 and 15 in Supplementary Videos 3–5; images in Extended Data Fig. 2b). **b**, Schematic showing that similar regions from the same animal were imaged before and after tamoxifen-induced recombination and expression of βcatGOF and H2BmCherry (**c**,**d**). **c**, In control littermates (*K14CreER; LSL-H2BmCherry*), recombination post-tamoxifen leads to the expression of H2BmCherry (white nuclei). These recombined cells or their neighbours show no significant reduction in NAD(P)H/FAD ratio when imaged at 5, 9 or 15 days after tamoxifen

administration. The average value from each mouse is labelled with a different colour for all graphs. Averages not significantly different (*n* = 3 each). Violin plot *n* (in order *x* axis) = 423, 250 and 221 cells from three mice. **d**, In βcatGOF mice (*K14CreER; βcatGOF; LSL-H2BmCherry*), mutant cells are indicated by coexpression of nuclear H2BmCherry (white, outlined by white dashed line). Despite there being no other morphological aberrancy, the NAD(P)H/FAD ratio of mutant cells (H2BmCherry-positive) steeply drops after 5 days when compared to values at day 0. The neighbouring WT cells (H2BmCherry-negative) also show a drop in redox ratio (NAD(P)H/FAD), although to a higher range than the mutant cells. Average redox *P* value from one-way analysis of variance (ANOVA); multiple comparisons; day 0 versus day 4–5 βcatGOF, ****$P$ < 0.0001; day 0 versus day 4–5 WT, **$P$ = 0.0027; day 4–5 βcatGOF versus day 4–5 WT, **$P$ = 0.0032. *n* = 3 mice each. Violin plot *n* (in order *x* axis) = 336, 231 and 198 cells from three mice. Scale bar, 50 μm. NS, not significant. Panel **b** created with BioRender.com.

tamoxifen to activate the Cre recombinase in the basal stem cell layer (which expresses keratin 14) and, in turn, induce the expression of βcatGOF in a mosaic manner. At day 5 post-tamoxifen, the basal layer consisted of a single layer of nuclei, as in homoeostasis, with increased nuclear β-catenin localization compared to control cells (Fig. 2a). By day 15, aberrant outgrowths with several layers of closely packed nuclei enriched in nuclear β-catenin had formed and protruded into the dermis, resembling hair follicle placodes (Fig. 2a, Extended Data Fig. 2a and Supplementary Video 5). To probe cell behavioural changes, we stained for proliferation markers at the above time points and found that there was no statistically significant changes in the number of phospho-histone H3 (pH3)-labelled nuclei at day 5 post-tamoxifen, but there was a significant increase in pH3-labelled nuclei at day 15 (Extended Data Fig. 2b,c). Hence, at day 5 post-tamoxifen, even though β-catenin protein showed greater nuclear (activated) localization than in controls, we did not observe aberrancies in tissue structure or quantifiably significant changes in proliferation in the stem cell compartment of βcatGOF mosaic tissue (Fig. 2a, Extended Data Fig. 2 and Supplementary Videos 3–5).

To interrogate whether metabolic changes precede or follow these morphological aberrancies in the mutant mosaic epidermis, we measured NAD(P)H and FAD levels in the stem cell layer beginning at day 5 post-tamoxifen, before the appearance of aberrant phenotypes. To distinguish between and follow recombined vs. unrecombined epidermal cells (for example βcatGOF versus WT in the mutant mosaic epidermis) over time in the adult mouse, we expressed *Rosa26-CAG-LSL-H2B-mCherry* (referred to as *LSL-H2BmCherry*) in the basal stem cell compartment using *K14CreER* (*K14CreER; LSL-H2Bm-Cherry*) (Fig. 2b). First, we measured the variance of redox values in control conditions including after tamoxifen-induced recombination, expression of fluorescent alleles and repeated imaging of the same epidermal region. We found that per-cell NAD(P)H/FAD is maintained in a consistent average value (with a slight increase in the range post-tamoxifen) in the homoeostatic basal stem cells in revisits 5, 9 or 15 days after tamoxifen treatment and the expression of fluorescent reporters (Fig. 2c).

To understand the metabolic changes in basal cells in the context of tissue and behavioural changes that happen upon induction of the βcatGOF mutation (Fig. 2a and Extended Data Fig. 2a), we used Cre-induced H2BmCherry and βcatGOF coexpression to identify and track mutant βcatGOF cells over time (*K14CreER; β-catn*^flox(Ex3)^*; LSL-H2BmCherry*). Approximately 80% of the stem cell layer was recombined and expressed H2BmCherry and βcatGOF. Five days after induction, the βcatGOF mutant stem cells (H2BmCherry-positive) reported a rapid drop of NAD(P)H/FAD ratio (Fig. 2d). This change is in the opposite direction to what was observed in recombined WT animals and of a much larger magnitude (7–10-fold higher). WT cells in the mosaic mutant tissue (H2BmCherry-negative) also reported a decreased NAD(P)H/FAD ratio, but to a range distinct from and higher than that of the mutant cells (Fig. 2d), suggesting that they respond to

alterations in the metabolic state of neighbouring cells. Hence, the stem cell layer undergoes a rapid change in its redox state in response to the presence of βcatGOF cells before the development of morphological and behavioural aberrancies, making it one of the first observable responses to the presence of βcatGOF mutation.

Over time, the βcatGOF mutation causes changes to stem cell proliferation and overall tissue architecture (Fig. 2a and Extended Data Fig. 2a). Thus, we asked if the early drop in redox ratio we observed in these cells is a permanent alteration in metabolic state. By tracking the NAD(P)H and FAD intensities in the same region of basal stem cell layer over time (Extended Data Fig. 2d), we observed that the cells expressing βcatGOF (H2BmCherry-positive) maintained a low NAD(P)H/FAD ratio range at 10 days post-tamoxifen, with negligible recovery from 5 days post-tamoxifen (Fig. 3a,b and Extended Data Fig. 2e). In contrast, the neighbouring WT cells in the basal stem cell layer increased their NAD(P)H/FAD ratio around 2.8-fold when compared with βcatGOF cells, to resemble homoeostatic redox values (day 0) (Extended Data Fig. 2e). Because of this, the redox differential between the βcatGOF cells and WT cells in the stem cell layer increased over time (Fig. 3c) through the selective recovery of NAD(P)H/FAD ratios in WT cells.

Our previous work showed that βcatGOF cells are eliminated from outgrowths over the course of months, aided by the presence and activity of WT cells[5]. To better understand the recovered metabolic signature of WT cells in the context of their competitive advantage, we followed the fate of the WT and βcatGOF cells over time. We observed that 5–10 days post-tamoxifen within the same region of skin, the area occupied by WT cells indicated by the absence of H2BmCherry expanded (Fig. 3e,f), whereas the H2BmCherry-positive mutant cells can be progressively seen enriched in the suprabasal layer, with WT cells underneath (Fig. 3d), indicating that these mutant cells are eliminated from the basal stem cell layer through differentiation. The mutant epidermis also had a progressively more pronounced cell-competition phenotype when followed over a longer time span. At 1.5–2 months post-tamoxifen, there was a prominent difference in the occupancy of recombined cells in both the WT control littermates (*LSL-H2BmCherry;K14CreER*) and βcatGOF mosaic (*βcatGOF; LSL-H2BmCherry;K14CreER*) models. At this time point, the recombined cells (H2BmCherry-positive) in WT control littermates still occupied around 80% of the basal stem cell layer, in agreement with neutral drift, but in the βcatGOF mosaic epidermis, only about 25% of the basal stem cell area was occupied by H2BmCherry/βcatGOF-positive cells, reflecting a rapidly shrinking coverage by the mutant cells over time (Extended Data Figs. 3a and 9a). The remaining recombined H2BmCherry/βcatGOF-positive cells were mostly packed into 3D aberrant placodes extending into the dermis (Extended Data Fig. 3b and schematic shown in Fig. 6i) and resembling hair follicles; this also occurred in nonhairy βcatGOF mosaic skin (Extended Data Fig. 3c). Altogether, WT cells recover their redox status after the initial redox drop, and ultimately outcompete the βcatGOF cells to occupy more area within the basal stem cell layer over time.

**Fig. 3 | βcatGOF and WT neighbour cells have different trajectories of redox recovery over time. a**, NAD(P)H/FAD intensities and H2BmCherry expression (white; indicates βcatGOF cells) from same region (outlined in yellow from larger regions shown in as Extended Data Fig. 3a) in *K14CreER; βcatGOF; LSL-H2BmCherry* mice revisited 5 days and 10 days after tamoxifen administration. **b**, Insets outlined in white from panel A, highlighting regions across time wherein (i) βcatGOF cells at day 5 and day 10 show little change of redox ratio and (ii) βcatGOF cells (H2BmCherry-positive) are replaced with WT cells (H2BmCherry-negative), which have recovering redox ratios. **c**, Redox differential between mutant βcatGOF cells (H2BmCherry-positive) normalized to neighbouring WT cells (H2BmCherry-negative) at day 4–6 and day 9–11 post-tamoxifen shows that redox differential between βcatGOF cell and neighbouring WT cells increases between 5 and 10 days because of the selective recovery of WT cells. Extended Data Fig. 2e shows same data plotted with respect to day 0. *P* value

(two-sided *t*-test) from *n* = 3 mice each. *\**P* = 0.012; \*\*\**P* = 0.0002 Violin plot *n* (in order *x* axis) = 323, 198, 273 and 227 cells. **d**, At day 10 post-tamoxifen, the βcatGOF cells (H2BmCherry-positive) are pushed to the suprabasal region, indicating differentiation (top), whereas WT cells occupy the basal layer (middle). Both layers can be see seen in the *xz* section (bottom) with the blue outline indicating the epidermal–dermal interface. **e**, The βcatGOF mosaic basal stem cell layer 5 days post-tamoxifen harbours adjacent patches of WT (H2BmCherry-negative; yellow outline) and mutant cells (H2BmCherry-positive), which were revisited 10 days post-tamoxifen; WT patches (regions outlined in yellow) expanded between day 5 and day 10. **f**, Graph shows area occupied by WT cells from the same 300 × 300 μm$^2$ regions (*n* = 6) from three mice quantified and shows an increase in coverage between day 5 and day 10. *P* = 0.0015 (paired *t*-test, two-sided). Scale bar, 50 μm.

## Hras^G12V initially cause then flatten the redox differential

While βcatGOF mutant cells are outcompeted from the mouse epidermis by WT cells, other cell-competition models include mutant cells that outcompete WT neighbours. This prompted us to ask whether the metabolic changes leading to altered redox and recovery fates that we observed in the βcatGOF mosaic are specific to the mutation itself or if they are indicative of more general cellular fitness during re-establishment of homoeostasis. To address this,

we used the Hras^G12V (constitutively active Hras) mutation model where the relationship between WT and mutant cells is opposite to that in the βcatGOF model: mutant Hras^G12V cells outcompete WT cells in the basal stem cell layer[5,7] (Extended Data Fig. 4a). Using similar genetic backgrounds and experimental time scales as in the βcatGOF experiments described above, we asked how the NAD(P)H/FAD intensities in *K14CreER;Hras^G12V;LSL-H2BmCherry* epidermal stem cells changed upon tamoxifen-induced recombination and expression

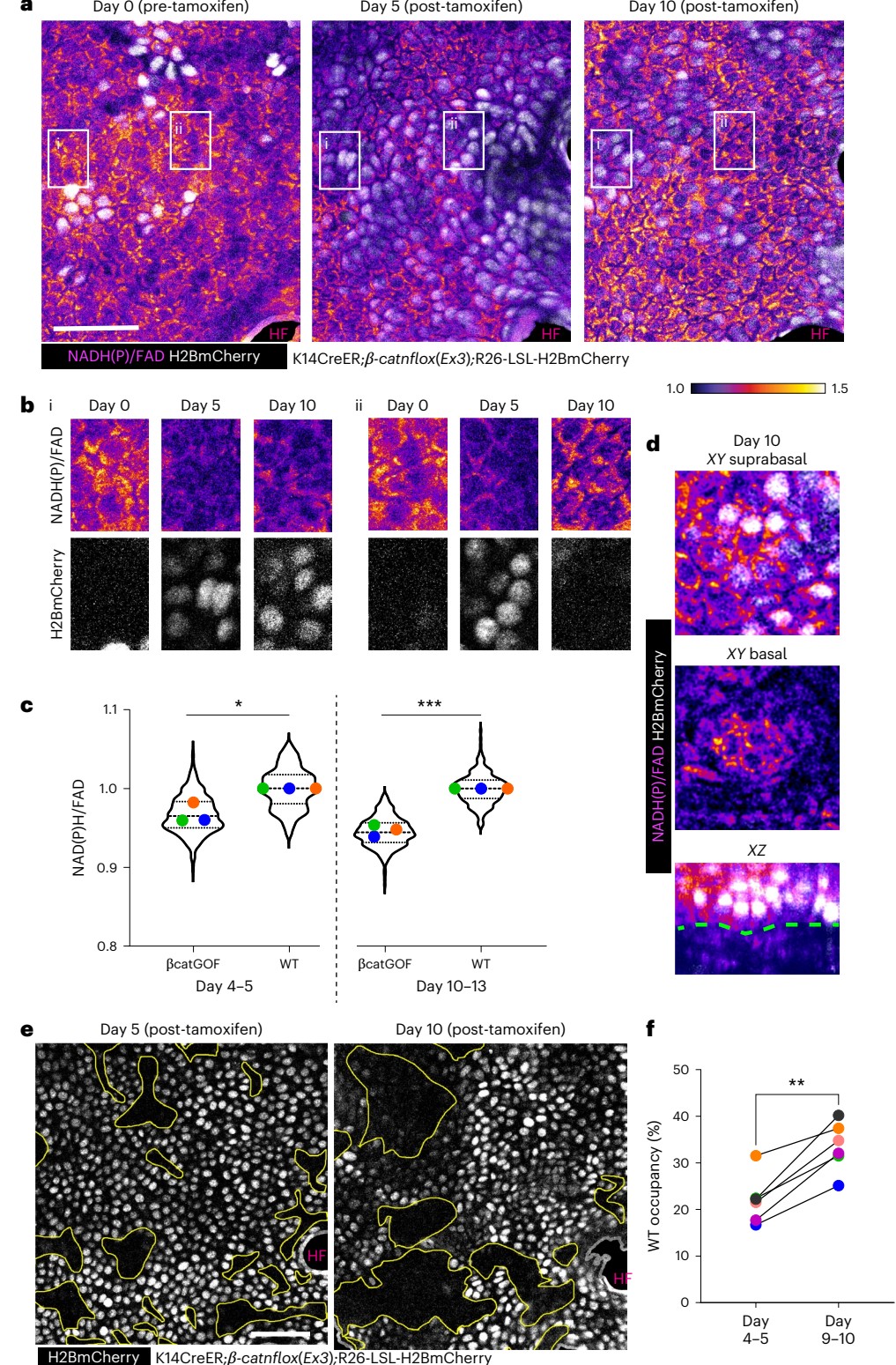

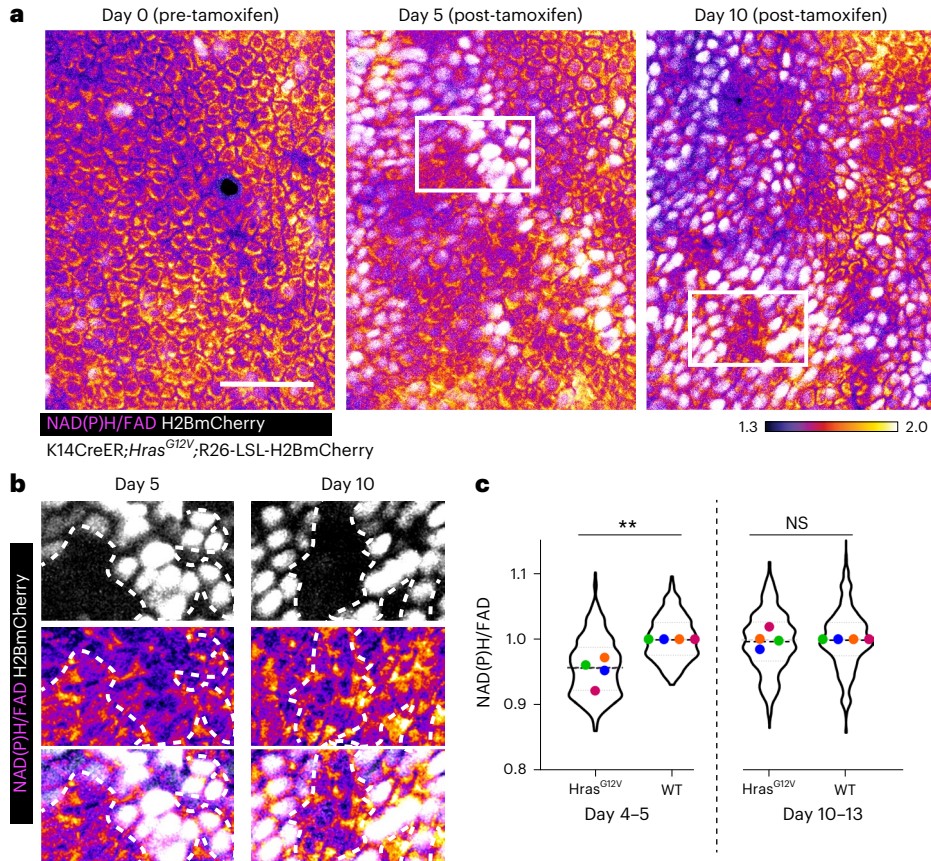

**Fig. 4 | Cells with Hras^G12V mutation first show a drop in NAD(P)H/FAD which recovers over time. a**, NAD(P)H/FAD intensities from same epidermal regions (outlined in yellow from larger regions shown in Extended Data Fig. 4a) revisited at 0, 5–6 and 10–13 days post-tamoxifen and expression of Hras^G12V and H2BmCherry (white) from *K14CreER; Hras^G12V; LSL-H2BmCherry* mice. **b**, Insets from white outlined regions in **a**. Regions consisting of Hras^G12V and WT cells side by side show that at day 5 (left) Hras^G12V (H2BmCherry-positive; outlined by white dotted line) cells have a lower NAD(P)H/FAD intensity than WT cells (H2BmCherry-negative). At day 10 (right), Hras^G12V (H2BmCherry-positive) cells have increased and recovered their NAD(P)H/FAD intensities to be similar to neighbouring WT

cells (H2BmCherry-negative). **c**, Quantification of NAD(P)H/FAD intensities from Hras^G12V (H2BmCherry-positive) cells normalized to their WT neighbours (H2BmCherry-negative) at day 4–5 (left) and day 10–13 (right) to show that redox differential day 4–5 is flattened by day 10–13. This is in contrast with βcatGOF epidermis in which the redox differential between mutant and WT increases (Fig. 3c). The average value from each mouse is labelled with a different colour. Average redox *P* value (two-sided *t*-test) from four mice, **P = 0.0044 for day 4–5 Hras^G12V versus WT; not significantly (NS) different for day 10–13 Hras^G12V versus WT. Violin plot *n* (in order *x* axis) = 425, 228, 560 and 226 cells from four mice. Extended Data Fig. 4b shows the same data plotted with respect to day 0. Scale bar, 50 μm.

of Hras^G12V. We observed an initial drop in NAD(P)H/FAD ratios at 5 days post-tamoxifen (Fig. 4a–c and Extended Data Fig. 4b) in the basal stem cells when the same region of skin was followed over time (Extended Data Fig. 4a). By following the changing redox over time points in which the recombined H2BmCherry/Hras^G12V-positive cells expand (5–10 days after induction), we asked how the metabolic status of these cells changed in the context of their competitive advantage. Between 5–10 days post-tamoxifen, the NAD(P)H/FAD ratio of the H2BmCherry/Hras^G12V-positive cells equalized with that of their neighbouring WT cells (Fig. 4c). At 10 days post-tamoxifen, mutant cells and WT neighbours in the Hras^G12V model changed their redox ratios (Fig. 4c and Extended Data Fig. 4b), flattening the redox differential between mutant and WT cells. This is unlike βcatGOF mutant cells, which widened their redox differential with neighbouring WT cells (Fig. 3c) and were ultimately outcompeted by WT cells. Thus, by comparing the Hras^G12V and βcatGOF mutation with different cell-competition outcomes, we discover that both mutations initially lower the redox ratio, but in the Hras^G12V mutant model, the redox differential between mutant and WT cells is flattened over time.

### Changes in glucose catabolism underlie redox changes
NAD(P)H and FAD ratios represent the relative balance of reduced to oxidized metabolites in a cell, which in turn, depends on the relative

rates of glycolytic to mitochondrial metabolism. To determine the mechanistic and biochemical basis of the changes in the NAD(P)H to FAD intensities observed through live imaging in the mutant models described so far, we asked how the glucose catabolic rates were changed in those models. We observed that the direction of redox change in βcatGOF and Hras^G12V models were consistent with drug treatment (dichloroacetic acid; DCA) that shunted more glucose into the tricarboxylic acid (TCA) cycle and downstream oxidation (Extended Data Fig. 4c,d). We first utilized untargeted metabolomics and pathway enrichment analysis assess global metabolic changes, out of which glycolysis and TCA cycle emerged among the most altered metabolic pathways in βcatGOF and Hras^G12V when compared with WT littermates (Extended Data Fig. 5 and Supplementary Table 1). In the metabolomics dataset, we also find a reduction in levels of pentose phosphate pathway components and accumulation of a few amino acids. To further understand how the specific steps of glucose metabolism that are likely to change the redox state of the mutant cells are altered, we performed targeted LC–MS/MS-based stable isotope tracer analysis following an infusion of $^{13}$C isotope-labelled glucose into P45–P62 mice 6 days after injecting high doses of tamoxifen to induce expression of βcatGOF or Hras^G12V in most if not all epidermal cells. Epidermal tissue isolated from the mice infused with $^{13}C_6$ glucose (Methods)

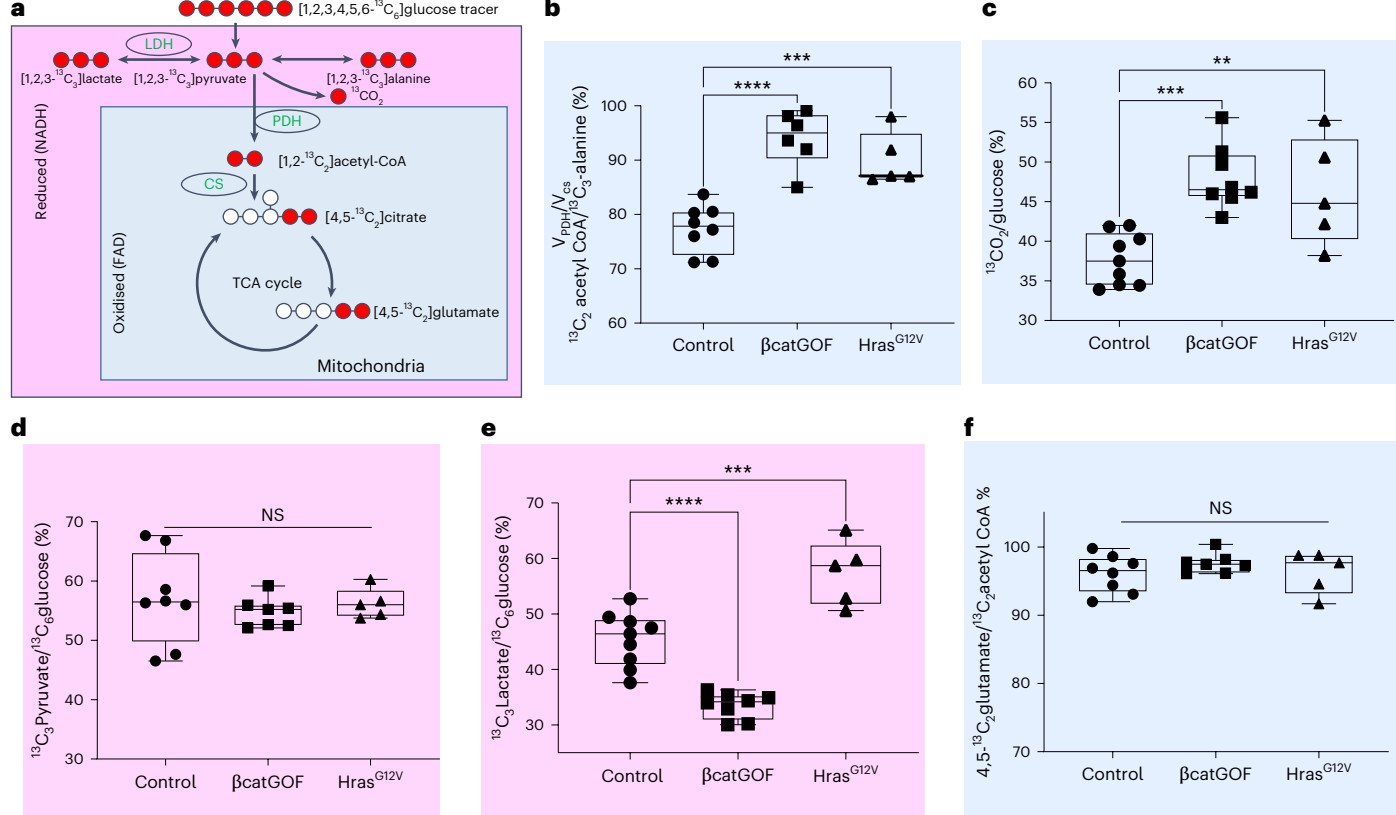

**Fig. 5 | Changes in glucose catabolic fluxes underlie altered NAD(P)H and FAD changes in mutant conditions. a**, Schematic showing the labelling of carbon (red) in downstream metabolites when mice are infused with $^{13}C_6$ glucose. The readouts which pertain to reduced NADH in the cytosol are labelled in pink and those pertaining to mitochondrial metabolism are labelled in blue. **b**, The ratio of $^{13}C_2$-acetyl-coA/$^{13}C_3$-alanine, also called $V_{PDH}/V_{CS}$, represents the percentage of TCA cycle fuelled by glucose. When compared with WT control, this ratio increase to almost a 100% showing that the βcatGOF and Hras$^{G12V}$ are utilizing glucose maximally for TCA cycle and downstream glucose oxidation. $n = 8$ (control), 6 (βcatGOF) and 5 (Hras$^{G12V}$) mice, $P$ values, ****$P < 0.0001$; ***$P = 0.0005$. **c**, Another measure of TCA flux, $^{13}CO_2$ normalized to $^{13}C_6$ glucose, also increases in both mutants. $n = 9$ (control), 8 (βcatGOF) and 5 (Hras$^{G12V}$) mice $P$ value, ***$P = 0.0003$; **$P = 0.0058$. **d**, In contrast, labelled pyruvate normalized to labelled glucose is unchanged in both mutants. $n = 8$ (control), 7 (βcatGOF) and

5 (Hras$^{G12V}$) mice. **e**, Labelled lactate relative to glucose, is decreased in βcatGOF, whereas in the Hras$^{G12V}$ epidermis, the $^{13}C_3$ labelled lactate to $^{13}C_6$ glucose ratio is increased. Hence the mutant Hras$^{G12V}$ epidermis makes more lactate from pyruvate when compared to the βcatGOF tissue, in which this conversion is lower than WT controls. $n = 9$ (control), 8 (βcatGOF) and 5 (Hras$^{G12V}$) mice, $P$ values, ****$P < 0.0001$;***$P = 0.0002$. **f**, $^{13}C_2$-labelled glutamate to acetyl-CoA is a measure of the dilution of the label due to glutamine entry into the TCA cycle. $n = 8$ (control), 7 (βcatGOF) and 5 (Hras$^{G12V}$) mice. The nearly 100% ratios suggest that there is not much entry of glutamine. This is not significantly different in either mutant epidermis when compared with WT epidermis. All graphs use a one-way ANOVA (multiple comparisons with respect to control). Each point represents epidermis (ear) from one mouse. All box plots show all points with whiskers going from minimum to maximum with a line at the median.

was then used to assess the label retention at various steps of glycolytic and oxidative metabolism to determine relative fluxes through various steps (Fig. 5a)[34]. We found that $V_{PDH}/V_{CS}$ (the ratio of pyruvate dehydrogenase flux to citrate synthase flux), the fraction of the TCA cycle fuelled by glucose[34,38,39], indicated by 4,5-$^{13}C_2$glutamate/$^{13}C_3$-alanine (Extended Data Fig. 6b) and the fractional contribution of glucose to acetyl-CoA, $^{13}C_2$-acetyl-CoA/$^{13}C_3$-alanine (Fig. 5b) were upregulated in both βcatGOF and Hras$^{G12V}$ epidermis (acetyl-CoA integrates contributions from glucose and fatty acid metabolism). Thus, the TCA cycle was fuelled almost entirely by glucose in both mutant tissues. As the TCA cycle and mitochondrial oxidative phosphorylation are tightly coupled, this means that mutants upregulate glucose utilization for mitochondrial oxidation. $^{13}C$-labelled $CO_2$ relative to glucose was also upregulated in both mutant tissues (Fig. 5c), directly indicating that the flux of glucose through the TCA cycle and downstream oxidation was enhanced by the mutations. Of note, this enhanced utilization of glucose for mitochondrial oxidation is in agreement with the drop in NAD(P)H to FAD ratio (a more oxidized redox state) that we observed through live imaging (Figs. 2c,d and 4a–c), providing a direct parallel biochemical validation.

To understand whether glycolysis changed, we focused on pyruvate and lactate labelling derived from glucose, which is expected to be higher in oncogenic mutations, consistent with the Warburg effect[19]. At the final step or readout of glycolysis, specifically the transfer of $^{13}C$ label from glucose to pyruvate, there was no parallel change in the labelling of pyruvate derived from glucose in either mutant epidermis (Fig. 5d). However, when we examined the exchange between pyruvate and lactate, the two mutations differed in their effects: cells in the Hras$^{G12V}$ epidermis made more lactate from glucose, whereas cells in the βcatGOF epidermis made less (Fig. 5e). Taken together, these data point toward a specific and differential modulation of pyruvate to lactate conversion in the two mutants when compared with control epidermis (Extended Data Fig. 6c). As NADH is generated at earlier steps of glycolysis before pyruvate, these specific changes in pyruvate to lactate step is consistent with the more oxidized redox ratio observed for both mutants. The $^{13}C$-labelled glutamate derived from $^{13}C$-acetyl-CoA (m + 2 $C_4C_5$-glutamate/m + 2-acetyl-CoA) was not different in the two mutant models (unlike the ratio of labelled glutamate to pyruvate), indicating that the glutamine contribution to the TCA cycle and mitochondrial oxidation (dilution of glutamate by unlabelled glutamine entry) was minimal

and did not differ between the WT and two mutant models (Fig. 5f and Extended Data Fig. 6e). There is no dilution of the $^{13}$C at the later step of m + 2 malate when compared with glutamate (2,3-$^{13}$C$_2$ malate/4,5-$^{13}$C$_2$-glutamate; Extended Data Fig. 6f) either. In these studies, 2,3-$^{13}$C$_2$ malate was considered an integrated average of 1,2-$^{13}$C$_2$ and 3,4-$^{13}$C$_2$ malate (both of which are unmeasurable by MS), weighting the fraction of anaplerosis generating C$_3$C$_4$ malate as compared with C$_1$C$_2$ malate. Thus, the major changes in glucose oxidation are at the level of glucose entry into the TCA cycle, and there is no evidence of altered anaplerosis from other substrates (Fig. 5f and Extended Data Fig. 6d–f).

## Metformin reverses mutant phenotypes and cell competition

The drop in NAD(P)H/FAD levels, increase in V$_{PDH}$/V$_{CS}$, and increase in $^{13}$CO$_2$/$^{13}$C$_6$-glucose all demonstrate a greater fractional contribution of glucose to mitochondrial oxidation in both βcatGOF and Hras$^{G12V}$ epidermis. To test whether these changes are simply associated with the oncogenic mutant phenotype or casual to it, we treated mice with metformin, a mild inhibitor of mitochondrial activity[40–43], also characterized before in oncogenic models[42,44] and known to change cytosolic redox to more reduced[45,46]. We acknowledge that multiple mechanisms of action of metformin have been demonstrated and that its systemic effects are likely not limited to mild inhibition of mitochondrial activity; however, as the preponderance of evidence suggests a role for it to reduce mitochondrial oxidation[47] within a physiologically relevant range[48], we selected it for further phenotype and flux assessment.

Mutant βcatGOF mice were given daily administration of metformin[48] in drinking water for continuous exposure to the drug. We first showed that homoeostatic structure and function of skin epidermis, including morphology of the different layers of epidermis, proliferation rate and differentiation patterns (Extended Data Fig. 7a–d) of the epidermal stem cell compartment, were not affected in WT littermates. We then tested the effects of metformin onto glucose metabolism in epidermis of WT and mutant βcatGOF and Hras$^{G12V}$ mice after $^{13}$C$_6$ glucose infusion. In line with an inhibition of mitochondrial oxidation, $^{13}$C$_2$-acetyl-CoA/$^{13}$C$_3$-alanine and $^{13}$CO$_2$/$^{13}$C$_6$ glucose is lowered in the epidermis of both WT and mutant mice (Extended Data Fig. 8a,b). Notably, the decrease of glucose contribution to the downstream TCA and oxidation is higher in both βcatGOF and Hras$^{G12V}$ mutants, abrogating the differences between the WT and mutant epidermis; indicative of dependence on optimal mitochondrial activity for rewiring metabolism by the mutant cells. In agreement with the reduction in pyruvate (derived from $^{13}$C$_6$-glucose) shunting to the TCA cycle and glucose oxidation, we also observe an increase in $^{13}$C$_3$- lactate/

$^{13}$C$_6$-glucose and $^{13}$C$_3$- lactate/$^{13}$C$_3$-pyruvate upon metformin treatment significantly in WT and βcatGOF mutants (Extended Data Fig. 8c,d). The Hras$^{G12V}$ mutant that had an elevated $^{13}$C$_3$- lactate/$^{13}$C$_6$-glucose even before treatment, retained similar levels, indicating that Hras mutant cells are already at a maximal rate of pyruvate to lactate exchange. Metformin also causes a mild dilution of $^{13}$C-labelled glutamate derived from $^{13}$C-acetyl-CoA (within 85–90%), indicative of some glutamine anaplerosis (Extended Data Fig. 8e) as reported previously[49]. Thus, metformin reduces glucose oxidation and flattens the metabolic differential between WT and mutant βcatGOF and Hras$^{G12V}$ mutant epidermis.

To test the effects of metformin on redox ratio upon mutation induction, we quantified NAD(P)H/FAD at 5 days, and show that metformin-treated βcatGOF animals do not display a significant change in redox (Fig. 6a,b and Extended Data Fig. 7e,f) between WT and mutant cells at day 4–5 post-mutation induction, mirroring the flattening of glucose metabolic differential between WT and mutant βcatGOF after metformin (Extended Data Fig. 8). The variance of per-cell redox upon metformin treatment, however, is higher than in untreated mice, reflecting variations in drug penetrance. To assess the impact of this treatment at the tissue level, we tracked the treated and untreated mutant mice to 3 weeks after treatment, the time frame during which the βcatGOF mosaic mutant animals develop epidermal outgrowths that deform the collagen and extend into the dermis (Fig. 6c,d and Supplementary Video 6). Notably, in βcatGOF mutant animals treated with metformin, there was a dramatic reduction in the number and size of these mutant driven aberrant epidermal outgrowths when compared with untreated mutant mice (Fig. 6c,d,g and Supplementary Videos 6 and 7). To further probe the cell-competition phenotype at the cellular level, we tracked H2BmCherry-positive (indicating mutant cells) versus H2BmCherry-negative (WT cells) regions at 2–2.5 months post-induction of βcatGOF. In mutant mice treated with metformin, the H2BmCherry-positive cells are largely not eliminated and are remarkably still present in the majority of the epidermis (~60%) (Fig. 6f,h) in contrast with what is observed in mutant mice without treatment (~20%; Fig. 6e,h). The 'loser' phenotype of βcatGOF is visually striking in revisited mice at this time point wherein large areas of skin free of H2BmCherry-positive mutant cells are seen previously occupied by mutant cells (Fig. 6h and Extended Data Fig. 9a). In contrast, the epidermis of metformin-treated mutant βcatGOF (Fig. 6h and Extended Data Fig. 9b) mice is still occupied by H2BmCherry-positive mutant cells, similar to control animals (littermates where only H2BmCherry was recombined and revisited; ~80% H2BmCherry occupied; Extended Data Figs. 3a and 9c).

**Fig. 6 | Metformin treatment inhibits βcatGOF mutant-induced epidermal outgrowths and reverses cell-competition outcome. a,b,** In βcatGOF mice (K14CreER; βcatGOF; LSL-H2BmCherry), treated with metformin, 5 days post-induction of mutation in a mosaic manner, there is no change in redox ratio in both recombined H2BmCherry-positive (white nuclei) βcatGOF mutant cells and WT neighbours (H2B-mCherry-negative), when compared with day 0 (before recombination; images in Extended Data Fig. 7e), plotted in **b**. One-way ANOVA of averages and nested with multiple comparisons shows no significant changes. Violin plot: n (in order x axis) = 326, 228 and 201 cells from three mice. Extended Data Fig. 7f plots the same data with mutant and WT neighbours compared with each other. Scale bar, 10 μm. **c,** Maximum projection of z-stacks from epidermis and dermis with epidermal H2BmCherry-positive βcatGOF mutant cells (white) and collagen through second harmonic imaging (SHG; yellow) in vivo (magnified in red insets). In βcatGOF mutant animal epidermal outgrowths are visible at week 3 post-induction (day 19–21) protruding into the dermis (video through the z-section also in Supplementary Video 6). Upon metformin treatment, these epidermal protrusions are no longer visible in the maximum projections through dermis (see **i** for schematic). **d,** The depth of epidermal protrusion from the top of the cornified layer from fixed epidermal preps from βcatGOF mice compared with βcatGOF mice given metformin. The depth is colour-coded according to the lookup table (LUT) bar on the left and quantified

in the graph in **g**. While the βcatGOF epidermis has deeper (yellow/red) epidermal outgrowths at week 3, these outgrowths are either absent or greatly reduced in size when mice are given metformin. Scale bar, 100 μm (**c,d**). **e,** When the βcatGOF animals are revisited after 2.5 months, H2B-mCherry-positive βcatGOF cells (white nuclei) are largely absent from regions they previously occupied at week 3. **f,** In contrast, βcatGOF animals treated with metformin when revisited at 2 months, still retained most of the H2B-mCherry-positive βcatGOF cells. Both **e** and **f** are zoomed-in regions cropped from larger ~1.8 mm$^2$ regions in Extended Data Fig. 9a,b where these differences in mutant cell occupancy can be observed too. **g,** Thickness from the top of the epidermis to the bottom plotted (images in D) showing reduction in the depth of outgrowths βcatGOF mutant epidermis when treated with metformin. n, average thickness from nine regions per mouse of size 400–500 μm$^2$ from five mice for mutant and metformin-treated. P value ****P < 0.0001 (two-sided t-test). **h,** Percentage of the area occupied by mutant (H2BmCherry-positive cells) quantified (images in **e,f**; Extended Data Fig. 9a,b) n = 3–4 regions per mice (size 700–1,200 μm$^2$) from three mice each. P value ****P < 0.0001 (two-sided t-test) Data are presented as mean ± s.d. (**g,h**). Scale bar, 50 μm (**e,f**). **i** Schematic showing temporal sequence of events in βcatGOF mutant epidermis where mutant cells are contained in epidermal outgrowths at weeks 2–3 (leading to expansion of WT cells in the basal layer; Fig. 3e,f), followed by extensive elimination of mutant cells by 2–2.5 months.

Thus, although the metformin-treated mice do not form morphological aberrancies or epidermal outgrowths at 3 weeks, mutant cells are also not efficiently eliminated from the skin and hence their cell-competition outcome is reversed. Interestingly the epidermis of βcatGOF treated with metformin is still hyperproliferative (Extended Data Fig. 10a,b) and the inhibition of morphological changes and cell-competition outcome cannot be explained merely by changes in proliferation. In addition, keratin 10 (K10; differentiation marker)

staining of metformin-treated βcatGOF mosaic epidermis shows a distribution of K10+ cells, similar to wild-type[50], but in contrast to the large ectopic clusters of K10+ cells adjoining pH3+ cells surrounding outgrowths in untreated βcatGOF mosaic epidermis (Extended Data Fig. 10c,d).

To also probe whether metabolic rewiring of cellular redox and glucose oxidation affects the phenotype of mutant Hras[G12V] mice, we asked what happens to the Hras[G12V] mutant epidermis, upon treatment

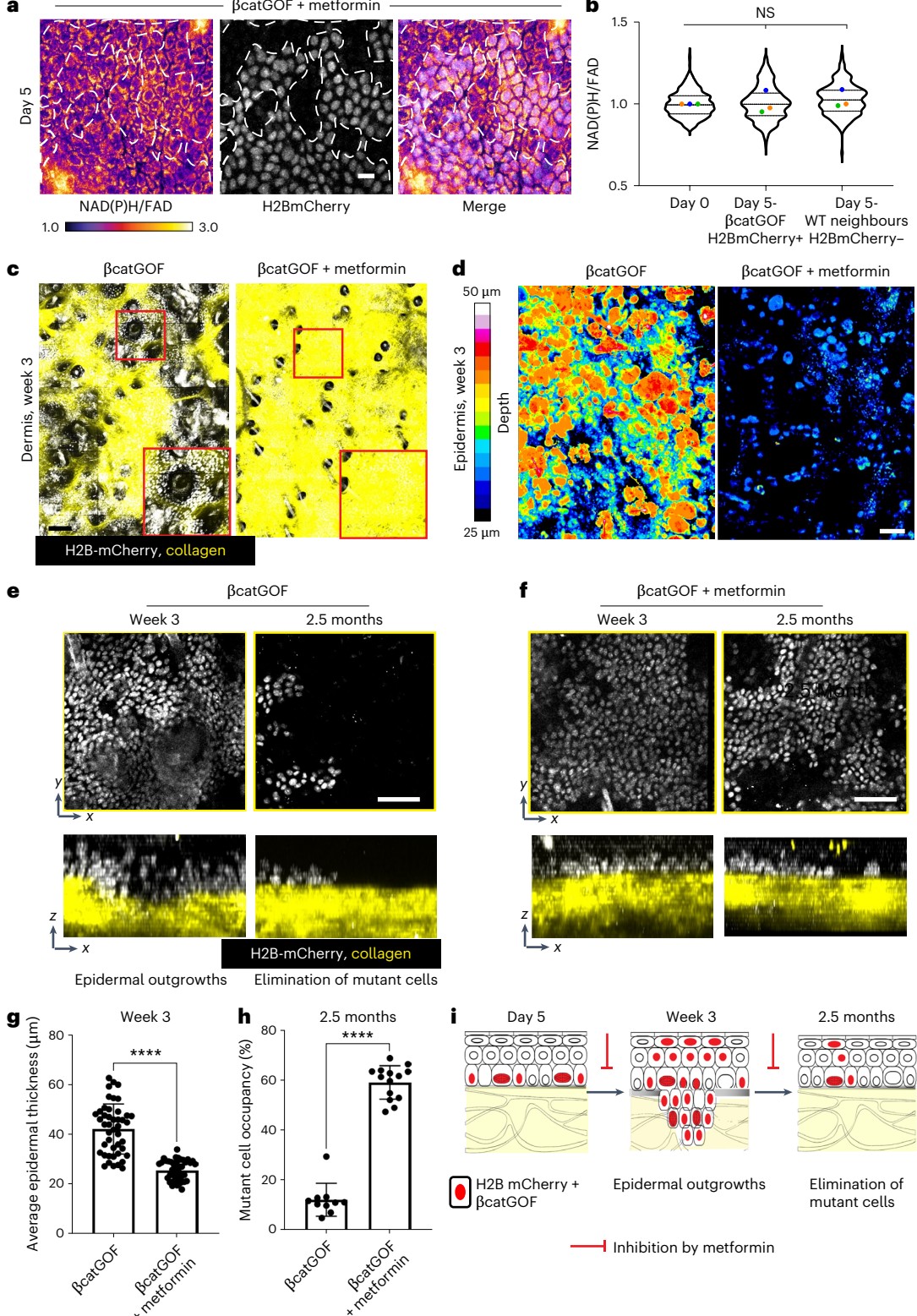

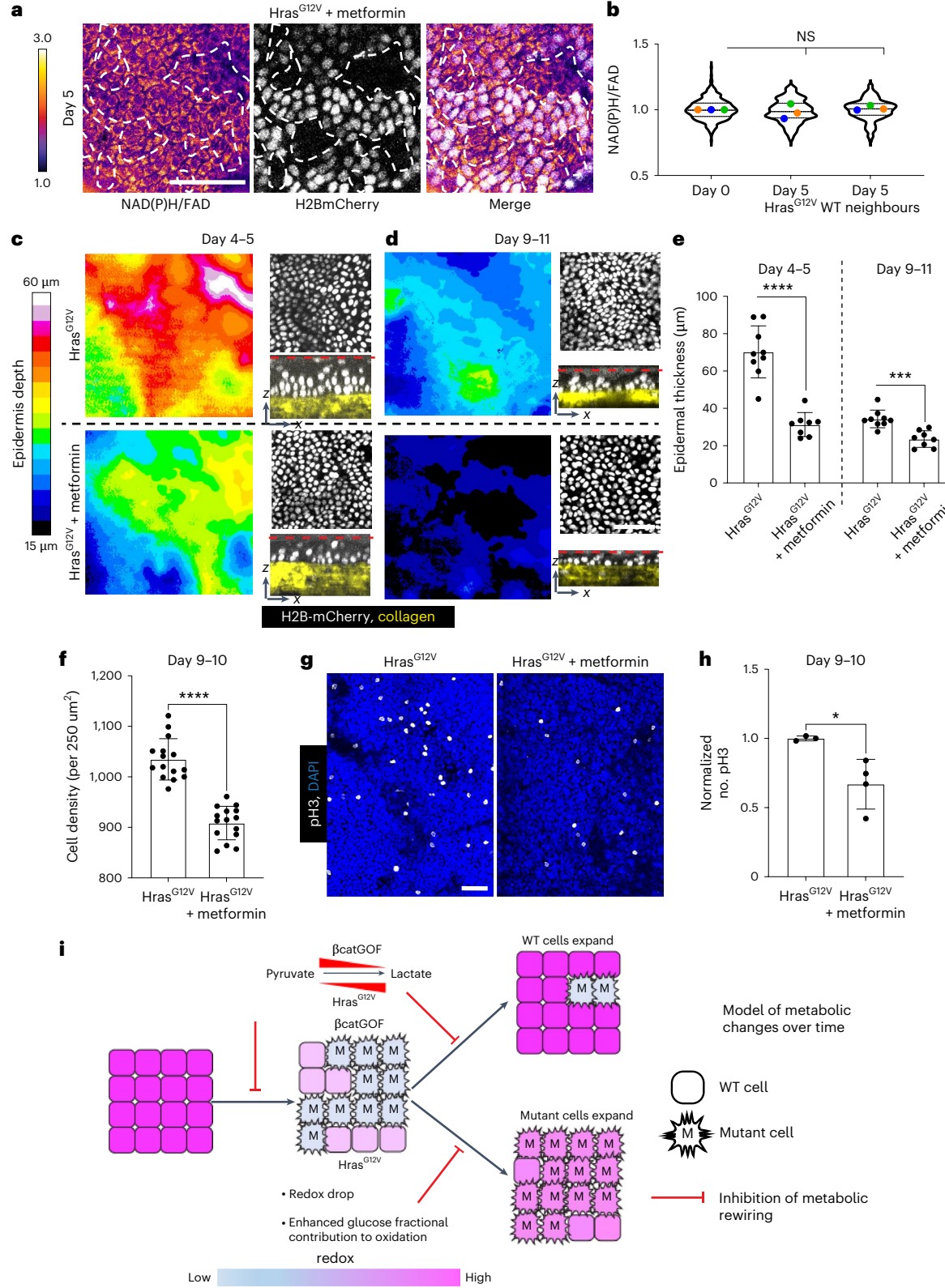

with metformin. The Hras[G12V] cells in metformin-treated animals also do not show a drop in NAD(P)H/FAD (Fig. 7a,b and Extended Data Fig. 7g,h), in contrast to our earlier observations in mutant animals (Fig. 4c). The characteristic phenotype of Hras[G12V] mutant epidermis (hyper-thickening at day 5, increased basal layer (stem cell layer) cell density at day 10 and increase in proliferation) are all attenuated when the animals are treated with metformin (Fig. 7c–h). Hence the

phenotypes downstream of Hras[G12V] were suppressed such that the mutant cells lose their proliferative advantage and mutant phenotype.

Taken together, our experiments showing the redox drop in mutant epidermis, changes in mitochondrial substrate utilization and metformin-mediated inhibition of metabolic rewiring, demonstrate that inhibiting the mutant-induced changes in cytosolic redox and mitochondrial glucose oxidation in the epidermal stem cell

**Fig. 7 | Metformin treatment inhibits Hras$^{G12V}$ mutant-induced tissue phenotypes and reverses cell-competition outcome. a,b,** In Hras$^{G12V}$ mice (*K14CreER;Hras$^{G12V}$;LSL-H2BmCherry*), treated with metformin, 5 days post-induction of mutation in a mosaic manner, there is no change in redox ratio in both recombined H2BmCherry-positive (white nuclei); Hras$^{G12V}$ mutant cells and WT neighbours (H2B-mCherry-negative), when compared to day 0 (before recombination; images in Extended Data Fig. 7g); plotted in **b**. A one-way ANOVA of averages and nested with multiple comparisons shows no significant changes. Violin plot, *n* (in order *x* axis) = 307,196 and 136 cells from three mice. Extended Data Fig. 7h plots same data with mutant and WT neighbours compared to each other. **c,d,** The increase in epidermal thickness (colour-coded in the LUT bar on left) in Hras$^{G12V}$ mutant epidermis is inhibited upon treatment with metformin both 4–5 days (thickest) and 9–11 days post-mutation induction. *xz* sections showing epidermal nuclei between cornified layer (red dotted line) and collagen imaged through second harmonic signal in yellow shows hyper-thickening in Hras$^{G12V}$ mutant mice compared to metformin-treated mutant mice, compare top (Hras$^{G12V}$) to bottom (Hras$^{G12V}$ + metformin). The hyper-thickening is mostly resolved by day 9–11 in Hras$^{G12V}$ mice. **e,** Graph quantifying epidermal thickness shown in images in **c,d**. The same regions were revisited between days 4–5 and day 9–11. *n* = 3 regions of interest each of 300 μm$^2$ from three mice for mutant and metformin-treated. *P* value (two-sided *t*-test) ****$P$ < 0.0001; ***$P$ = 0.0002. **f,** Cell density in the basal stem cell layer quantified at day 9–10 from epidermal preps shows metformin treatment inhibiting the enhanced density in Hras$^{G12V}$

mice; also seen in the *xy* panels shown in **d**; compare top (Hras$^{G12V}$) to bottom (Hras$^{G12V}$ + metformin), *n* = five 250-μm$^2$ regions from three mice each. *P* value (two-sided *t*-test) ****$P$ < 0.0001. **g,h,** pH3-positive dividing cells in Hras$^{G12V}$ mice and Hras$^{G12V}$ mice treated with metformin show a decrease in the proliferation rate quantified in **h**. *n* = 3, 4 mice each for Hras and Hras + metformin (normalized from 900–1,200 μm$^2$ regions). *P* value (two-sided *t*-test) *$P$ = 0.026. Data are presented as mean ± s.d. (**e,f,h**). Scale bar, 50 μm (**a–g**). **i,** Model showing metabolic rewiring drives tissue phenotypes that result in opposite cell-competition outcomes. Upon induction of βcatGOF and Hras$^{G12V}$ mutations there is a rapid reduction in NAD(P)H/FAD of epidermal stem cells in both mutant and WT cells at early time points, indicating a more-oxidized redox ratio. The mutant epidermis also enhances relative flux of glucose through the TCA cycle and mitochondrial oxidation consistent with more-oxidized cellular redox. Revisits over time reveal that in the βcatGOF mutant model, the redox differential between winner WT cells neighbouring the mutant cells is maintained, whereas in the Hras$^{G12V}$ mutant model, the redox differential is only transient and flattens over time. In parallel, the βcatGOF mutant cells are outcompeted by WT cells and the Hras$^{G12V}$ cells expand in the basal stem cell layer of the epidermis. Upon mild inhibition of mitochondrial oxidation by metformin, the redox drop in the mutant mosaic epidermis is inhibited and tissue phenotypes downstream are also inhibited, leading to inhibition of the cell-competition outcome, the elimination of βcatGOF cells and proliferative advantage of Hras$^{G12V}$ cells.

compartment inhibit downstream βcatGOF and Hras$^{G12V}$ cellular and tissue changes that support their opposite cell-competition phenotypes. These data demonstrate that inducing metabolic changes downstream of mutations in βcatGOF and Hras$^{G12V}$ is the mechanism by which these mutations exert their effect on the tissue.

## Discussion

With unprecedented resolution of the metabolic state in parallel with tracking cell behaviours over time in cell competition in vivo, we discover that the NAD(P)H/FAD ratio is cell-type specific and rapidly lowers in response to oncogenic mutations, before any observable morphological and behavioural changes. While βcatGOF cells maintain a low redox ratio and get eliminated over time, when compared with their WT neighbours, the mutant Hras$^{G12V}$ cells rapidly flatten their redox differences with WT neighbours. The reduction in the NAD(P)H to FAD ratio suggests increased net mitochondrial oxidation, and we show that this is attributable to an increased preference for glucose as a fuel for the TCA cycle and downstream mitochondrial oxidation. Inhibiting the changes in redox and mitochondrial oxidation using metformin inhibits specific downstream phenotypes of both mutant epidermis and reverses their cell-competition outcomes such that that the βcatGOF cells are no longer effectively eliminated and Hras$^{G12V}$ cells lose their proliferative advantage.

A high glycolytic rate and glucose oxidation may be a characteristic of rapidly cycling tissues with an active stem cell pool. Of note, in the presence of the oncogenic mutations βcatGOF and Hras$^{G12V}$, contribution from glucose to mitochondrial oxidation increases and almost reaches 100%. Although this is paradoxical to the conventional understanding of the Warburg effect, an increasing number of in vivo studies suggest that increased glycolysis does not always come at the expense of the TCA cycle and mitochondrial oxidation[51–54]. Notably, however, the brain also exhibits a high $V_{PDH}/V_{CS}$[34], similar to skin epidermis, raising the possibility that their developmental origin (ectoderm-derived) could also define these metabolic signatures. Moreover, pathways such as fatty acid oxidation, which were found to be upregulated in the Ras cancer cell lines[55], are not enriched in metabolomics analysis from Hras$^{G12V}$ epidermis, underscoring the constrained and regulated metabolic changes in vivo after induction of Hras$^{G12V}$, when this mutation is tolerated in the epidermis.

Cell–cell communication of the metabolic state across the epidermal basal stem cell layer is an unexpected finding, wherein mutant cells exerted influence on neighbouring WT cells to change their redox state. Exploring the composition and nature of this cell–cell communication

and understanding whether it is mediated indirectly through substrate competition or directly through metabolites that communicate the redox state of a cell to its neighbour would be interesting avenues for further studies.

While cell-competition mechanisms have historically implicated apoptosis[8,10,56], work from us and others have shown that homoeostatic mechanisms prevalent in tissues are also employed. For instance, undesirable mutations in skin epithelial cells are eliminated from the skin via differentiation[5,57]. In contrast, proliferative advantage, increased cell density and hyper-thickening of the epidermal architecture have been documented to confer a fitness advantage that allows cells to persist within the tissue. Thus, cell-competition phenotypes that we study in this manuscript in vivo arise from interactions of mutant cells with the WT cells in the context of the tissue architecture.

The language of cell competition ('winner', 'loser' and 'fitness') implies the existence of an ideal state, in this case, an ideal redox ratio that characterizes the cells of a tissue. While we do uncover metabolic signatures that are maintained during homoeostasis in the tissue, our study also shows that winners and losers are context dependent. Thus, metabolic plasticity plays an important role in the maintenance of homoeostasis in the epidermis. Alterations in redox ratio and glucose catabolic flux precede changes in proliferation and morphological aberrancies. Hence, manipulating redox state can modulate the cell-competition outcome, which has implications for therapeutically eliminating oncogenic mutations from the skin epidermis.

## Online content

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

## Methods

This study complies with all relevant ethical regulations and all animal procedures were approved by the Yale University Institutional Animal Care and Use Committee under protocols 11303 and 20290.

### Mice

*K14CreER*[58], *Rosa26-CAG-LSL-H2B-mCherry*[59] mice were obtained from the Jackson Laboratory, *R26p-Fucci2* (ref. [60]) mice were obtained from S. Aizawa (RIKEN), *β-catenin*[flox(Ex3)] (ref. [61]) mice were obtained from M.M. Taketo (Kyoto University) and *Hras*[G12V] (ref. [62]) mice were obtained from S. Beronja (Fred Hutch) and bred with CD1 mice for a white background. Primers used for genotyping are listed in Supplementary Table 2. Fucci2-positive mice were identified using fluorescent goggles. For live imaging and revisits after mutation induction of *K14CreER*; *β-catenin*[flox(Ex3)/+]; *LSL-H2BmCherry* and *K14CreER*; *flox and replace(FR)-HrasG12V/+*; *LSL-H2BmCherry* mice, a single intraperitoneal injection of tamoxifen (1 mg in corn oil) was given to induce recombination near age P45 (second telogen) of around 80%. For full induction (-100%) and expression of mutant alleles (MS studies), a single intraperitoneal injection of tamoxifen (2 mg in corn oil) was given at around P45. Mice of either sex were randomly selected from experimental and control groups for live imaging experiments. No blinding was carried out for data collection and analysis. All animal procedures were approved by the Yale University Institutional Animal Care and Use Committee and housed and fed according to approved protocol 11303. The water and chow (2018SC Rodent diet from Inotiv) were autoclaved before use. The housing used was consistent with the Guide for the Care and Use of Laboratory Animals and compliant with the Animal Welfare Act and Regulations. Mice were housed on ventilated Tecniplast lixit racks with ambient temperature of 22 °C and 50 ± 10% humidity with a 12-h light–dark cycle (lights on 7:00–19:00) and fed ad libitum. Tumours were not expected to appear in the experimental time window.

### In vivo imaging

Preparing the skin for live imaging and the custom live-imaging platform were similar to those described previously[35,37]. Imaging was conducted in the distal regions of the ears during the second telogen (-postnatal day (P)45–65) after hair removal (using depilatory cream, Nair) 4–5 days before the experiment. Imaging from nonhairy skin was carried out from the paw. The paw or ear of each mouse (placed on a heating pad as part of a custom-built stage) was mounted with a glass coverslip placed directly against the tissue imaged. Mice were anaesthetized using a vaporized isoflurane chamber and anaesthesia was maintained throughout with a nose cone supplying isoflurane and oxygen. Image stacks were acquired with a LaVision TriM Scope II (LaVision Biotec) laser-scanning microscope equipped with a Chameleon Vision II and Discovery ultrafast lasers (Coherent) for different wavelengths. Serial optical sections were imaged using a ×40 water immersion lens (Nikon; NA 1.15) with a scan field of view of 300 μm × 300 μm at 400 Hz and pixel size of 0.2 μm or 0.3 μm and a z-step of 0.5 μm or 2–3 μm (large mosaics). Large regions of skin (>300 × 300 μm) were captured by scanning adjoining scan fields and stitching together images post-acquisition. Wavelengths of 750 nm and 890 nm were used to collect intensities at blue range (425–475 nm; NAD(P)H) and green range (500–550 nm; FAD). In addition, 940 nm (GFP; Fucci2) and 1,040 nm (572–647.5 nm; mCherry) were used and second harmonic signal was captured in the blue range at 890 nm or at 1,040 nm (green range). Laser power at sample was measured periodically and maintained between 25–35 mW. Line averaging of 2 was carried out to improve the signal-to-noise ratio. Throughout the manuscript, we have used the ratio of NAD(P)H intensities to FAD intensities as the redox ratio. The definition of redox ratio is not consistent within the optical redox imaging literature and should be interpreted according to what is measured. In this manuscript, a more oxidized redox ratio indicates a drop in NAD(P)H/FAD ratios and more reduced ratios indicate increased NAD(P)H/FAD.

To revisit the same epidermal regions over many days, micro-tattoos (applied 4–5 days before imaging) and inherent landmarks in the skin (recognizable array of hair follicles distributed in rows or clusters or prominent blood vessels) were used to navigate[37]. For example, in Extended Data Fig. 1b, the right image shows a mouse mounted for imaging. The skin area to be imaged (ear) was mounted under a coverslip and a tattoo (made 5 days before imaging) was placed in relation to visible landmarks, such as blood vessels, to identify the larger (1,800 × 1,800 μm² 6 × 6 stitched tiles each of 300 μm²) regions. The characteristic pattern of hair follicles in relation to the tattoo was used to identify the same region of skin and imaged as a revisit (reviewed previously[7]). While revisiting a mouse, imaging conditions were kept constant for NAD(P)H and FAD between day 0, day 4–6 and day 9–13 and day 15, as shown in the manuscript.

### Image quantification

For isolating the fluorescence intensities from the basal stem cell layer, the signal from collagen (second harmonic) was first used to make an intensity threshold-based mask, wherein all pixels below the epidermis–dermis interface (identified by the appearance of collagen intensities) were included using MATLAB-based code. This image with the undulating dermis mask was then imported into Imaris 3D processing software. The dermis mask was used to generate a surface and the distance (0 to -2.5–3.0 μm for cytosolic NAD(P)H; 0–5.0 μm for nuclear H2BmCherry) from this surface was used to obtain fluorescence intensities of the basal layer using the distance-transform function in Imaris, such that overlap from suprabasal cell outlines was avoided. Such selected fluorescence signals from the z-stack were projected using maximum intensity projection or mean intensity projection to obtain images of the basal layer (Figs. 1c, 2c,d, 3a,b,e and 4a,b and Extended Data Figs. 1d, 2c and 4a). Regions were manually drawn in ImageJ around individual cells recognized by both NAD(P)H and/or nuclear signal for quantification of mean intensities per cell. The NAD(P)H intensity value at each pixel was divided by the FAD intensity value to obtain an NAD(P)H/FAD redox image using the Image calculator function of ImageJ. In Fig. 3e, basal layer projections were obtained as described above and to quantify mutant cell coverage, regions were manually drawn around mCherry-negative areas and represented as a percentage of the total area after excluding hair follicles. For quantification of mutant cell coverage from large mosaics, shown in Extended Data Figs. 3 and 9, z-sections were imported into Imaris and straightened such that the dermal–epidermal interface fell along a straight line. The z-sections above and below the epidermal–dermal interface were then projected to obtain a maximum projection of epidermis (A) and z-sections below the epidermal–dermal interface (B). Regions were then drawn around mCherry-positive areas and their coverage calculated as a percentage of the total area. Images used for representation were filtered using the smooth function of ImageJ. Per-cell redox values were normalized to average baseline control value (day 0 for time-lapse revisits; WT cells for comparison of WT to mutant cells in the same animal) from each animal before plotting differences shown in the accompanying graphs. The upper and lower values of LUT bars shown accompanying each experiment represents arbitrary intensity units used to scale both control and mutant/drug-treated model images to the same range.

For quantification of proliferation, 4 × 4 or 5 × 5 mosaics were acquired at 300 μm² each and stitched in ImageJ. Multiple regions (sizes are mentioned in the figure legends, usually 400 μm²) were cropped from each mouse and the pH3-positive nuclei were counted while scanning through the z-stack to account for unevenness in the preparation of tissue. Similarly for the density analysis, the number of nuclei in the basal layer was counted while scanning through the z-stacks from regions (sizes are mentioned in the figure legends; 200 μm²) cropped from large, stitched tiles.

For quantifications of thickness from epidermal preps (Fig. 6d,g) in Imaris, an artificial surface was created just above the cornified layer

of the whole-mount section. A new distance transformation channel, with values representing distance from the artificial surface, was created using the distance transformation plugin. A final surface was made from the autofluorescence signal present within the tissue due to fixation. This surface was used to mask the distance transformed signal. A maximum intensity projection of the masked distance transformed signal was used to determine the average thickness of the whole-mount sample.

For quantifications of thickness from live imaging of z-stacks (Fig. 7c,e), a dermis mask was first made using MATLAB and Imaris using the collagen signal from second harmonic signal as described before. An artificial air surface was then created manually above the cornified layer of the entire image using the background signal from K14CreER-driven LSL-H2BmCherry. A distance transformation from the air surface was used to create a channel representing the thickness of the tissue. The dermis surface was then used to mask out all the signal from the dermis so that the distance transformed channel shows the distances from the air interface to the dermis–epidermis interface. A maximum intensity projection of the masked distance-transformed signal was used to determine the average thickness of the live-imaged sample.

ImageJ, MATLAB, Imaris, GraphPad Prism and Microsoft Excel were used in the analysis of data.

## Untargeted metabolomics

Untargeted metabolomics analysis was performed through the Yale IOMIC Chemical Biology Core. The epidermis was isolated quickly from mutant and control mice following killing with IV Euthasol (pentobarbital/phenytoin). The skin was immediately collected from the ears by first removing the hair (using Nair) and isolating the top layer (side of the ear imaged in NAD(P)H, FAD measurements). The epidermis was then separated from the dermis by incubating in 5 mg ml$^{-1}$ Dispase solution in PBS for 10–12 min, after which the epidermis was separated manually and flash frozen in liquid nitrogen. The epidermal samples were then lyophilized and subjected to Dounce homogenization. Then, 200 µl ice-cold extraction solution (50% methanol + 50% water + 5.5 µg ml$^{-1}$ $^2H_8$-phenylalanine for internal standard) was added to each sample, and samples were subjected to three cycles of vortexing for 30 s, then rested on ice for 10 min. Samples were then centrifuged and the supernatant was transferred to a new tube, with 800 µl ice-cold water then added. After freezing at −80 °C overnight and lyophilization, 50 µl of 10% acetonitrile solution with $^2H_4$-taurine (25 µM) as a second internal standard was added. Samples were again centrifuged and the supernatant transferred immediately to an LC–MS/MS plate. Then, 5 µl of the supernatant was injected for each analysis mode on the mass spectrometer.

LC–MS/MS analysis was performed on the Sciex TripleTOF 6600. Two columns were used separately: a Thermo Scientific Hypercarb column (100 × 4.6 mm, 3 µm) and a Phenomenex Kinetex F5 Core-shell LC column (100 × 2.1 mm, 2.6 µm). The metabolite peak area was curated using El-MAVEN software.

## Metabolomics pathway analysis

The metabolites upregulated and downregulated in the epidermis with HrasG12V or βcatGOF were imported into MetaboAnalyst 5.0 (ref. 63). These metabolites were analysed using the hypergeometric test and relative-betweenness centrality and were visualized in a scatter-plot. The metabolic pathways that are significantly altered and have more than one match status were plotted in the figure.

## Targeted in vivo glucose tracer analysis

One week before in vivo $^{13}C_6$- glucose infusion, mice underwent surgery under isoflurane anaesthesia to place polyethylene catheters in the right internal jugular vein and advanced into the right atrium. Buprenorphine analgesia was provided in surgery, and carprofen for 72 h thereafter. After recovery and following an overnight fast, mice were provided a primed-continuous infusion of U-$^{13}C_6$ glucose (prime 3 mg kg$^{-1}$ min$^{-1}$ for 5 min, continuous infusion rate 2.0 mg kg$^{-1}$ min$^{-1}$

for a total of 120 min), after which they were killed with IV Euthasol (pentobarbital/phenytoin). We confirmed that isotopic steady state is achieved in the skin within 2 h of infusion (Extended Data Fig. 6g–j). Skin was immediately collected from the ears by first removing the hair (using Nair) and isolating the top layer (to be similar to the tissue imaged in NAD(P)H, FAD measurements). The epidermis was then separated from the dermis by incubating in 5 mg ml$^{-1}$ Dispase solution in PBS supplemented with 5 mM $^{13}C_6$ glucose for 10–12 min after which the epidermis was separated manually and flash frozen in liquid nitrogen to follow the steps for MS as described below. Care was taken to ensure that the epidermis was isolated and frozen within half an hour of killing each mouse.

$^{13}$C enrichment of key metabolites was measured by gas chromatography–MS (glucose[34], pyruvate[38], lactate[34] and $CO_2$ (ref. 64)) and by LC–MS/MS (acetyl-CoA[65], $C_4C_5$ glutamate[34]). The ratio of PDH/CS flux (the fraction of the TCA cycle fuelled by glucose), can be measured as $^{13}C_2C_4C_5$ glutamate/$^{13}C_3$ alanine (Fig. 5a)[34,66]. To the extent that the ratio $^{13}C_2 C_4C_5$ glutamate/$^{13}C_3$ alanine differs from $^{13}C_2$ acetyl-CoA/$^{13}C_3$ alanine, this would reflect dilution of labelled glutamate by unlabelled glutamine via flux through glutaminase; a lack of difference between these ratios, as observed in the current study, reflects negligible net carbon flow through glutaminase. LC–MS/MS (AbSCIEX 6500 QTRAP with a Shimadzu ultrafast liquid chromatography system in negative-ion mode) was used to monitor the ion pairs: [m0] $C_4$ and $C_5$ glutamate 146/41 and [m + 2] $C_4$ and $C_5$ glutamate, 148/48).

## Metformin administration

Adult mice at P41 were given metformin in drinking water at 2 mg ml$^{-1}$ (refs. 45,48) in light-protected bottles, while mutant littermates for comparison as untreated/control were given only drinking water. Then, 5 days after metformin treatment was started, tamoxifen was administered, and the development of mutant phenotypes was imaged as described above.

## DCA administration

Adult WT mice at P41 were given DCA in drinking water at 0.5 mg ml$^{-1}$ in light-protected bottles, while littermates were given only drinking water. After 5 days of DCA treatment, mice were imaged for redox changes as described above.

## Cyanide experiments

In the 293T cell line (obtained from Thermo Fisher, 293FT cell line cat. no. R70007), NAD(P)H and FAD were imaged in live cells. Sodium cyanide diluted in PBS (to reach a final concentration of 4 mM) was added to the dishes without removing them from the stage and images were captured immediately after (within 5 min). In WT mice, the ear was mounted and images were acquired within a 6 × 6 mosaic (1,800 µm$^2$ identified and mapped using by a tattoo, made 5 days before). Then, 10 µl of 40 mM sodium cyanide was injected intradermally near the tattoo (outside the imaging area). The mice were quickly remounted and imaged in the previously identified mosaic region within 5–10 min. As the priority was to take a z-stack soon after injection, we did not attempt to find the same cells that were previously imaged but made sure that images after cyanide injection were within the previously identified 6 × 6 mosaic approximately at the same position with respect to the tattoo. This process was repeated for all three replicates. Mice were killed after 10 min of injection and imaging.

## Epidermal prep and staining

The dorsal ear flap was separated from the cartilage and was incubated in Dispase solution (5 mg ml$^{-1}$; Sigma) at 37 °C for 15 min following which, the epidermis was separated from the dermis using forceps. This epidermal tissue was fixed in 4% paraformaldehyde (in PBS) for 30 min at room temperature. The tissue was then washed, permeabilized and blocked using 2% Triton X, 5% normal donkey serum, 1% BSA in

PBS solution. This was followed by incubation in primary antibodies overnight at 4 °C followed by incubation in secondary antibodies for 1 h at room temperature. These antibodies were used: mouse anti-β-catenin (1:100 dilution; BD 610153), rabbit anti-phosphohistone H3 (Ser10; 1:300 dilution; Millipore 06-570), guinea pig anti-keratin 10 antibody (1:200 dilution; ARP 03-GP-K10) donkey anti-mouse AF488 (1:300 dilution; Thermo Fisher A-21202) and goat anti-rabbit AF633 (1:300 dilution; Thermo Fisher A-21071), goat anti-guinea pig AF546 (1:300 dilution; Thermo Fisher A-11074). Fixed tissue was then mounted in Vectashield with 4,6-diamidino-2-phenylindole (Vector; H1200) underneath a no. 1 coverslip.

## Statistics and reproducibility

The number of cells and mice used in each plot is reported in the figure legends. No animals were excluded, except when measurements could not be completed on a mouse due to variations in laser power or other variations in imaging conditions during a revisit. No statistical methods were used to predetermine sample sizes but our sample sizes are similar to those reported in previous publications[5,10,32,33,37,67,68]. All statistical tests were carried out in GraphPad Prism. Data distribution was assumed to be normal, but this was not formally tested. For measurements where two groups were measured, a two-tailed *t*-test or nested *t*-test was used, as detailed in the figure legends. For measurements involving three or more test groups with respect to controls, a one-way ANOVA was used with multiple comparisons against the WT mice. For violin plots of NAD(P)H/FAD, per-cell redox ratios normalized and pooled from three mice (or more as mentioned in legends) were plotted as violin plots. *P* values were calculated after taking averages per mouse replicate (plotted as coloured points overlaid on the violin plots using the layout option in GraphPad Prism) to depict variations between biological replicates. For measurements where the same regions from same mice were revisited and compared (Fig. 3), paired *t*-tests or repeated measures ANOVA were used and detailed in the figure legends. Box plots were plotted using GraphPad Prism with whiskers going from minimum to maximum, showing all points. The box extends from the 25th to 75th percentiles with the central line at the median. All microscopy images shown, including Figs. 1a, 2a, 3a,d, 4a,b and 6a,c,d and Extended Data Figs. 1b, 2a,b,d and 7c are representative of data from three mice each.

## Reporting summary

Further information on research design is available in the Nature Portfolio Reporting Summary linked to this article.

## Data availability

All data supporting the findings of this study are available from the corresponding author on reasonable request. Source data are provided with this paper.

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

## Acknowledgements

We thank all members of the Greco laboratory, especially T. Xin for critical feedback on the manuscript as well as advice on technique and tool development and D. Monedero-Alonso for help with the text. We thank Q. Sun, R. Cardone and the IOMIC core at Yale for metabolomics experiments. We thank H. Zhao and B. Cai for consultation on statistics. This work was supported by National Institutes of Health grant nos. 1R01AR063663-01, DP1AG066590-01 and R01AR072668 and a Leo Foundation Grant (all to V.G.) and 1R37CA258261-01A1 (to R.J.P.). A.H. is supported by the New York Stem Cell Foundation Druckenmiller Fellowship.

## Author contributions

A.H., R.J.P. and V.G. designed experiments. A.H. performed two-photon imaging, mouse genetics and image analysis. Z.L., R.J.P. and A.H. performed mass spectrometric studies. M.S., D.G. and D.G.G. advised and assisted with optimizing redox imaging within our in vivo imaging platform. D.G.G. assisted with development of analysis tools and image analysis. C.M., E.L. and K.T. performed whole-mount staining for proliferation, density and image analysis. S.G. assisted with mouse genetics. A.H., R.J.P. and V.G. wrote the manuscript with input throughout from L.G.

## Competing interests

The authors declare no competing interests.

## Additional information

**Extended data** is available for this paper at https://doi.org/10.1038/s41556-024-01574-w.

**Correspondence and requests for materials** should be addressed to Rachel J. Perry or Valentina Greco.

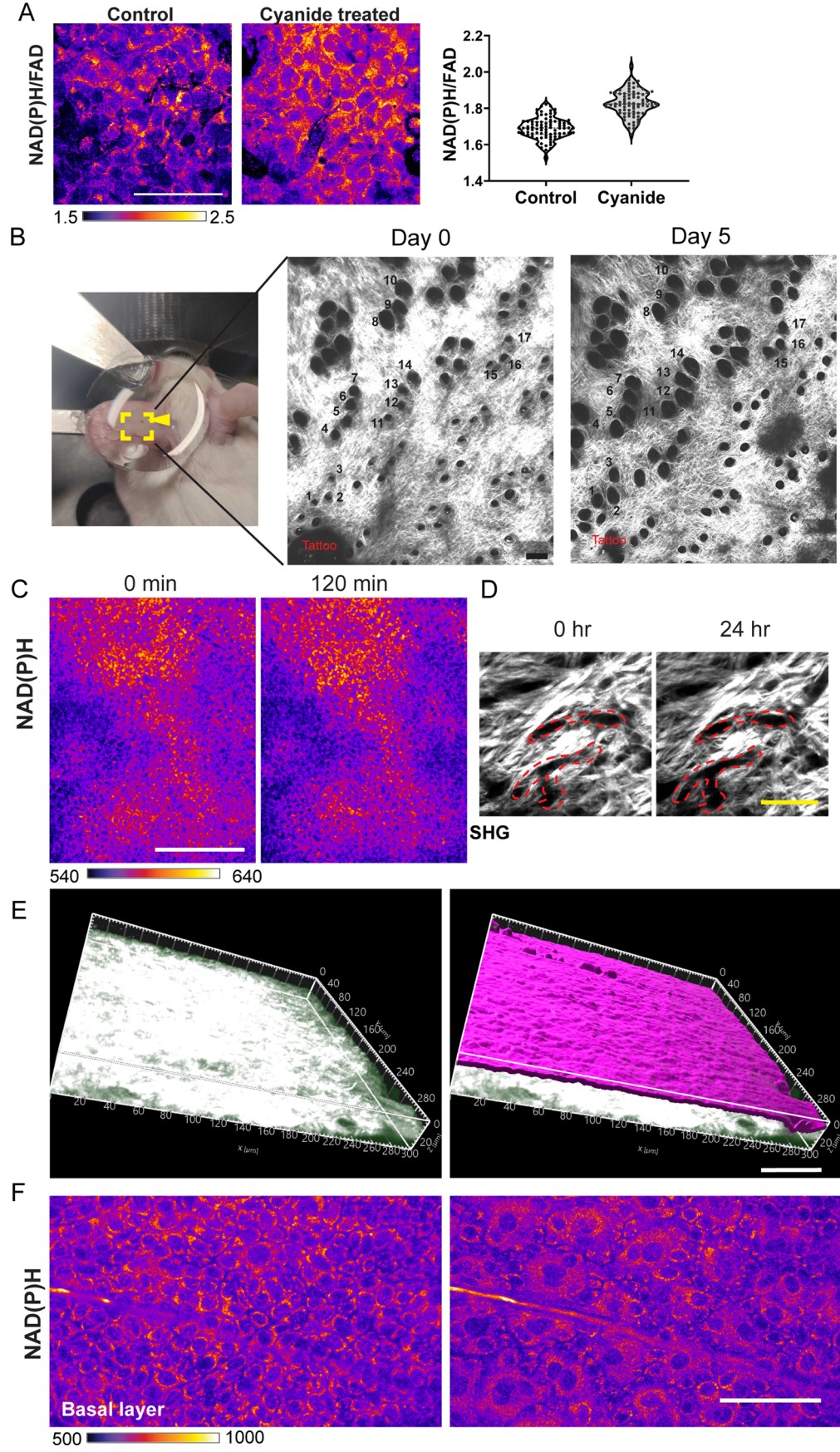

**Extended Data Fig. 1 | See next page for caption.**

**Extended Data Fig. 1 | Imaging NAD(P)H and FAD. a.** NAD(P)H/FAD intensities from HEK293T cells treated with 4 mM Sodium cyanide imaged after 5 minutes of adding cyanide. NAD(P)H/FAD ratio rapidly increases in response to the inhibition of Complex IV and oxidative phosphorylation, as expected, because blockage of mitochondrial electron transport leads to accumulation of NAD(P)H in the cells. n(In order x axis) = 86, 78 cells. representative of 2 independent replicates **b.** Revisits- Skin area to be imaged (ear) is flattened under a coverslip and a tattoo (made 5 days before imaging) placed at a stereo-typical distance from visible landmarks like blood vessels is used to identify the larger (1800×1800 μm² 6 by 6 stitched tiles of 300 μm²) regions. The characteristic pattern of hair follicles (please see numbers labelled in Day 0 and Day 5 images for easier identification of the same follicles) in relation to the tattoo is used to identify the same region of skin and imaged as a revisit. Scale bar = 100 μm **c.**

The same region of the basal epidermis is imaged over 2 hours from non-hairy epidermis (paw) to show that pattern of NAD(P)H per-cell intensity are stable in their spatial distribution although there are small fluctuations in intensity. **d.** The second harmonic signal (SHG) from collagen in regions from Fig. 1c and d at 0 hours and revisited after 24 hours. The collagen fibrils and blood vessels (outlined in red dotted line) are used to identify the same region of epidermis. **e.** 3D projection from the Imaris software used to isolate the basal layer. After thresholding and surfacing the second harmonic signal (SHG) using the Distance-Transform function in Imaris, pixels from selected distances from the epidermal-dermal interface are isolated and projected to isolate the basal layer. **f.** NAD(P)H from Basal (left) and suprabasal (right) layers isolated at different distances from the epidermal-dermal interface as described in A-F. **c**–**f**, Scale bar, 50 μm.

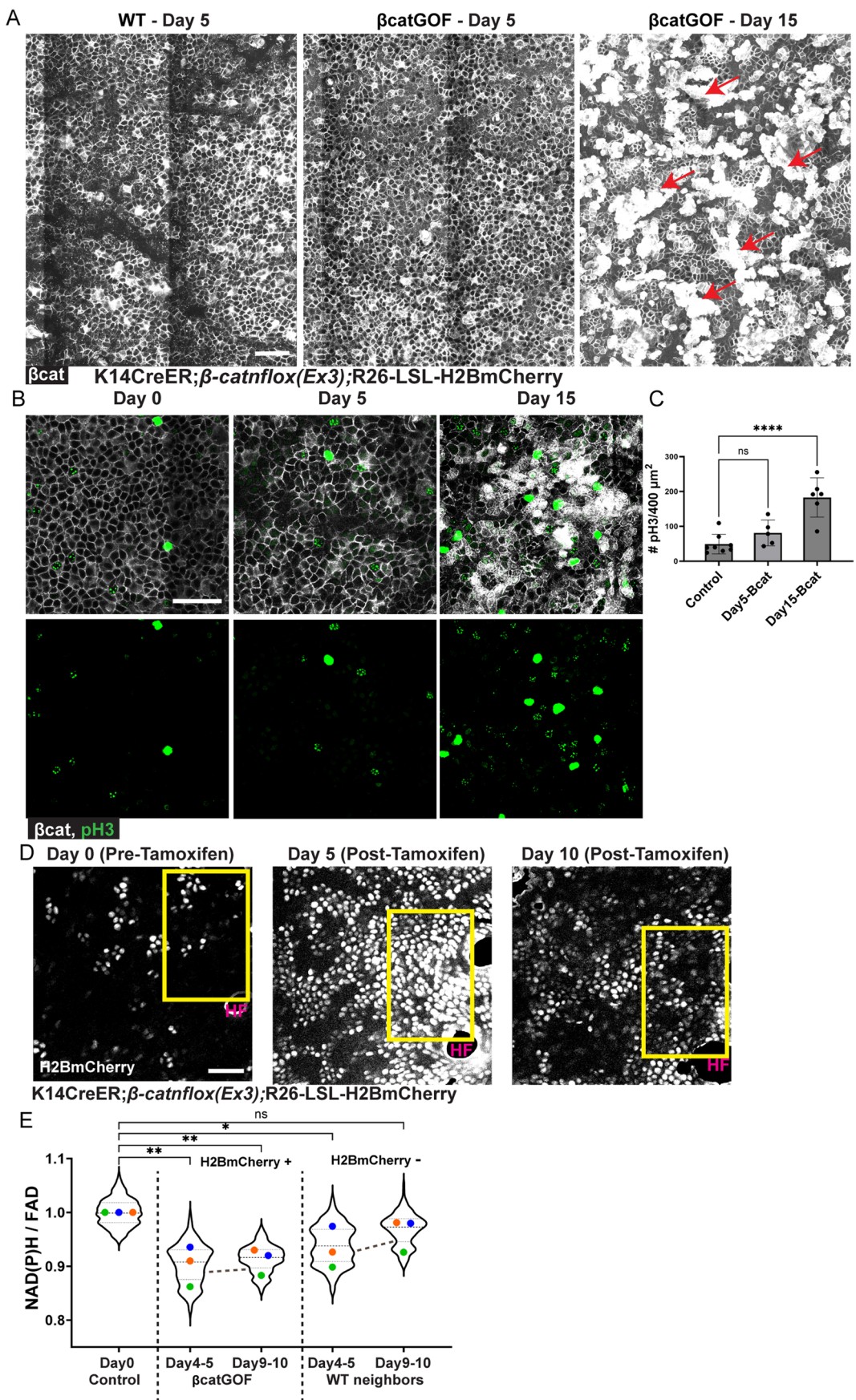

**Extended Data Fig. 2 | See next page for caption.**

**Extended Data Fig. 2 | Morphological and Behavioural changes in the βcatGOF mutant epidermis follow redox changes. a.** β-catenin immunofluorescence staining (grayscale) in the basal layer of epidermis from large regions in mice of genotype *K14CreER; βcatGOF* 5 and 15 days' post-tamoxifen-induced recombination and expression of βcatGOF, similar to Fig. 2a. Control shown is from littermate without mutation 5 days post-tamoxifen. The morphology of the basal layer is indistinguishable between Day 0 and Day 5. However, at Day 15, several aberrant structures (placodes) are visible (examples: red arrows) in the basal layer with stacked layers of nuclei enriched in β-catenin. More magnified images of these regions are in Fig. 2a. **b.** phosho-Histone 3 (pH3-green) and β-catenin staining in epidermis at Day 0 (before tamoxifen), Day 5, and Day 15 post-tamoxifen. **c.** pH3 positive nuclei were counted and quantified from 400×400 µm² regions averaged from multiple mice. n = 8 (Control), 5(Day 5 βcatGOF), 6 (Day 15 βcatGOF). **** = p value < 0.0001, ns = not significant (One-way ANOVA; Multiple comparisons with respect to Control/Day 0). Data are presented as Mean ± SD **d.** The larger regions from which the epidermal revisits magnified in Fig. 3a (outlined in yellow) are shown. White nuclei (H2BmCherry-positive) label the cells that are recombined and hence are considered βcatGOF mutant in mice of genotype *K14CreER; βcatGOF; LSL-H2BmCherry*. **e.** Same data in Fig. 3c plotted in comparison to Day 0. Cells expressing βcatGOF (H2BmCherry+) have low NAD(P)H/FAD ratio at Day 4–5 and 9–11 post-tamoxifen. WT cells (H2BmCherry -) recover their NAD(P)H/FAD ratio at Day 9–11 (not significantly different from control). Average redox (coloured dots) from 3 mice each. p value for Control vs all categories (RM One-Way ANOVA; Multiple measures) **<0.0023; *= 0.0133; ns- not significant. Violin plot n(in order x axis)=379, 323, 198, 273, 227 cells from 3 mice. Scale bar= 50 µm.

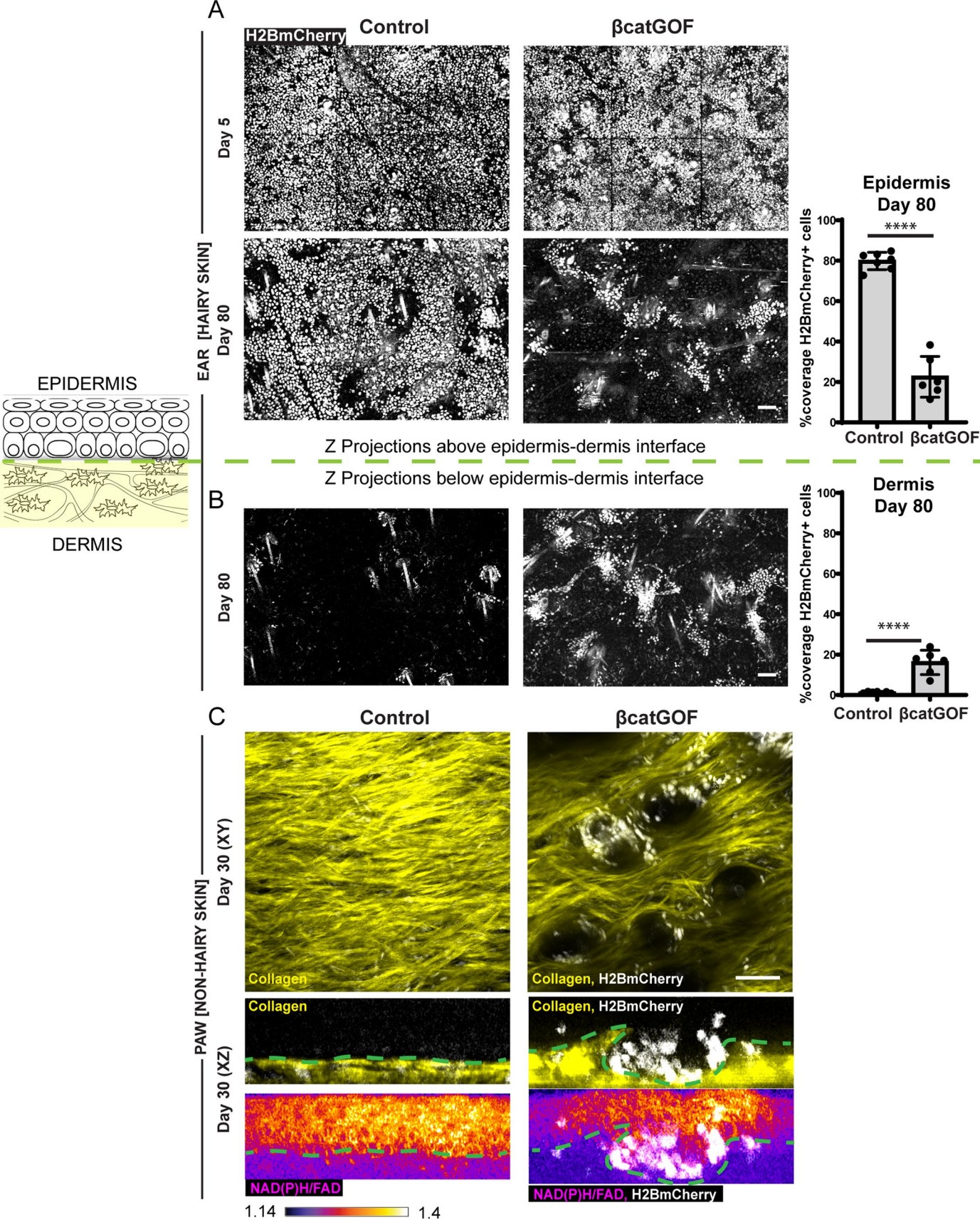

**Extended Data Fig. 3 | See next page for caption.**

**Extended Data Fig. 3 | WT cells outcompete βcatGOF after 1–2 months while the majority of the remaining βcatGOF cells are found in placodes that extend into the dermis. a**. Z-projections from large regions of skin above the epidermal-dermal interface from Control (genotype: *K14CreER; LSL-H2BmCherry*) and βcatGOF-induced mice (genotype: *K14CreER; βcatGOF; LSL-H2BmCherry*) shows similar rates of recombination, with recombined cells (mCherry+; white nuclei) representing control (left) or mutant cells (right) 5 days after recombination covering ~80% of the epidermis five days post-tamoxifen. However, after two months (bottom panel), while control tissue shows no major differences in the coverage of recombined white nuclei, large regions of βcatGOF epidermis are free of the recombined mutant cells (white nuclei). The **graph** on right quantifies the percentage of area covered by recombined cells at 2 months with control epidermis showing 80% occupancy (graph right), while βcatGOF epidermis has

only ~20% of their area covered by recombined mutant cells. **b**. The majority of recombined mutant cells (H2BmCherry-positive; white nuclei) extend in sub-epidermal structures taking up ~20% of the sub-epidermal space compared to negligible coverage of H2BmCherry-positive epidermal cells in controls. A-B- ****= p value < 0.0001 (2-sided t-test) from ~1200–1400 μm² regions n = 7 (Control), 6 (βcatGOF) regions from 3 mice. Data are presented as Mean ± SD. **c**. In non-hairy skin (paw), the βcatGOF-induced cells (H2BmCherry-positive; white nuclei) are found in long-lived hair follicle-like structures in the βcatGOF-induced epidermis (right). In contrast, in control tissue (*K14CreER*) the second harmonic signal (SHG) from collagen in dermis (yellow) is uninterrupted by hair follicles. XZ sections show the placode-like structures with βcatGOF cells that extend into the dermis with an altered NAD(P)H signal that interrupts the collagen (SHG). Scale bar=50 μm.

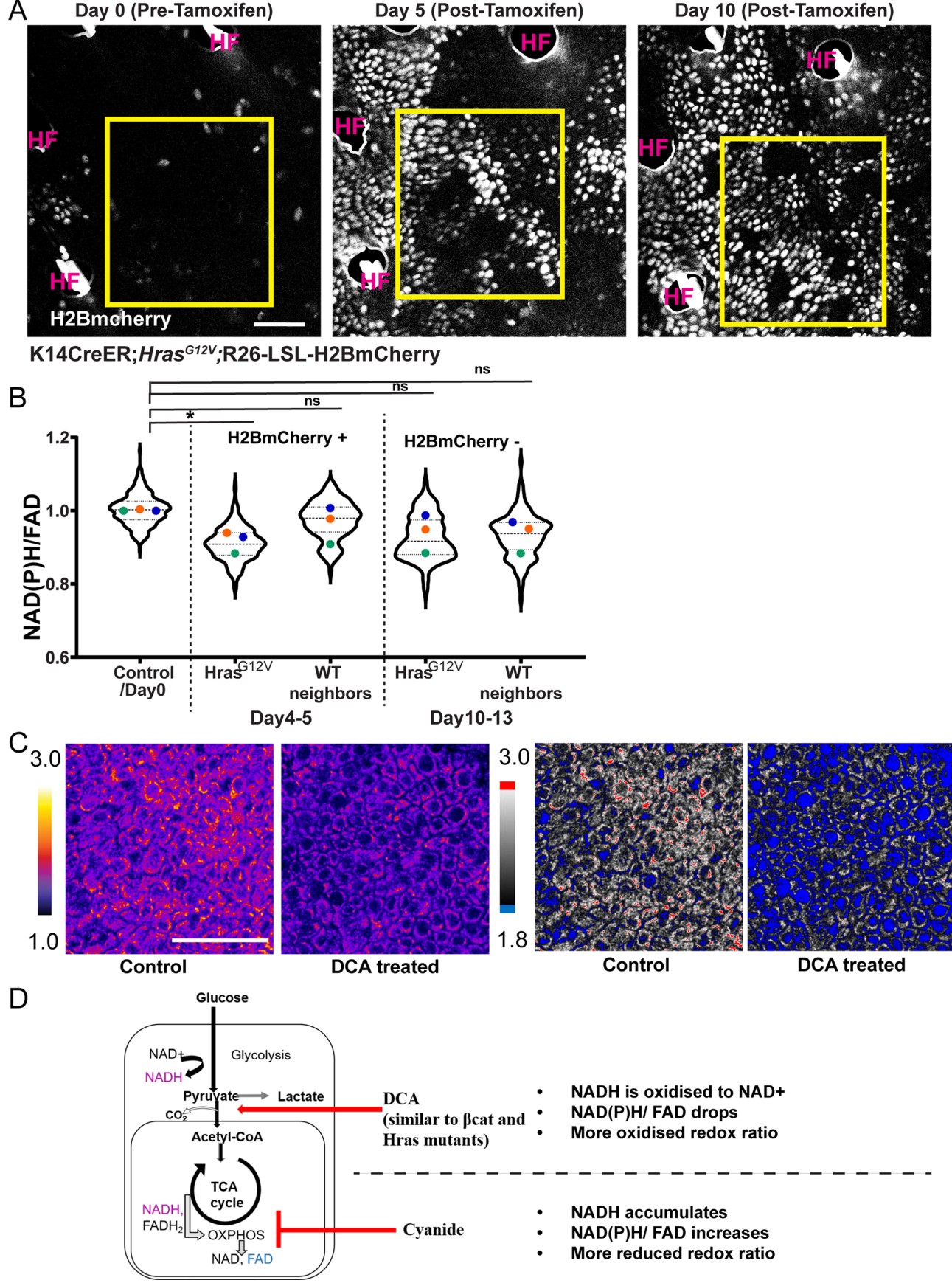

**Extended Data Fig. 4 | See next page for caption.**

**Extended Data Fig. 4 | Revisits of the same epidermal regions shows different cell-competition phenotypes and redox changes in the Hras^G12V mutant model. a**. White nuclei (H2BmCherry-positive) label cells that are recombined and carry the Hras^G12V mutation in mice of genotype *K14 CreER; Hras^G12V; LSL-H2BmCherry*. **b**. Same data in Fig. 4c plotted in comparison to Day 0. NAD(P)H/FAD intensities from Hras^G12V cells within the epidermis at Day 4–5, and Day 10–13 post-tamoxifen compared to Day 0 or the control littermate (*K14CreER;LSL-H2BmCherry)* imaged on the same day under same imaging conditions. Average redox of only Day 4–5 Hras^G12V cells are significantly different from Control. Day 4-5 WT and Day 10-13 Hras^G12V and WT cells are not significantly different. The Hras^G12V cells have a redox differential at Day 4-5 that is flattened by Day 10-13 as plotted in Fig. 4c. p value (2-sided t-test) *= 0.011 (n = 3 mice each -coloured dots). Violin plot n(in order x axis)=317, 316, 129, 347, 152 cells from 3 mice. **c**. Images showing NAD(P)H/FAD intensities from the basal stem cell layer of control animals and animals given DCA (Dichloroacetic acid) for 5 days in Fire(left) and HiLo(right) LUT scales showing reduction in DCA-treated mice. **d**. Schematic explaining the direction of redox change and the corresponding changes in glycolysis, TCA cycle and mitochondrial oxidation. The more reduced redox ratio (high NAD(P)H/ FAD) corresponds to inhibition of mitochondrial oxidation and conversely more oxidized redox ratio (low NAD(P)H/ FAD) corresponds to increased glucose oxidation. Scale bar= 50 μm.

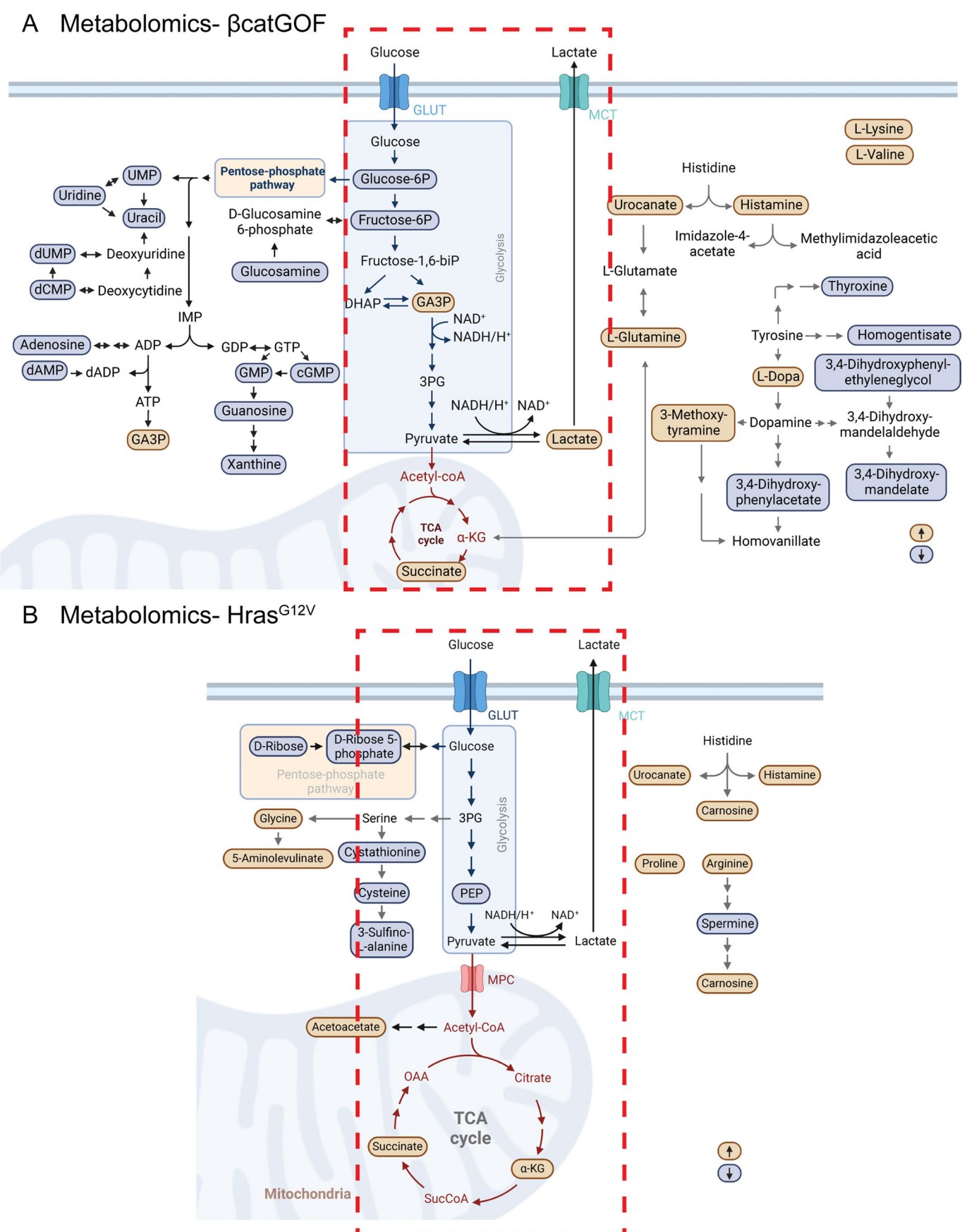

**Extended Data Fig. 5 | See next page for caption.**

**Extended Data Fig. 5 | Metabolite pathways differentially enriched or depleted in βcatGOF and Hras$^{G12V}$ mutant epidermis.** Pathway enrichment analysis after untargeted metabolomics with differentially enriched or decreased metabolites in βcatGOF (A) and Hras$^{G12V}$ (B). Upregulated concentrations are indicated by orange colour and downregulated by blue colour. The region within the red dotted lines highlights the altered metabolites in glycolysis and TCA cycle. Lists of all differentially enriched or depleted metabolites and pathways that have more than 2 hits altered in βcatGOF and Hras$^{G12V}$ epidermis when compared to control are shown in Supplementary Table 1.

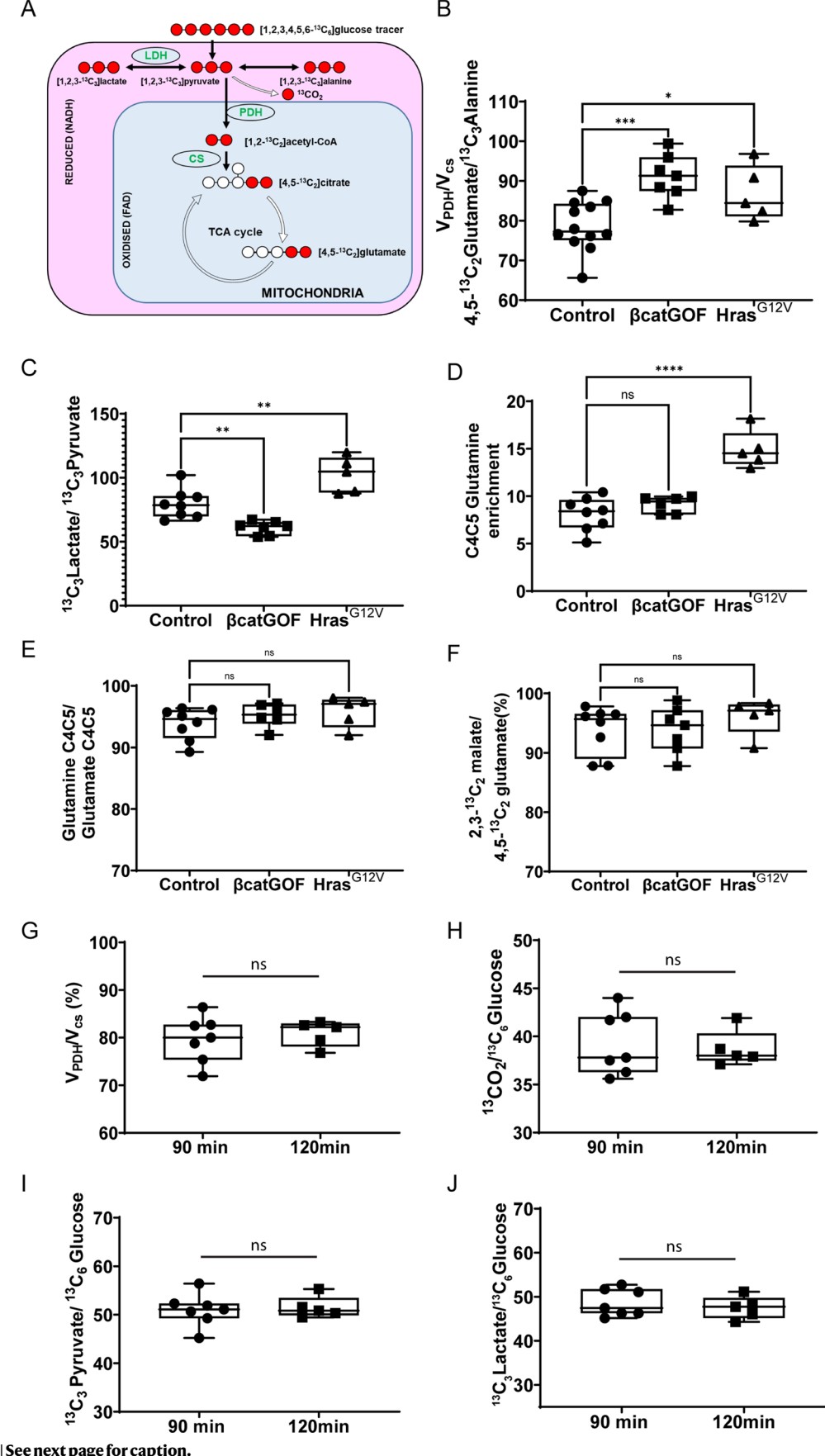

**Extended Data Fig. 6 | See next page for caption.**

**Extended Data Fig. 6 | Relative fluxes of metabolites after $^{13}C_6$- glucose infusion. a**. Schematic showing the labelling of carbon (red) in downstream metabolites when mice are infused with $^{13}C_6$ glucose- repeated from Fig. 5a. **b**. $^{13}C_2$Glutamate/ $^{13}C_3$.Alanine also a measure of $V_{PDH}/V_{CS}$, (or the percentage of TCA cycle fuelled by glucose) is upregulated in βcatGOF and Hras$^{G12V}$ mutant epidermis. n = 13(Control), 7(βcatGOF), 5(Hras$^{G12V}$) mice. p value *** = 0.0004. * = 0.0261. **c**. Less $^{13}C$-labelled lactate is generated from pyruvate in βcatGOF mutant epidermis and more is generated in Hras$^{G12V}$. n = 8(Control), 7(βcatGOF), 5(Hras$^{G12V}$)mice. p value **= 0.0058(βcatGOF); **=0.0024 (Hras$^{G12V}$). **d**. Percentage of labelled glutamine as a fraction of total glutamine is unchanged in βcatGOF when compared to wild-type littermates. The labelled fraction of glutamine increase in Hras$^{G12V}$ suggesting more carbon entry from glucose into glutamate and glutamine. n = 8(Control), 6(βcatGOF), 5(Hras$^{G12V}$) mice.

p value ****<0.0001. **e**. Labelled Glutamine/Glutamate ratio does not change significantly n = 8(Control), 6(βcatGOF), 5(Hras$^{G12V}$) mice **f**. $^{13}C$ label dilution between malate and glutamate, products of TCA cycle show very little dilution (>90%) and no significant change between the mutant models and control animals. n = 8(Control), 7(βcatGOF), 5(Hras$^{G12V}$) mice **g-j**- Mice infused with $^{13}C_6$ glucose for 90 minutes and 120 minutes *in vivo* show no significant differences in $V_{PDH}/V_{CS}$ (G), $^{13}CO_2/^{13}C_6$ Glucose (H), $^{13}C_3$-Pyruvate / $^{13}C_6$ Glucose (I) or $^{13}C_3$-Lactate / $^{13}C_6$ Glucose. Thus, isotopic steady state is achieved in the skin within two hours of infusion. n = 7(90 min), 5(120 min) mice. One -way Anova with Multiple comparisons with Control was used for B-F and 2-sided t-test for G-J. Each point on the plots represents a mouse replicate. All box plots show all data with whiskers going from minimum to maximum with line at the median.

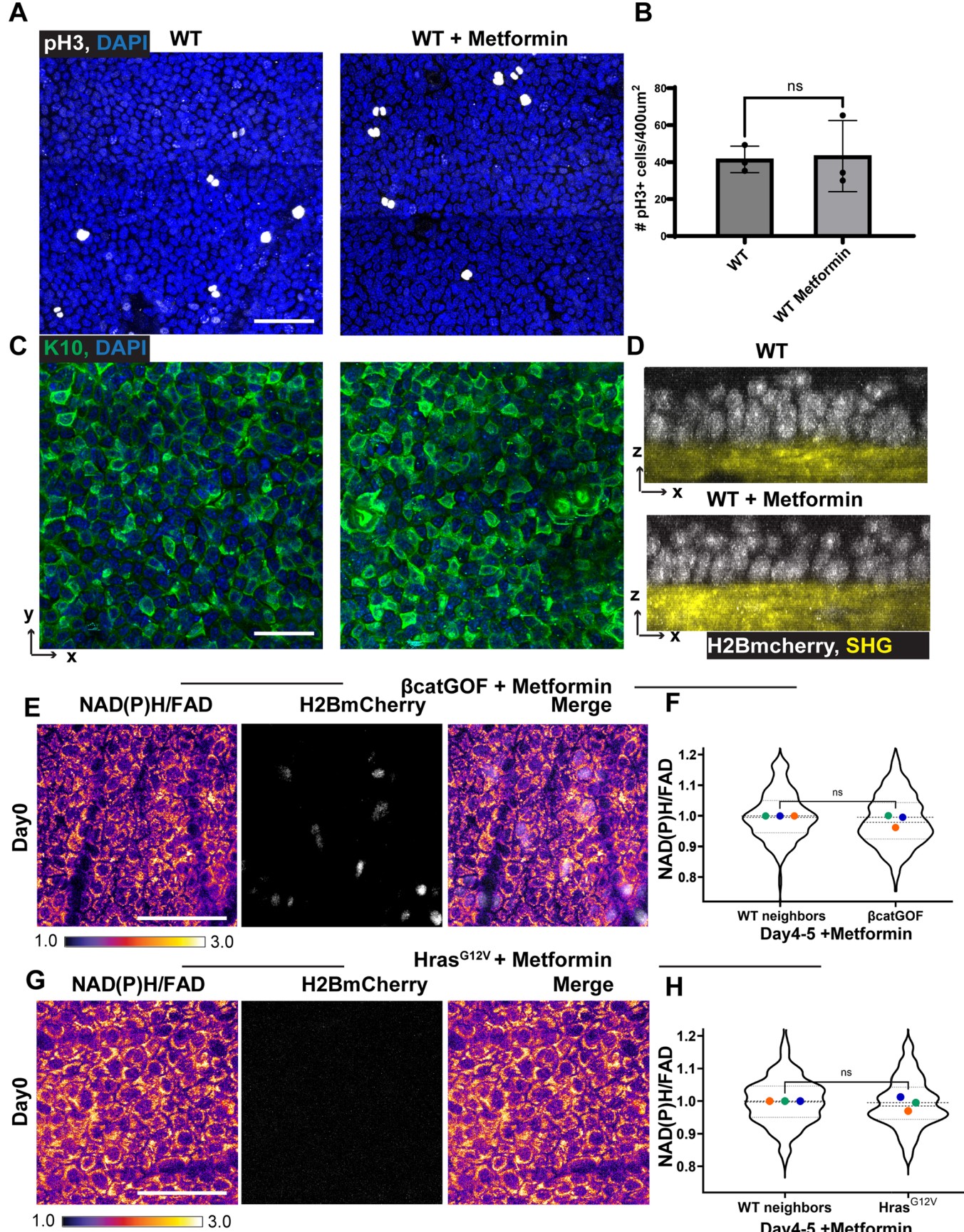

**Extended Data Fig. 7 | See next page for caption.**

**Extended Data Fig. 7 | Wild-type epidermis does not show changes in morphology or proliferation upon Metformin treatment. a-b**. Number of phospho-histone H3 (white) per 400×400 μm² was counted from wild-type epidermal preps and wild-type mice administered metformin similar to mutant conditions in Figs. 6 and 7 and plotted on the graph on right showing no differences (2-sided t-test) -numbers averaged from 3 regions each per mice; n = 3 mice each (plotted as dots). Data are presented as Mean ± SD. The morphology, and cell density of the basal layer also do not show any obvious changes. **c**. Keratin 10 staining of the basal layer (Max projection of 3 z-stack of 0.5 um step size) also do not show any obvious change in the pattern between control and Metformin-treated wild-type mice. **d**. Thickness of the epidermis as shown in the representative images in the z-sections do not show any obvious differences between control and Metformin-treated wild-type mice. **e**. Day 0 images of mutant mice revisited in Fig. 6a (βcatGOF + Metformin) **f**. Graphs comparing WT neighbours and H2Bmcherry+ βcatGOF mutant cells in the same animals derived from graph in Fig. 6B. 2-sided t-test of averages and nested t-test show no significant differences. n(left to right)= 200, 288 cells from 3 mice. **g**. Day 0 images of mutant mice revisited in Fig. 7a (Hras^{G12V}+ Metformin). **h**. Graphs comparing WT neighbours and H2Bmcherry+Hras^{G12V} mutant cells in the same animals derived from graph in Fig. 7B. 2-sided t-test of averages and nested t-test show no significant differences. n(left to right)= 136,196 cells from 3 mice. Scale bar = 50 μm.

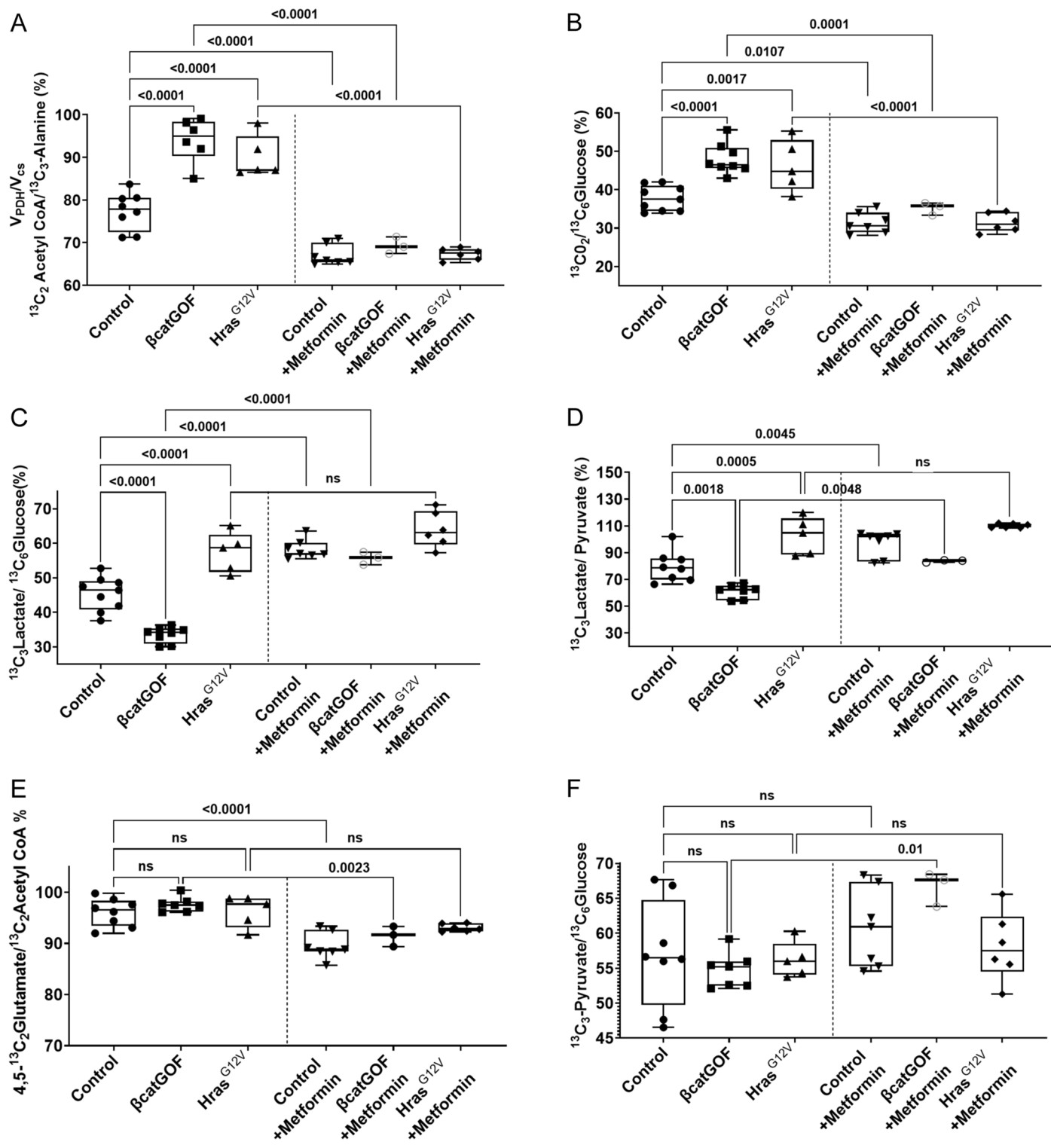

**Extended Data Fig. 8 | See next page for caption.**

**Extended Data Fig. 8 | Inhibition of metabolic rewiring upon Metformin treatment.** Graphs showing $^{13}C_6$ glucose-derived labelling in glycolysis and TCA readouts after in vivo infusion of $^{13}C_6$ glucose from epidermis of βcatGOF, Hras$^{G12V}$ and wild-type littermates with or without metformin administration. Please note that all data without metformin (left side of A-F) for Control, βcatGOF and Hras$^{G12V}$ animals ***are the same data*** shown in Fig. 5b–f and Extended Data Fig. 6c. They have been plotted here again only for comparison with animals when administered metformin. **a.** The ratio of $^{13}C_2$- Acetyl CoA/$^{13}C_3$ Alanine also called $V_{PDH}/V_{CS}$ that represents the percentage of TCA cycle fuelled by glucose, is reduced in Control, βcatGOF and Hras$^{G12V}$ animals treated with metformin (right half) when compared to animals without treatment (left- also in Fig. 5b). The differences between mutant and controls are flattened after metformin treatment. **b.** Another measure of TCA flux, $^{13}CO_2$ normalized to $^{13}C_6$ glucose, is also reduced when all three genotypes are administered metformin (right half). The differences between between mutant and controls are flattened. **c-d.** In contrast, Labelled Lactate relative to glucose (C), and relative to pyruvate (D) goes up in wild-type (Control) and βcatGOF when animals are administered Metformin. Hras$^{G12V}$ epidermis which already had higher

$^{13}C_3$- lactate/$^{13}C_3$- pyruvate is not significantly different between Metformin-treated and untreated animals.; **e.** $^{13}C_2$ labelled Glutamate to Acetyl CoA is a measure of the dilution of the label due to glutamine entry into the TCA cycle. There is a modest dilution (upto 85%) in wild-type (Control) and βcatGOF animals' indicative of glutamine anaplerosis in animals treated with metformin when compared to animals without metformin (left– also in Fig. 5f). There is no significant change in Hras$^{G12V}$ animals treated with metformin. **f.** While there are no changes in labelled pyruvate relative to glucose in wild-type (Control) and Hras$^{G12V}$ animals upon metformin treatment, there is a small increase in βcatGOF animals upon metformin treatment. Each point represents epidermis (ear) from one mouse. Statistics: One-Way ANOVA – Multiple comparisons shown between Control and Control +metformin; βcatGOF and βcatGOF+ metformin; Hras$^{G12V}$ and Hras$^{G12V}$ + metformin. n = 7 (Control + metformin), 3 (βcatGOF+ metformin) 6 (Hras$^{G12V}$ + metformin). All p values from A-F are displayed on top of each comparison (Control versus Metformin-treated for all genotypes). Statistics reported for Control vs βcatGOF and Hras$^{G12V}$ (left) in Fig. 5b–e and Extended Data Fig. 6c. All box plots show all data points with whiskers going from minimum to maximum with line at the median.

*K14CreER;β-catnflox(Ex3);R26-LSL-H2BmCherry*

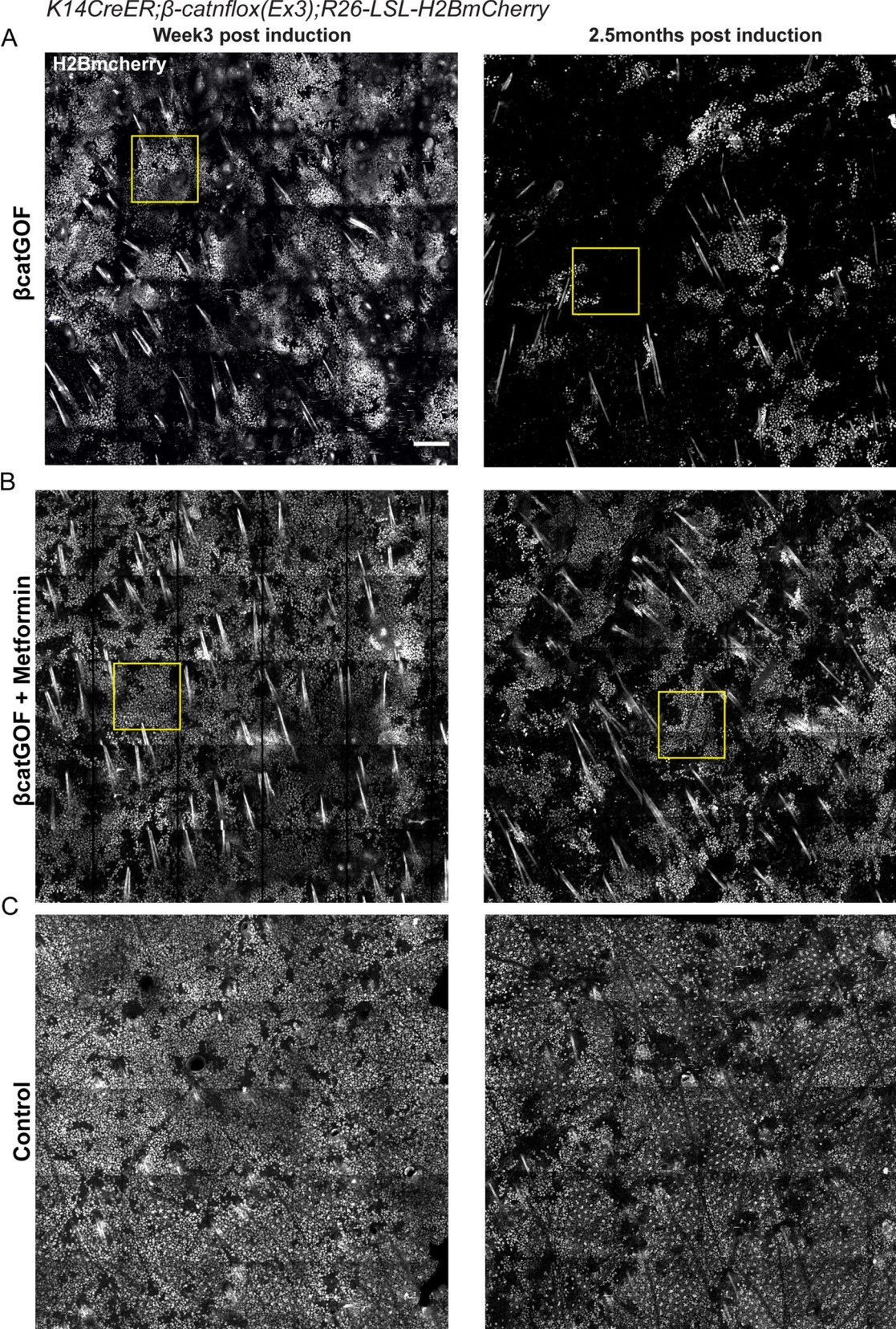

**Extended Data Fig. 9 | See next page for caption.**

**Extended Data Fig. 9 | Large regions of mutant βcatGOF epidermis shows inefficient elimination after administration of Metformin. a**. Revisits between Week 3 and 2.5 months show large regions (~1.4 mm) from βcatGOF mice (*K14CreER; βcatGOF; LSL-H2Bmcherry*). Whereas most of the epidermis was covered by recombined mutant H2Bmcherry-positive cells (white nuclei) at Week 3 post-induction, after 2.5 months, the epidermis is largely clear of the recombined mutant cells and they cover only ~15 % of the total area (quantified in Fig. 6h). **b** When βcatGOF mice are administered metformin and revisited, the recombined H2Bmcherry-positive mutant cells (white nuclei) still cover large areas of the epidermis at 2.5 months (~60 %), showing inefficient elimination.

Yellow insets in A and B are magnified in Fig. 6e and f. **c**. Wild-type epidermis does not show much change in percentage of area covered by recombined H2Bmcherry-positive cells over time once induced, retaining ~80% coverage (quantified in Extended Data Fig. 3a). Scale bar = 100 μm. Tattoos were used to identify the same large areas for revisits. While exact regions could be identified using hair follicle patterns for Wild-type as described before, approximate areas were identified using the manually applied tattoo for the mutant epidermis since the epidermal outgrowths and morphological aberrancy at Week 3 make exact identification of the same hair follicles difficult.

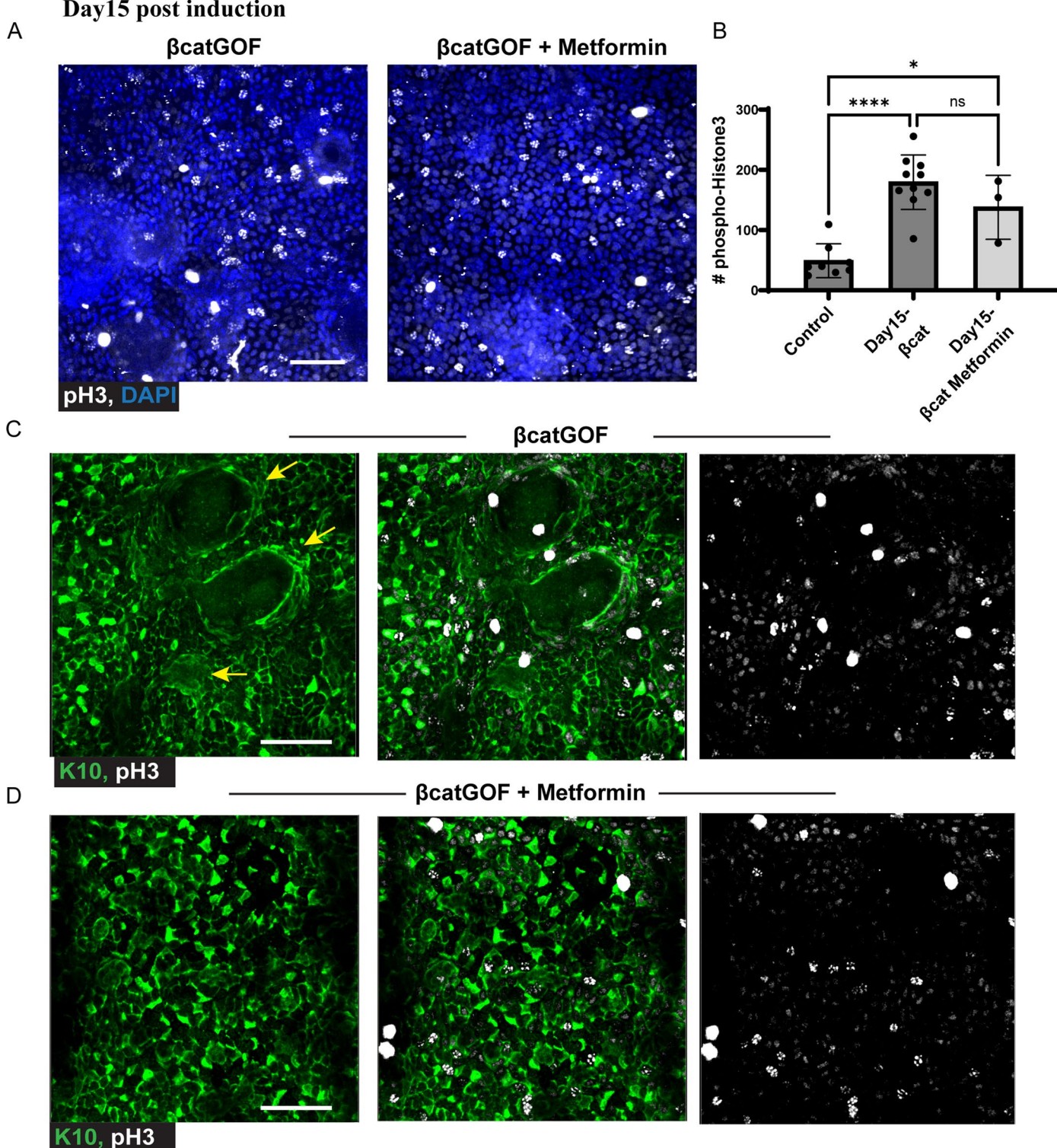

**Extended Data Fig. 10 | βcatGOF mutant administered metformin show alterations in patterns of Keratin 10 differentiation marker. a.** Phospho-histone H3 staining (white) from epidermal preps of βcatGOF mice (*K14CreER; βcatGOF; LSL-H2Bmcherry*) and βcatGOF mice administered Metformin. The metformin-treated mice still have a hyperproliferative epidermis at Day 15 when compared to mutant mice, quantified in graph **b**. There is a slight reduction in pH3 number (per 3-4 400 µm² regions averaged from multiple mice plotted as dots) although not statistically significant (ns) between metformin-treated and untreated βcatGOF mice. Both metformin-treated and untreated βcatGOF mice are hyperproliferative when compared to wild-type epidermis (control).

p value (One-Way ANOVA) ****<0.0001; *= 0.012. Please note that data plotted in Extended Data Fig. 2c has been replotted in B for comparison between earlier replicates of βcatGOF mice and Wild-type (control) mice with 3 mouse replicates each for βcatGOF and βcatGOF + Metformin added additionally. Data are presented as Mean ± SD. **c.** 3D projections (rendered in Imaris) of the basal layers of βcatGOF mutant epidermis at Day 15 with ectopic clusters of Keratin 10 differentiation marker arranged concentrically around outgrowths (yellow arrows) with pH3 around them. **d**. In contrast, 3D projections of the basal layer of βcatGOF mutant mice administered metformin shows more typical Keratin 10 staining like wild-type epidermis (Extended Data Fig. 7c). Scale bar = 50 µm.

RACHEL PERRY

# Reporting Summary

## Statistics

For all statistical analyses, confirm that the following items are present in the figure legend, table legend, main text, or Methods section.

| n/a | Confirmed | |
|---|---|---|
| ☐ | ☒ | The exact sample size (*n*) for each experimental group/condition, given as a discrete number and unit of measurement |
| ☐ | ☒ | A statement on whether measurements were taken from distinct samples or whether the same sample was measured repeatedly |
| ☐ | ☒ | The statistical test(s) used AND whether they are one- or two-sided<br>*Only common tests should be described solely by name; describe more complex techniques in the Methods section.* |
| ☒ | ☐ | A description of all covariates tested |
| ☒ | ☐ | A description of any assumptions or corrections, such as tests of normality and adjustment for multiple comparisons |
| ☐ | ☒ | A full description of the statistical parameters including central tendency (e.g. means) or other basic estimates (e.g. regression coefficient) AND variation (e.g. standard deviation) or associated estimates of uncertainty (e.g. confidence intervals) |
| ☐ | ☒ | For null hypothesis testing, the test statistic (e.g. *F*, *t*, *r*) with confidence intervals, effect sizes, degrees of freedom and *P* value noted<br>*Give P values as exact values whenever suitable.* |
| ☒ | ☐ | For Bayesian analysis, information on the choice of priors and Markov chain Monte Carlo settings |
| ☒ | ☐ | For hierarchical and complex designs, identification of the appropriate level for tests and full reporting of outcomes |
| ☒ | ☐ | Estimates of effect sizes (e.g. Cohen's *d*, Pearson's *r*), indicating how they were calculated |

*Our web collection on statistics for biologists contains articles on many of the points above.*

## Software and code

Policy information about availability of computer code

| Data collection | Imspector PRO, Agilent MSD Chemstation software, Sciex TripleTOF 6600 |
|---|---|
| Data analysis | Image J, MATLAB R2021a, IMARISx64 9.7.2, GraphPad Prism10.3.1, Microsoft Excel Version 2005, MetaboAnalyst 5.0, EL MAVEN(Elucidata.io) |

For manuscripts utilizing custom algorithms or software that are central to the research but not yet described in published literature, software must be made available to editors and reviewers. We strongly encourage code deposition in a community repository (e.g. GitHub). See the Nature Portfolio guidelines for submitting code & software for further information.

## Data

Policy information about availability of data

All manuscripts must include a data availability statement. This statement should provide the following information, where applicable:

- Accession codes, unique identifiers, or web links for publicly available datasets
- A description of any restrictions on data availability
- For clinical datasets or third party data, please ensure that the statement adheres to our policy

Data Availability statement has been added. " Source data are provided with this study. All other data supporting the findings of this study are available from the corresponding author on reasonable request."

# Field-specific reporting

Please select the one below that is the best fit for your research. If you are not sure, read the appropriate sections before making your selection.

☒ Life sciences          ☐ Behavioural & social sciences          ☐ Ecological, evolutionary & environmental sciences

For a reference copy of the document with all sections, see nature.com/documents/nr-reporting-summary-flat.pdf

# Life sciences study design

All studies must disclose on these points even when the disclosure is negative.

| | |
|---|---|
| Sample size | Three mice were used in general for imaging studies. Exact numbers are detailed in figure legends. Around 5-8 mice were used for mass spec studies- exact n is shown in the graph itself. Each point is plotted from a single animal ( described in legends). The sample size was chosen according to previously published papers. For non-imaging experiments also, a minimum of 3 mice was used , again based on previously publications.  A sentence has been added to the methods section with references for both imaging and mass spec studies which used similar sample sizes: " No statistical methods were used to pre-determine sample sizes but our sample sizes are similar to those reported in previous publications 56,65,70–74" |
| Data exclusions | No data was excluded to analyse.  Image quantification is described in methods. "No animals were excluded except when measurements could not be completed on a mouse due to variations in laser power or other variations in imaging conditions during a revisit. "- in Methods |
| Replication | For all imaging experiments, three separate mice imaged at different times (independent biological replicates) are shown. For all non- imaging experiments , a minimum of 3 mice and as many as 7-10 mice ( independent biological replicates) were used for mass spectrometry studies in the paper and reported in legends of each experiment with exact n. All attempts at replication were successful and plotted in the manuscript. |
| Randomization | Mice were first genotyped and identified as control or experimental animals. Within each experimental and control group, animals   were chosen randomly for all studies. |
| Blinding | No blinding was done since  the animals are readily identifiable by their toe biopsies and expression of reporter genes during experimentation and analysis. The animals under study need to be clearly identified by their genotype using toe biopsies before data collection. |

# Reporting for specific materials, systems and methods

We require information from authors about some types of materials, experimental systems and methods used in many studies. Here, indicate whether each material, system or method listed is relevant to your study. If you are not sure if a list item applies to your research, read the appropriate section before selecting a response.

## Materials & experimental systems

| n/a | Involved in the study |
|---|---|
| ☐ | ☒ Antibodies |
| ☐ | ☒ Eukaryotic cell lines |
| ☒ | ☐ Palaeontology and archaeology |
| ☐ | ☒ Animals and other organisms |
| ☒ | ☐ Human research participants |
| ☒ | ☐ Clinical data |
| ☒ | ☐ Dual use research of concern |

## Methods

| n/a | Involved in the study |
|---|---|
| ☒ | ☐ ChIP-seq |
| ☒ | ☐ Flow cytometry |
| ☒ | ☐ MRI-based neuroimaging |

## Antibodies

| | |
|---|---|
| Antibodies used | mouse anti-β-catenin (1:100; BD 610153), rabbit anti-phosphohistoneH3 (Ser10;1:300; Millipore 06-570), donkey anti-mouse AF488 (1:300; ThermoFisher A-21202), goat anti-rabbit AF633 (1:300; ThermoFisher A-21071). |
| Validation | mouse anti-β-catenin- BD Biosciences has validated specific binding (see technical date sheet of BD 610153). Cited previously in doi: 10.1126/science.1248373.<br>Rabbit anti- phosphohistone H3: Specific binding shown in technical data sheet for Millipore 06-570 anti-phospho histone H3. Cited previously in doi: 10.1038/s41556-021-00670-5 |

## Eukaryotic cell lines

Policy information about cell lines

| | |
|---|---|
| Cell line source(s) | 293T cells were purchased from Thermofisher. |

| Authentication | The 293T cell line, originally referred as 293tsA1609neo, is a highly transfectable derivative of human embryonic kidney 293 cells, and contains the SV40 T-antigen. It has been authenticated using STR profiling, morphology and expression of SV40 T-antigen(PCR) by ATCC as described in the product page of ATCC and cited widely . Examples: PubMed: 3031469, PubMed: 7690960. |
| --- | --- |
| Mycoplasma contamination | No mycoplasma contamination was observed. |
| Commonly misidentified lines (See ICLAC register) | No commonly misidentified cell line was used in this study. |

# Animals and other organisms

Policy information about studies involving animals; ARRIVE guidelines recommended for reporting animal research

| Laboratory animals | Mouse : Strains used: K14-CreER , Rosa26-CAG-LSL-H2B-mCherry ( Jackson Laboratory) R26p-Fucci2 mice were obtained from S. Aizawa (RIKEN), β-cateninflox(Ex3) mice were obtained from M.M. Taketo (Kyoto University)  and  HrasG12V  mice were obtained from S. Beronja (Fred Hutch). Mice of age post P21  were used : For most of the imaging and mass spec studies P45-P65 mice were used. For long term tracking mice of age 3 -4 months were used. Mice of both sex were randomly slected. All animal procedures were approved by the Yale University Institutional Animal Care and Use Committee and housed and fed according to approved protocol 11303. The water and chow (2018SC Rodent diet from Inotiv, US) was autoclaved before use. The housing used is consistent with the Guide for the Care and Use of Laboratory Animals and compliant with the Animal Welfare Act and Regulations.  Mice were housed on ventilated Tecniplast lixit racks with ambient temperature of 22°C and 50% ± 10% humidity with a 12h:12h light:dark cycle (light on 07:00am -7:00pm) and fed adlib. There are no tumors expected to appear in the experimental time window. |
| --- | --- |
| Wild animals | No wild animals were used in this study |
| Field-collected samples | No field collected samples were used in this study. |
| Ethics oversight | This study complies with all relevant ethical regulations and all animal procedures were approved by the Yale University Institutional Animal Care and Use Committee under the protocol number 11303 and 20290. |

Note that full information on the approval of the study protocol must also be provided in the manuscript.

