## [Peer Review File · Nature Cell Biology]

Metabolic rewiring in skin epidermis drives tolerance to oncogenic mutations

Corresponding Author: Professor Valentina Greco

This manuscript has been previously reviewed at another journal. This document only contains information relating to versions considered at Nature Cell Biology.

Version 0:

Decision Letter:

Dear Valentina,

Thank you very much for allowing Nature Cell Biology the opportunity to consider your manuscript, "Metabolic rewiring in skin epidermis drives tolerance to oncogenic mutations through two distinct modes of cell competition.", for publication. Having discussed the manuscript and the Nature referees' comments in detail with my colleagues, we have regretfully concluded that we cannot offer to publish it in Nature Cell Biology.

Nature and Nature Cell Biology are editorially independent. However, in cases such as this, when authors transfer a manuscript to Nature Cell Biology after it has been reviewed for Nature, Nature passes the referees' comments (and their identities) to us, with the author's permission.

Although this work is within the scope of this journal, the criticisms raised by the referees at Nature in the final round of review are serious enough to preclude publication in Nature Cell Biology.

Although we cannot publish your paper, it may be appropriate for another journal in the Nature Portfolio. If you wish to explore the journals and transfer your manuscript please use our manuscript transfer portal. You will not have to re-supply manuscript metadata and files, unless you wish to make modifications. For more information, please see our [manuscript transfer FAQ](http://www.nature.com/authors/author_resources/transfer_manuscripts.html?WT.mc_id=EMI_NPG_1511_AUTHORTRANSF&WT.ec_id=AUTHOR) page.

We are very sorry that our response could not be more positive on this occasion, but we thank you for the opportunity to consider this work. Please, let me know if you would like me to consult with our colleagues at Nature Communications and EMBO to see if they would be interested in taking the manuscript forward.

Best regards,
Stelios

Stylianos Lefkopoulos, PhD
He/him/his
Senior Editor, Nature Cell Biology
Springer Nature
Heidelberger Platz 3, 14197 Berlin, Germany

E-mail: stylianos.lefkopoulos@springernature.com
Twitter: [@s_lefkopoulos](https://twitter.com/s_lefkopoulos)
LinkedIn: [linkedin.com/in/stylianos-lefkopoulos-81b007a0](https://www.linkedin.com/in/stylianos-lefkopoulos-81b007a0)

** For Nature Portfolio general information and news for authors, see <http://npg.nature.com/authors>.

Version 1:

Decision Letter:

*Please delete the link to your author homepage if you wish to forward this email to co-authors.

Dear Valentina,

We apologize again for the delay in getting back to you with a decision. The original reviewer #3 had already signed off from the previous round of review, so we did not need to go back to them. On the other hand, the original reviewer #2 was unable to re-review, so we had to ask reviewer #1 to cross-comment on how they believe your revised manuscript addresses the issues that were raised by reviewer #2 in the previous round of review. Your manuscript, "Metabolic rewiring in skin epidermis drives tolerance to oncogenic mutations through two distinct modes of cell competition.", has therefore been seen by two of the original referees, specifically referee #1 and referee #4. As you will see from their comments (attached below) they find this work of interest, but have raised some important points (especially referee #4). Although we are also very interested in this study, we believe that their concerns should be addressed before we can consider publication in Nature Cell Biology.

We therefore invite you to address all the remaining reviewer concerns. Apart from addressing the remaining reviewer comments, please also pay close attention to our guidelines on statistical and methodological reporting (listed below) as failure to do so may delay the reconsideration of the revised manuscript. In particular please provide:

- a Supplementary Figure including unprocessed images of all gels/blots in the form of a multi-page pdf file. Please ensure that blots/gels are labeled and the sections presented in the figures are clearly indicated.
- a Supplementary Table including all numerical source data in Excel format, with data for different figures provided as different sheets within a single Excel file. The file should include source data giving rise to graphical representations and statistical descriptions in the paper and for all instances where the figures present representative experiments of multiple independent repeats, the source data of all repeats should be provided.

We therefore invite you to take these points into account when revising the manuscript. In addition, when preparing the revision please:

- ensure that it conforms to our format instructions and publication policies (see below and www.nature.com/nature/authors/).
- provide a point-by-point rebuttal to the full referee reports verbatim, as provided at the end of this letter.
- provide the completed Editorial Policy Checklist (found here <https://www.nature.com/authors/policies/Policy.pdf>), and Reporting Summary (found here <https://www.nature.com/authors/policies/ReportingSummary.pdf>). This is essential for reconsideration of the manuscript and these documents will be available to editors and referees in the event of peer review. For more information see <http://www.nature.com/authors/policies/availability.html> or contact me.

Nature Cell Biology is committed to improving transparency in authorship. As part of our efforts in this direction, we are now requesting that all authors identified as 'corresponding author' on published papers create and link their Open Researcher and Contributor Identifier (ORCID) with their account on the Manuscript Tracking System (MTS), prior to acceptance. ORCID helps the scientific community achieve unambiguous attribution of all scholarly contributions. You can create and link your ORCID from the home page of the MTS by clicking on 'Modify my Springer Nature account'. For more information please visit <http://www.springernature.com/orcid>.

Link Redacted

We would like to receive the revision within four weeks. If submitted within this time period, reconsideration of the revised manuscript will not be affected by related studies published elsewhere, or accepted for publication in Nature Cell Biology in the meantime. We would be happy to consider a revision even after this timeframe, but in that case we will consider the published literature at the time of resubmission when assessing the file.

We hope that you will find our referees' comments, and editorial guidance helpful. Please do not hesitate to contact me if there is anything you would like to discuss.

Best wishes,

Stelios

Stylianos Lefkopoulos, PhD
He/him/his
Senior Editor, Nature Cell Biology
Springer Nature
Heidelberger Platz 3, 14197 Berlin, Germany

E-mail: stylianos.lefkopoulos@springernature.com
Twitter: @s_lefkopoulos

Reviewers' Comments:

Reviewer #1:

Remarks to the Author:

The authors have improved and clarified the manuscript. A few minor/discretionary comments are below for the authors to consider:

1. Regarding comment 1 in the last critique, I do think it would be helpful for the authors to include an illustration similar to the one they provided in the rebuttal. This should make it helpful for readers to follow.
2. Regarding comment 4, I am fine with the authors using [2,3-13C]malate as a marker of TCA cycle turning beyond SDH. But please be sure to clarify the assumptions in the text. The figures provided in the rebuttal seem to suggest that all m+2 malate is in the form of [2,3-13C]malate (the correlation is perfect and the slope is 1.0). I assume I am missing something, but this seems impossible given the level of [4,5-13C]glutamate. The authors say that "2,3-13C malate" represents an "integrated average" of 1,2-13C malate and 3,4-13C-malate. Please spell out what this means, because it is still unclear to this reviewer.

I was further asked to comment on how the authors addressed the remaining points by reviewer #2. Regarding the first comment by reviewer #2, I feel that the conclusions that can be drawn are substantive enough for publication. I also agree with the authors that using the models suggested by this reviewer would eliminate the utility of their metabolite imaging, which was the entire basis of the paper. I feel the paper can move forward despite the fact that this comment was not completely addressed. The second comment by reviewer #2 had to do with the metformin experiments. Indeed, this is a complicated experiment to interpret (I had a similar comment about metformin). The authors performed new isotope labeling experiments to validate that metformin induces the expected perturbations in the skin, and they are now doing a better job of acknowledging the limitations of the experiment. I think this comment has been addressed.

Reviewer #4:

Remarks to the Author:

I would like to re-iterate my strong appreciation for the elegant, novel, and difficult experiments that are presented in this manuscript. However, I continue to have significant concerns related to the analysis methods and the interpretation of the metformin results.

A. The authors indicate that they do not calibrate for variations in incident laser power that may occur over different experimental days. This is a significant limitation as the level of two photon excited fluorescence depends on the square of the incident power. While not performing such a calibration does not impact comparison of redox ratios from different cells present in the same field, such calibrations are critical for comparing data acquired between different days. I hope the authors noted illumination powers in their experimental notes, or have meta data files which at least report on the laser power emitted by the laser for a given experiment for each excitation wavelength. Otherwise, as the authors note themselves, small variations can have a significant impact. In fact, this is highlighted in the figure provided by the authors in the response document, highlighting redox ratio variations for different mice. Given how tightly redox ratios are controlled within cells, the big variations seen in the ratios for mouse 1 vs mouse 2 and 3 are likely due to small changes in the relative incident powers of the two wavelengths used to acquire NAD(P)H and FAD images. These variations are much more significant than the differences seen between Day 0 and Day 5, so it is not impossible that subtle differences between Day 0 and Day 5 and then again on Day 10. This could really impact the interpretation of the optical redox ratio data.

B. There are a number of manuscripts reporting on endogenous two-photon excited fluorescence images acquired from tissues in vivo and ex vivo with similar resolution that employ traditional and deep learning methods to isolate cytoplasmic regions and/or segment nuclear regions (e.g. PMID: 34797690, 33741692, 32577625, 35653192). If the quality of the images included in the manuscript is representative of the SNR of the images acquired, it should be feasible to segment nuclei. The results reported in Fig. 2 and 3 of the response are interesting. In the reviewers opinion they do highlight a higher nucleus/cell area for the BcatGOF cells vs WT cells. At the same time they do also seem to indicate that the redox ratio of cells with nuclear/cell areas that are different by almost a factor of two are similar. If indeed FAD levels are similar to those of NAD(P)H, then the redox defined as NAD(P)H/FAD of the nucleus should in principle be infinitely high, because there is no FAD in the cells. Admittedly, NADH signals from the nucleus are significantly lower than the cytoplasm and the images are not perfectly noise free, so this likely yields the observed hues. Nevertheless if let's say the min to max values of the color bars represent a range of even 0.5 to 1.2, and the area occupied by the nucleus doubles, if the redox state of the cytoplasm remains the same, the redox for a given cell which includes both cytoplasm and nucleus should decrease significantly. If it doesn't, that implies that somehow the cytoplasmic redox of the cell with the bigger nucleus is a lot larger. As noted in my previous review, especially if the authors would like to compare differences between different days, the normalized redox ratio should provide a more robust metric, assuming of course that intensity variations are properly calibrated.

C. Also as noted in my previous review, comparisons of the un-normalized redox ratios highlighted in Fig. S3 and Fig. S5, indicate that the fact that WT and Hras neighbors are similar is more likely driven by changes in the WT cells than the Hras cells. This would not be consistent with the conclusion "However, the winner cells in each model ... rapidly recover their redox ratios, irrespective of the mutation induced." The Hras+ cells are not recovering their redox based on what is shown in Fig. S5, but they are somehow making their neighbors reduce their redox. Of course, and as noted in point A comparisons of data between days is not robust if variations in incident intensity are not calibrated for. The conclusion that in the BcatGOF cases differences remain than in the Hras model they are abrogated at 10 days remains valid. But how these differences are abrogated (i.e. by the "winner" cells in the Hras case recovering their redox ratios) cannot be asserted.

D. Concerns regarding the relative magnitude of the impact of metformin alone vs the impact of metformin in the Hras and BcatGOF model remain. The additional mass spec data along with data on Fig. S9 highlight that the metabolic impacts of metformin on control cells are significant, leading to significantly higher variations in the redox values. Thus, from a statistical perspective, a lot more measurements would be required to be able to detect significant differences for a similar size effect (i.e. the differences induced by Hras and BcatGOF vs. WT on day 5).

E. The actual colorbar values should be indicated, instead of min and max.

F. Not clear why in Fig. 1Ci, the corresponding redox ratio images are not also shown and then plotted quantitatively in Fig. 1Ciii.

G. There are issues with the formatting of the supplementary figure legends and the legend of Fig. S1 doesn't seem to correspond to the panels as labeled in the figure.

H. References to previous tissue optical redox imaging studies are limited.

GUIDELINES FOR SUBMISSION OF NATURE CELL BIOLOGY ARTICLES

ARTICLE FORMAT

ABSTRACT – should not exceed 150 words and should be unreferenced. This paragraph is the most visible part of the paper and should briefly outline the background and rationale for the work, and accurately summarize the main results and conclusions. Key genes, proteins and organisms should be specified to ensure discoverability of the paper in online searches.

TEXT – the main text consists of the Introduction, Results, and Discussion sections and must not exceed 3500 words including the abstract. The Introduction should expand on the background relating to the work. The Results should be divided in subsections with subheadings, and should provide a concise and accurate description of the experimental findings. The Discussion should expand on the findings and their implications. All relevant primary literature should be cited, in particular when discussing the background and specific findings.

REFERENCES – are limited to a total of 70 in the main text and Methods combined,. They must be numbered sequentially as they appear in the main text, tables and figure legends and Methods and must follow the precise style of Nature Cell Biology references. References only cited in the Methods should be numbered consecutively following the last reference cited in the main text. References only associated with Supplementary Information (e.g. in supplementary legends) do not count toward the total reference limit and do not need to be cited in numerical continuity with references in the main text. Only published papers can be cited, and each publication cited should be included in the numbered reference list, which should include the manuscript titles. Footnotes are not permitted.

Methods should be written concisely, but should contain all elements necessary to allow interpretation and replication of the results. As a guideline, Methods sections typically do not exceed 3,000 words. The Methods should be divided into subsections listing reagents and techniques. When citing previous methods, accurate references should be provided and any alterations should be noted. Information must be provided about: antibody dilutions, company names, catalogue numbers and clone numbers for monoclonal antibodies; sequences of RNAi and cDNA probes/primers or company names and catalogue numbers if reagents are commercial; cell line names, sources and information on cell line identity and authentication. Animal studies and experiments involving human subjects must be reported in detail, identifying the committees approving the protocols. For studies involving human subjects/samples, a statement must be included confirming that informed consent was obtained. Statistical analyses and information on the reproducibility of experimental results should be provided in a section titled "Statistics and Reproducibility".

All Nature Cell Biology manuscripts submitted on or after March 21 2016, must include a Data availability statement as a separate section after Methods but before references, under the heading "Data Availability". For Springer Nature policies on data availability see <http://www.nature.com/authors/policies/availability.html>; for more information on this particular policy see <http://www.nature.com/authors/policies/data/data-availability-statements-data-citations.pdf>. The Data availability statement should

include:

- Accession codes for primary datasets (generated during the study under consideration and designated as "primary accessions") and secondary datasets (published datasets reanalysed during the study under consideration, designated as "referenced accessions"). For primary accessions data should be made public to coincide with publication of the manuscript. A list of data types for which submission to community-endorsed public repositories is mandated (including sequence, structure, microarray, deep sequencing data) can be found here <http://www.nature.com/authors/policies/availability.html#data>.
- Unique identifiers (accession codes, DOIs or other unique persistent identifier) and hyperlinks for datasets deposited in an approved repository, but for which data deposition is not mandated (see here for details <http://www.nature.com/sdata/data-policies/repositories>).
- At a minimum, please include a statement confirming that all relevant data are available from the authors, and/or are included with the manuscript (e.g. as source data or supplementary information), listing which data are included (e.g. by figure panels and data types) and mentioning any restrictions on availability.
- If a dataset has a Digital Object Identifier (DOI) as its unique identifier, we strongly encourage including this in the Reference list and citing the dataset in the Methods.

We recommend that you upload the step-by-step protocols used in this manuscript to [protocols.io](https://www.protocols.io). More details can found at <https://www.protocols.io/help/publish-articles>.

DISPLAY ITEMS – main display items are limited to 6-8 main figures and/or main tables. For Supplementary Information see below.

FIGURES – Colour figure publication costs \$395 per colour figure. All panels of a multi-panel figure must be logically connected and arranged as they would appear in the final version. Unnecessary figures and figure panels should be avoided (e.g. data presented in small tables could be stated briefly in the text instead).

All imaging data should be accompanied by scale bars, which should be defined in the legend.

Cropped images of gels/blots are acceptable, but need to be accompanied by size markers, and to retain visible background signal within the linear range (i.e. should not be saturated). The boundaries of panels with low background have to be demarked with black lines. Splicing of panels should only be considered if unavoidable, and must be clearly marked on the figure, and noted in the legend with a statement on whether the samples were obtained and processed simultaneously. Quantitative comparisons between samples on different gels/blots are discouraged; if this is unavoidable, it has to be performed for samples derived from the same experiment with gels/blots were processed in parallel, which needs to be stated in the legend.

Regardless of format, all figures must be vector graphic compatible files, not supplied in a flattened raster/bitmap graphics format, but should be fully editable, allowing us to highlight/copy/paste all text and move individual parts of the figures (i.e. arrows, lines, x and y axes, graphs, tick marks, scale bars etc). The only parts of the figure that should be in pixel raster/bitmap format are photographic images or 3D rendered graphics/complex technical illustrations.

Unprocessed scans of all key data generated through electrophoretic separation techniques need to be presented in a supplementary figure that should be labeled and numbered as the final supplementary figure, and should be mentioned in every relevant figure legend. This figure does not count towards the total number of figures and is the only figure that can be displayed over multiple pages, but should be provided as a single file, in PDF or TIFF format. Data in this figure can be displayed in a relatively informal style, but size markers and the figures panels corresponding to the presented data must be indicated.

The total number of Supplementary Figures (not including the "unprocessed scans" Supplementary Figure) should not exceed the number of main display items (figures and/or tables (see our Guide to Authors and March 2012 editorial <http://www.nature.com/ncb/authors/submit/index.html#suppinfo>; <http://www.nature.com/ncb/journal/v14/n3/index.html#ed>). No restrictions apply to Supplementary Tables or Videos, but we advise authors to be selective in including supplemental data.

GUIDELINES FOR EXPERIMENTAL AND STATISTICAL REPORTING

REPORTING REQUIREMENTS – To improve the quality of methods and statistics reporting in our papers we have recently revised the reporting checklist we introduced in 2013. We are now asking all life sciences authors to complete two items: an Editorial Policy Checklist (found here <https://www.nature.com/authors/policies/Policy.pdf>) that verifies compliance with all required editorial policies and a Reporting Summary (found here <https://www.nature.com/authors/policies/ReportingSummary.pdf>) that collects information on experimental design and reagents. These documents are available to referees to aid the evaluation of the manuscript. Please note that these forms are dynamic 'smart pdfs' and must therefore be downloaded and completed in Adobe Reader. We will then flatten them for ease of use by the reviewers. If you would like to reference the guidance text as you complete the template, please access these flattened versions at <http://www.nature.com/authors/policies/availability.html>.

Version 2:

Decision Letter:

3rd September 2024

Dear Valentina,

Thank you for submitting your revised manuscript "Metabolic rewiring in skin epidermis drives tolerance to oncogenic mutations through two distinct modes of cell competition." (NCB-A53518B). We sincerely apologize for the delay in getting back to you, but after we received the report by reviewer #4, we wanted to have additional input into the points they continued to raise from another expert; we did this in our effort to have as balanced and constructive of a peer review process as possible. Therefore, your manuscript has now been seen by the original referee #4 and an arbitrator (referee #5) who was asked to focus on commenting on the remaining points by referee #4 and not raise new points. All the referee comments are below. Taking all the expert advice into account, we have decided that we'll be happy in principle to publish your manuscript in Nature Cell Biology, pending minor revisions to satisfy referee #5's requests and to comply with our editorial and formatting guidelines.

If the current version of your manuscript is in a PDF format, please email us a copy of the file in an editable format (Microsoft Word or LaTeX)-- we cannot proceed with PDFs at this stage.

We are now performing detailed checks on your paper and will send you a checklist detailing our editorial and formatting requirements in about 2 weeks. Please do not upload the final materials and make any revisions until you receive this additional information from us.

Thank you again for your interest in Nature Cell Biology. Please do not hesitate to contact me if you have any questions.

Best regards,
Stelios

Stylianos Lefkopoulos, PhD
He/him/his
Senior Editor, Nature Cell Biology
Springer Nature
Heidelberger Platz 3, 14197 Berlin, Germany

E-mail: stylianos.lefkopoulos@springernature.com
Twitter: @s_lefkopoulos
LinkedIn: [linkedin.com/in/stylianos-lefkopoulos-81b007a0](https://www.linkedin.com/in/stylianos-lefkopoulos-81b007a0)

Reviewer #4 (Remarks to the Author):

Remaining concerns with revision:

It is appreciated that some of the wording of the manuscript has been modified to address concerns raised by the reviewer. However, significant concerns remain.

Robust power calibrations

The differences between the mutant and WT cells that are presented in this manuscript appear to be on the order of 5%. If on Day 5, data from the same mouse are taken with a 1.1% increase at 750 and a 1.1% decrease at 890 nm relative to Day 0, this would result in a 5% increase in the NADH/FAD signal, even if the redox ratio didn't really change. In the Methods the authors state (lines 593-594):

"Laser power at samples was measured periodically and maintained between 25-35 mW." First, it is not clear at whether this statement refers to all wavelengths. Secondly, the reported range varies by 30%. So, this is certainly not consistent with the added statement (lines 608-609):

"While revisiting a mouse, imaging conditions were kept constant for NAD(P)H and FAD between Day0, Day4-6 and Day 9-13 shown in the manuscript."

A variation of 30% (10 mW variation with a mean of 30 mW) in incident power is not constant.

Also, the authors mention that they use two lasers for their measurements. Information regarding what laser is used for what wavelengths, and what is the order of image acquisition. Also, the full depth over which images were acquired and the image acquisition times are not provided. Laser emission can vary with changes in temperature and humidity in a room, so even if the laser settings are the same, the actual power delivered to a specimen can vary. For robust measurements, which rely on multi-day comparisons, it is important to have such fluctuations accounted for.

It is certainly encouraging that variations in litter mate controls do not show significant variations over the 10-15 days of imaging. Still, the days for the littermate control measurements appear very different from the data for the BcatGOF mice. However, as indicated above laser fluctuations can occur over the course of even a multi-hour experiment, so this is not necessarily proof that such fluctuations did not contribute to the observed changes, which limits the rigor of the results and conclusions.

Lack of nuclear segmentation

In the previous review, this reviewer made a fairly detailed argument regarding the importance of segmenting the cytoplasmic regions for making robust quantitative comparisons based on which important functional conclusions are drawn. In the figure submitted by the authors in the previous response:

The nucleus to cytoplasm ratio for BcatGOF#1 mouse varies from about 0.3 to about 0.6. If we start with a case where the nucleus to cytoplasm ratio is 0.3 and NAD(P)H/FAD is 1.1. Assuming a background redox of 1.0 from the nucleus (consistent with some of the scale bars shown in Figures), that means that the redox of the cytoplasm occupying 60% of the cell is about 1.14. Now, if we consider a cell for which the nucleus to cytoplasm ratio is 0.6, and the background (nucleus) redox is the same (i.e. 1.0), then the redox of the 40% cytoplasmic area should be 1.25 to yield the same cell redox of 1.1. This is a 15% difference, 3 times higher than the range of redox ratio differences reported for most of the comparisons made. This means that even though these cells are actually significantly metabolically different, the analysis would identify them as metabolically very similar. Similar arguments can be made for cells that are metabolically similar, but identified as different because their nuclear to cytoplasmic ratio is different. This is very relevant for all the comparisons made, including the metformin ones.

The authors do have the mCherry signal, which in principle should be easy to segment automatically via standard thresholding approaches. The nuclei of the WT mutant cells may be more challenging and in the worst case scenario may require manual

segmentation. It appears that the authors have already the segmented cell outlines, since they include cell-based measurements in their results (lines 621-622). At any rate, if the authors would like to draw important functional conclusions from their elegant experiments, then it is the opinion of this reviewer at least, that these more careful assessments need to be undertaken.

Data normalization presentation

It is great to see that the authors have modified their previous interpretation of the data from "However, the winner cells in each model ... rapidly recover their redox ratios, irrespective of the mutation induced." to indicate that the differences are now flattened. However, the manner in which the data is presented in the main figure of the paper still presents the data in a manner which suggests recovery of the redox ratio for the Hras group. This remains misleading and in the opinion of this reviewer, data in the format presented in supplemental figures 2 and 4 should replace the format of the data shown in figures 3 and 4.

Metformin result interpretation

Given the magnitude of the effects of metformin, the limitations with the proper calibration and nuclear exclusion from redox estimates, the results remain unconvincing. Inclusion of experiments from more mice would make results more solid. Given the impact of metformin on variation and the size of the effect the authors observe in the non-metformin treatment groups, it should be possible to calculate the number of independent samples (i.e. mice) that need to be sampled. If such analysis indicates that 3 mice suffice, and the images are analyzed in a more robust manner, then the lack of significance reported would certainly be more robust.

Reviewer #5 (Remarks to the Author):

In this work from Greco et al., both in vivo optical ratios and isotope tracing are used to discern changes in single cell redox ratios with respect to gain of function mutations in b-catenin and mutant Hras. The work encompasses both in vitro and in vivo experiments, highlighting the possibility of cell to cell communication as the rationale for differential changes in redox state. I would only offer minor suggestions with respect to the remaining concerns of Reviewer #4:

1. The authors show that tumors formed in vivo from BcatGOF and HrasG12V rely on differential metabolism. Clearly glucose uptake and metabolism drives the redox phenotype. I would encourage the authors to comment on glucose uptake between the 2 models and if what is being observed is dependent on the amount of glucose carbons that the cell takes up?
2. I find the author's response with respect to the system calibration adequate for this study. They are not measuring absolute values and they provide reasonable metrics for stability given such a challenging experiment.
3. I would however suggest they tone down their use of the word "winner" or eliminate altogether. I find it confusing and also unnecessary.

The authors spend a paragraph in the discussion (ln 523-537) discussing the evidence for cell-cell communication. Given that the authors don't actually prove cell-cell communication through experimentation, meaning that they do not do an experiment where they show one cell makes a metabolite and the other cell consumes it, I would tone this language down and consider also that the communication may also just be substrate competition.

Version 3:

Decision Letter:

Dear Valentina,

I am pleased to inform you that your manuscript, "Metabolic rewiring in skin epidermis drives tolerance to oncogenic mutations", has now been accepted for publication in Nature Cell Biology. Congratulations to the whole team!

Once your paper has been scheduled for online publication, the Nature press office will be in touch to confirm the details. An online order

form for reprints of your paper is available at <https://www.nature.com/reprints/author-reprints.html>. All co-authors, authors' institutions and authors' funding agencies can order reprints using the form appropriate to their geographical region.

Please note that *Nature Cell Biology* is a Transformative Journal (TJ). Authors may publish their research with us through the traditional subscription access route or make their paper immediately open access through payment of an article-processing charge (APC). Authors will not be required to make a final decision about access to their article until it has been accepted. [Find out more about Transformative Journals](https://www.springernature.com/gp/open-research/transformative-journals)

If you have not already done so, we strongly recommend that you upload the step-by-step protocols used in this manuscript to protocols.io (<https://protocols.io>), an open online resource that allows researchers to share their detailed experimental know-how. All uploaded protocols are made freely available and are assigned DOIs for ease of citation. Protocols and Nature Portfolio journal papers in which they are used can be linked to one another, and this link is clearly and prominently visible in the online versions of both. Authors who performed the specific experiments can act as primary authors for the Protocol as they will be best placed to share the methodology details, but the Corresponding Author of the present research paper should be included as one of the authors. By uploading your Protocols onto protocols.io, you are enabling researchers to more readily reproduce or adapt the methodology you use, as well as increasing the visibility of your protocols and papers. You can also establish a dedicated workspace to collect your lab Protocols. Further information can be found at <https://www.protocols.io/help/publish-articles>.

Nature Cell Biology encourages authors presenting evidence for cell, biological, molecular, and genetic interactions to consider communicating these findings using Biofactoid (<https://biofactoid.org/>). This tool helps users share a searchable representation of interactions (e.g. binding, gene expression, post-translational modification) between genes, gene products, or chemicals. Information added to Biofactoid, with author attribution, is shared on social media and public databases, such as Pathway Commons, where it can be discovered and analyzed in the context of a large and growing corpus of knowledge.

With kind regards,
Stelios

Stylios Lefkopoulos, PhD
He/him/his
Senior Editor, Nature Cell Biology
Springer Nature
Heidelberger Platz 3, 14197 Berlin, Germany

E-mail: stylios.lefkopoulos@springernature.com
Twitter: [@s_lefkopoulos](https://twitter.com/s_lefkopoulos)
LinkedIn: [linkedin.com/in/stylios-lefkopoulos-81b007a0](https://www.linkedin.com/in/stylios-lefkopoulos-81b007a0)

** Visit the Springer Nature Editorial and Publishing website at http://editorial-jobs.springernature.com?utm_source=ejp_NCB_email&utm_medium=ejp_NCB_email&utm_campaign=ejp_NCB for more information about our career opportunities. If you have any questions please click [here](mailto:editorial.publishing.jobs@springernature.com).

[REDACTED]

Reviewers' Comments:

Reviewer #1:

Remarks to the Author:

The authors have improved and clarified the manuscript. A few minor/discretionary comments are below for the authors to consider:

1. Regarding comment 1 in the last critique, I do think it would be helpful for the authors to include an illustration similar to the one they provided in the rebuttal. This should make it helpful for readers to follow.

Thank you for this suggestion. We have now added the below illustration and images reporting on the DCA treatment *in vivo* to Extended Data 4 C-D.

2. Regarding comment 4, I am fine with the authors using [2,3-¹³C]malate as a marker of TCA cycle turning beyond SDH. But please be sure to clarify the assumptions in the text. The figures provided in the rebuttal seem to suggest that all m+2 malate is in the form of [2,3-¹³C]malate (the correlation is perfect and the slope is 1.0). I assume I am missing something, but this seems impossible given the level of [4,5-¹³C]glutamate. The authors say that “2,3-¹³C malate” represents an “integrated average” of 1,2-¹³C malate and 3,4-¹³C-malate. Please spell out what this means, because it is still unclear to this reviewer.

Our assertion is that 1,2-¹³C and 3,4-¹³C malate, both of which the field cannot measure with positional specificity, can be represented by 2,3-¹³C malate enrichment. We are pleased that the reviewer is fine with the use of 2,3-¹³C malate enrichment, and we thank the reviewer for the opportunity to clarify the enrichment data. The most important point of clarification is that the fractional enrichment in [2,3-¹³C] malate represents the fraction of the 2,3-¹³C fragment that is labeled m+2, not the fraction of total malate enrichment that is found in the 2,3 fragment. However, [4,5-¹³C] glutamate label will end up in [3,4-¹³C] and [1,2-¹³C] malate (we cannot distinguish between the two as malate is a symmetrical molecule as noted previously). So, the fact that the [4,5-¹³C] glutamate label is identical to the [2,3-¹³C] malate enrichment indicates that all four malate carbons must have very similar labeling patterns (i.e. each carbon must be labeled almost identically), as is expected with PDH flux predominating over PC flux (as indicated by other data presented in the previous response to this reviewer). Further: the fractional enrichment of C2C3 malate does not mean depletion of the label from other isotopomers

of malate: the presented enrichment simply refers to the ^{13}C (as opposed to ^{12}C) labeling of the relevant carbons. We have added the following clarification to the main text:

“In these studies, $2,3\text{-}^{13}\text{C}_2$ malate was considered an integrated average of $1,2\text{-}^{13}\text{C}_2$ and $3,4\text{-}^{13}\text{C}_2$ malate (both of which are unmeasurable by mass spectrometry), weighting the fraction of anaplerosis generating C_3C_4 malate as compared to C_1C_2 malate.”

I was further asked to comment on how the authors addressed the remaining points by reviewer #2. Regarding the first comment by reviewer #2, I feel that the conclusions that can be drawn are substantive enough for publication. I also agree with the authors that using the models suggested by this reviewer would eliminate the utility of their metabolite imaging, which was the entire basis of the paper. I feel the paper can move forward despite the fact that this comment was not completely addressed. The second comment by reviewer #2 had to do with the metformin experiments. Indeed, this is a complicated experiment to interpret (I had a similar comment about metformin). The authors performed new isotope labeling experiments to validate that metformin induces the expected perturbations in the skin, and they are now doing a better job of acknowledging the limitations of the experiment. I think this comment has been addressed.

We thank the reviewer for their investment in reading and responding to the comments from the Reviewer # 2 and recognizing our efforts to address their critique.

Reviewer #4:

Remarks to the Author:

I would like to re-iterate my strong appreciation for the elegant, novel, and difficult experiments that are presented in this manuscript. However, I continue to have significant concerns related to the analysis methods and the interpretation of the metformin results.

We appreciate the reviewer's appreciation for our work as well as their continued investment in improving the manuscript. As requested by this reviewer, we have further clarified the analysis methods and interpretation of metformin data as described below.

A. The authors indicate that they do not calibrate for variations in incident laser power that may occur over different experimental days. This is a significant limitation as the level of two photon excited fluorescence depends on the square of the incident power. While not performing such a calibration does not impact comparison of redox ratios from different cells present in the same field, such calibrations are critical for comparing data acquired between different days. I hope the authors noted illumination powers in their experimental notes, or have meta data files which at least report on the laser power emitted by the laser for a given experiment for each excitation wavelength. Otherwise, as the authors note themselves, small variations can have a significant impact. In fact, this is highlighted in the figure provided by the authors in the response document, highlighting redox ratio variations for different mice. Given how tightly redox ratios are controlled within cells, the big variations seen in the ratios for mouse 1 vs mouse 2 and 3 are likely due to small changes in the relative incident powers of the two wavelengths used to acquire NAD(P)H and FAD images. These variations are much more significant than the differences seen between Day 0 and Day 5, so it is not impossible that subtle differences between Day 0 and Day 5 and then again on Day 10. This could really impact the interpretation of the optical redox ratio data.

We would like to clarify that the aim of our experiments and representations are to quantify relative changes in redox ratios (not absolute redox ratios) after recombination and expression of oncogenic mutations between days while revisiting a single mouse. Laser power for NAD(P)H and FAD measurements and all other acquisition parameters were noted for each imaging session and are kept the same throughout these revisits between Day 0, Day 5 (4-6) and Day 10 (9-13) in *the same mouse* (as explained in Methods). Time series are

normalized to baseline average redox value at Day 0 to plot the extent of these differences in multiple mice, which are sometimes acquired within a gap of months in the same graph. Therefore, we are not comparing Day 0 redox ratios across 3 mice; rather, we are comparing the extent of redox change/drop, 5 days after recombination, from baseline (Day 0) in the same mice. These differences between Day 0 and Day 5 are the same in magnitude and direction in the 3 mice and, hence, statistically significant.

To control for possible minor variations in excitation conditions between Day 0, Day 5, Day 10 and even longer revisits (Day 15), we have shown that wild-type littermates do not show a drop in average redox ratios that we observe in the mutant models (Fig. 2C). These experiments followed the same protocol while keeping laser power and other imaging parameters constant between the different days of revisits in each mouse. Here too, different wild-type mice start at different baseline (Day 0) redox values due to mice born in different litters and imaged at different times. However, these wild-type mice do not show a significant drop in average redox ratio after recombination when compared to those baselines, unlike mutant mice which start expressing the mutations after Day 0. Hence, even if minor fluctuations in imaging conditions (that we may not detect in spite of keeping the imaging parameters the same between days of revisits in the same mouse) are present between Day 0, Day 5 and Day 10 in mice that are revisited, they do not lead to any statistically significant difference in average redox ratios over time in wild-type recombined controls. Thus, the NAD(P)H/FAD specifically changes from baseline values (Day 0) during revisits only upon expression of the mutations in these mice.

Normalizing the whole time series to the baseline intensities is a standard practice to compile multiple experiments from time lapse imaging data and, to the best of our understanding, completely reflects what we observe in the images.

Figure showing wild-type litter mates (top row; K14CreR: LSL-H2Bmcherry) and β catGOF (bottom row: K14CreER; LSL-H2B-mcherry; LSL- β catGOF) mice. Here raw values for NAD(P)H/ FAD are shown (in the same range of arbitrary intensity units -0.4) following revisits in each mouse while keeping the imaging conditions the same. They were normalized to Day0/ baseline average redox and combined to get the graphs in Fig. 2C and 2D.

B. There are a number of manuscripts reporting on endogenous two-photon excited fluorescence images acquired from tissues in vivo and ex vivo with similar resolution that employ traditional and deep learning methods to

isolate cytoplasmic regions and/or segment nuclear regions (e.g.PMID: 34797690, 33741692, 32577625, 35653192). If the quality of the images included in the manuscript is representative of the SNR of the images acquired, it should be feasible to segment nuclei. The results reported in Fig. 2 and 3 of the response are interesting. In the reviewers opinion they do highlight a higher nucleus/cell area for the BcatGOF cells vs WT cells. At the same time they do also seem to indicate that the redox ratio of cells with nuclear/cell areas that are different by almost a factor of two are similar. If indeed FAD levels are similar to those of NAD(P)H, then the redox defined as NAD(P)H/FAD of the nucleus should in principle be infinitely high, because there is no FAD in the cells. Admittedly, NADH signals from the nucleus are significantly lower than the cytoplasm and the images are not perfectly noise free, so this likely yields the observed hues. Nevertheless if let's say the min to max values of the color bars represent a range of even 0.5 to 1.2, and the area occupied by the nucleus doubles, if the redox state of the cytoplasm remains the same, the redox for a given cell which includes both cytoplasm and nucleus should decrease significantly. If it doesn't, that implies that somehow the cytoplasmic redox of the cell with the bigger nucleus is a lot larger. As noted in my previous review, especially if the authors would like to compare differences between different days, the normalized redox ratio should provide a more robust metric, assuming of course that intensity variations are properly calibrated.

There are 2 parts to this question. We have clarified the answer to the second question about comparing intensities between days in the same mouse while keeping imaging conditions the same in the previous answer to **point A** and clarified the controls used: wild-type animals taken through recombination of only the fluorescent reporters and revisited over time; and comparison between recombined and unrecombined neighboring cells from the same animals that show differences only upon the presence of mutant cells (Figure 2C-D).

To clarify the quantification including nuclear signal, the reason that there are no 0 or infinite pixel values in the images we quantify is because noise has not been removed from the NAD(P)H and FAD images before they are represented as a ratio. Hence, the lowest signals fall to the level of noise and no further. We have found that using raw images with no non-linear denoising is essential to be able to do any kind of ratiometric representation of intensities- be it NAD(P)H/ FAD or FAD/ NAD(P)H + FAD. This finding is consistent with the experience of our collaborator Dr. Melissa Skala, who has a longer history of studying and publishing redox ratios in different model systems. Hence there is no danger of catching irrational numbers while quantifying the redox ratio.

We thank the reviewer for pointing us to these elegant studies wherein researchers have used the NAD(P)H signal to segment out the nuclei. While one of them (34797690) is on isolated hepatocytes, others are papers where the sole focus is to quantify redox signal from complex 3D environments and, in the future, this will help us evolve our strategy. We face 3 challenges: first, isolate the basal stem cells from the 3dimensional- skin in vivo data that we described in methods, Fig. S1 and [Redacted]; second, segment the cells and third, identify mutant and wild-type so as to measure per cell-average redox. Isolating the basal stem cell layer from the other differentiated layers of the skin with different morphology and cell states is crucial to make relevant biological conclusions about these stem cells.

We did try to run the isolated basal cells through a software that allows segmentation of cells with a membrane and nuclear markers. In our system, these markers are obtained by inverting the NAD(P)H images and yet we found that there were several errors including multiple cells being lumped together and portions of the isolated basal planes being missed. This is likely because the NAD(P)H signal resembles mitochondria in pattern and does not label the cytosol uniformly. While we think neural network models, like the reviewer referred in PMID 33741692 can be trained to go through this data and segment single cells with nuclei excluded with greater accuracy, we respectfully believe that this would be beyond the scope of the paper. Currently, our manual quantifications are able to get per-cell redox values that are mutant (H2Bmcherry positive) or wild-type (H2Bmcherry negative) and show differences only observed once the mutation is induced.

How redox changes subcellularly with changes in cell morphology is indeed a question of great interest. We observe that cells that are dividing and have a different nuclear to cytoplasmic ratio than G1 cells do have small changes (Fig. 1C). However, the population range of average redox per cell is maintained through time (Fig. 2C) in the basal stem cell compartment. In contrast, mutant cells sitting next to wild-type cells have much wider differences in their redox values than can be attributed to cycling cells. How average redox value is maintained in a tight range within the basal stem cells is definitely a question we intend to pursue. It is likely that cell-cell communication of redox state (because of which mutant cells influence wild-type neighbors) plays a role. Regardless, our data and controls in this manuscript show that average redox ratio of stem cells in the basal layer of skin is maintained in a tight range that only changes in the presence of mutant cells.

Regarding FAD/ FAD+ NAD(P)H, we would like to clarify the images and our explanation in the previous response. We showed that FAD/ FAD+ NAD(P)H had *the exact same trend* as NAD(P)H/ FAD: in each mouse, the mutant cells and wild-type neighbors were different from basal / Day 0 controls. Once normalized to Day 0 / basal redox in the *same manner* in both cases, both NAD(P)H/ FAD and FAD/ NAD(P)H/ FAD show statistically significant differences in the 3 groups - just in opposite directions, as is expected. This is because the differences are not an artifact created by ratios: NAD(P)H itself drops in the mutant cells and can be observed in images as well as quantification (right).

C. Also as noted in my previous review, comparisons of the un-normalized redox ratios highlighted in Fig. S3 and Fig. S5, indicate that the fact that WT and Hras neighbors are similar is more likely driven by changes in the WT cells than the Hras cells. This would not be consistent with the conclusion “However, the winner cells in each model ... rapidly recover their redox ratios, irrespective of the mutation induced.” The Hras+ cells are not recovering their redox based on what is shown in Fig. S5, but they are somehow making their neighbors reduce their redox. Of course, and as noted in point A comparisons of data between days is not robust if variations in incident intensity are not calibrated for. The conclusion that in the BcatGOF cases differences remain than in the Hras model they are abrogated at 10 days remains valid. But how these differences are abrogated (i.e. by the “winner” cells in the Hras case recovering their redox ratios) cannot be asserted.

As explained in detail in answer A, we do keep all imaging conditions and analyses the same between Day 0, Day 5 and Day 10 while *revisiting the same mouse*, and thus we can specifically see differences induced by mutation not observed in wild-type controls.

We agree with the reviewer that our data on Hras from Fig. S5 indicate that Hras cells influence wild-type neighbors as well to reduce their redox ratio. This probably has biological significance since in Hras mutant epidermis alone we observe enhanced pyruvate to lactate generation (and likely secretion) as mentioned in the discussion. We thank the reviewer for providing us the opportunity to correct the interpretation of the data and have changed the text as follows (orange indicates changes in the text):

“Thus, by comparing the “winner” Hras^{G12V} and “loser” βcatGOF mutation, we discover that both mutations initially lower the redox ratio, but in the Hras^{G12V} mutant model, the redox differential between mutant and WT cells is flattened over time.”

D. Concerns regarding the relative magnitude of the impact of metformin alone vs the impact of metformin in the Hras and BcatGOF model remain. The additional mass spec data along with data on Fig. S9 highlight that the metabolic impacts of metformin on control cells are significant, leading to significantly higher variations in the redox values. Thus, from a statistical perspective, a lot more measurements would be required to be able to detect

significant differences for a similar size effect (i.e. the differences induced by Hras and BcatGOF vs. WT on day 5).

We agree that Metformin does indeed have an effect on wild-type controls. However, the decrease in fractional contribution of glucose to mitochondrial oxidation is much more in mutant mice compared to the decrease observed in wild-type mice. This makes Metformin a reasonable option compared to other drugs or electron transport chain mutants which would completely shut down mitochondrial oxidation and likely result in cell death or cell health issues.

Statistical analysis for the graphs in Fig. 6A and Fig. 7A were done in Graphpad Prism using 1-way ANOVA by comparing average redox as well as nested 1-way ANOVA. Nested 1-way ANOVA takes into account all technical replicates (number of cells—approximately 100 cells per mouse per condition) to determine the confidence of prediction of the mean value. Both tests do not show significant differences after metformin treatment whereas these tests showed significant differences between mutant and wild-type cells without Metformin.

Graphical representation of multiple comparisons using Dunnett's test for Bcat and Hras mutants treated with metformin after nested 1-way ANOVA. Overlap with 0 shows lack of significant differences in spite of considering the wide spread of the technical replicates.

We have also now added graphs with direct comparison of mutant cells and wild-type neighbors (*newly added Extended Data 7 F,H*) which although still greater in variance when Metformin treated, is comparable in scale to graphs in Fig. 3B and Fig. 4C. Statistical analysis has been done in Graphpad prism wherein the average redox values were compared using students' ttests, ttest with Welch's correction for unequal variance and nested ttests. All these tests, while incorporating standard deviation in the case of nested, show no significant differences between Day 5 mutant and wild-type cells upon Metformin treatment, while they were significantly different for Fig. 3B and 4C under untreated conditions with >95% confidence. Thus, in both cases, while the variation of the data is large, we have used large numbers (number of cells) to determine the means that which have been compared to show no significant differences.

We agree however with the reviewer that changes in the variance of per-cell redox ratio could be biologically significant and we have now amended the text to comment on the larger variation of per-cell redox observed in mice treated with Metformin:

"The variance of per cell redox upon metformin treatment however is higher than in untreated mice, reflecting variations in drug penetration."

E. The actual colorbar values should be indicated, instead of min and max.

Thank you for pointing this out. We have now added numerical values to the color bars and added their explanation to Methods.

F. Not clear why in Fig. 1Ci, the corresponding redox ratio images are not also shown and then plotted quantitatively in Fig. 1Ciii.

We thank the Reviewer for this question. In Fig. 1C (as was explained with Rebuttal Fig. 20 and the following point in the first round of reviews), we have imaged NAD(P)H from transgenic animals expressing Fucci2, a cell cycle reporter. While G1 cells express a red nuclear tag compatible with OMI, G2/S cells express a green nuclear tag that spectrally overlap with FAD which is also emitted in the green wavelength range and is brighter. Thus, it is not possible to get a clean image of FAD intensities in Fucci2 mice. We can however measure NAD(P)H in these mice, which is what we reported.

G. There are issues with the formatting of the supplementary figure legends and the legend of Fig. S1 doesn't seem to correspond to the panels as labeled in the figure.

Thank you for noticing this. We have now corrected the Fig. S1 legend.

H. References to previous tissue optical redox imaging studies are limited.

We have added more references, especially those summarizing the state of the field now. Please note that given we are beyond the reference limit proposed by the journal we may have to move some of these citations to the Methods section in the future.

Reviewer #4:

Remarks to the Author:

Remaining concerns with revision:

It is appreciated that some of the wording of the manuscript has been modified to address concerns raised by the reviewer. However, significant concerns remain.

Robust power calibrations

The differences between the mutant and WT cells that are presented in this manuscript appear to be on the order of 5%. If on Day 5, data from the same mouse are taken with a 1.1% increase at 750 and a 1.1% decrease at 890 nm relative to Day 0, this would result in a 5% increase in the NADH/FAD signal, even if the redox ratio didn't really change. In the Methods the authors state (lines 593-594):

“Laser power at samples was measured periodically and maintained between 25-35 mW.” First, it is not clear at whether this statement refers to all wavelengths. Secondly, the reported range varies by 30%. So, this is certainly not consistent with the added statement (lines 608-609):

“While revisiting a mouse, imaging conditions were kept constant for NAD(P)H and FAD between Day0, Day4-6 and Day 9-13 shown in the manuscript.”

A variation of 30% (10 mW variation with a mean of 30 mW) in incident power is not constant.

Also, the authors mention that they use two lasers for their measurements. Information regarding what laser is used for what wavelengths, and what is the order of image acquisition. Also, the full depth over which images were acquired and the image acquisition times are not provided. Laser emission can vary with changes in temperature and humidity in a room, so even if the laser settings are the same, the actual power delivered to a specimen can vary. For robust measurements, which rely on multi-day comparisons, it is important to have such fluctuations accounted for.

It is certainly encouraging that variations in litter mate controls do not show significant variations over the 10-15 days of imaging. Still, the days for the littermate control measurements appear very different from the data for the BcatGOF mice. However, as indicated above laser fluctuations can occur over the course of even a multi-hour experiment, so this is not necessarily proof that such fluctuations did not contribute to the observed changes, which limits the rigor of the results and conclusions.

To summarise and clarify our previous responses in answer to the point about laser fluctuations:

First, we routinely measure laser power at the sample plane using a power meter and monitor fluctuations as the laser power at the sample needs to be monitored and controlled to be able to quantify changes over time. A graph showing such fluctuations over 3 years was shown in the first response to a similar question (Rebuttal Fig. 18).

Second, as reported in the methods, we routinely monitored power range for both lasers which were maintained between 25-35 mW (acquisition settings modified accordingly) over the span of 4-5 years during which the reported results in this paper were collected. This range was specifically mentioned in the methods as the range that allowed us to keep the tissue healthy and to perform revisits without photodamage. This does not mean that between revisits of a single mouse, there were fluctuations of that order – simply the power range that we observed did not cause visible changes in morphology, behavior or redox of skin stem cells in the animal over revisits.

Third, acquisition settings were kept identical between days of revisits (i.e. Day 0, Day 5 and Day 10) as documented in the methods. In addition, power at the sample was tested between revisits to make sure that it was the same as Day 0. When significant fluctuations were observed (which happened only twice- due to alterations after a microscope service and such variability did not persist during routine revisits), that mouse was neither pursued further or used.

Lastly, laser power can indeed fluctuate hourly with temperature and humidity. This is a limitation that applies to all laser scanning microscopy and confocal microscopy. To make meaningful conclusions, we have followed the standards of the field by recording multiple controls that demonstrate that the changes we observe are beyond those due to fluctuating imaging conditions as follows:

1. We follow littermates through revisits taken through similar days of revisits as the mutant animal and show that there is no drop in redox ratio as observed after induction of mutations. In Fig. 2C and figure accompanying the previous response to this reviewer - 3 mice were followed between Day 0 and Day 5; Day 0 and Day 10; and Day 0 and Day 15 - in revisits similar to that shown for revisits in the mutant animals. It is not clear what the reviewer means by “very different for BcatGOF mice”. In both BcatGOF and Hras^{G12V} mice, revisits were imaged between Day 0 and Day 5 and Day 0 and Day 9-11.
2. In addition to following the same animal over time, we have demonstrated redox changes over time by comparing mutant and neighboring wild-type cells in the same animal wherein all imaging conditions are the same (since they are in the same image).

Lack of nuclear segmentation

In the previous review, this reviewer made a fairly detailed argument regarding the importance of segmenting the cytoplasmic regions for making robust quantitative comparisons based on which important functional conclusions are drawn. In the figure submitted by the authors in the previous response:

The nucleus to cytoplasm ratio for BcatGOF#1 mouse varies from about 0.3 to about 0.6. If we start with a case where the nucleus to cytoplasm ratio is 0.3 and NAD(P)H/FAD is 1.1. Assuming a background redox of 1.0 from the nucleus (consistent with some of the scale bars shown in Figures), that means that the redox of the cytoplasm occupying 60% of the cell is about 1.14. Now, if we consider a cell for which the nucleus to cytoplasm ratio is 0.6, and the background (nucleus) redox is the same (i.e. 1.0), then the redox of the 40% cytoplasmic area should be 1.25 to yield the same cell redox of 1.1. This is a 15% difference, 3 times higher than the range of redox ratio differences reported for most of the comparisons made. This means that even though these cells are actually significantly metabolically different, the analysis would identify them as metabolically very similar. Similar arguments can be made for cells that are metabolically similar, but identified as different because their nuclear to cytoplasmic ratio is different. This is very relevant for all the comparisons made, including the metformin ones.

The authors do have the mCherry signal, which in principle should be easy to segment automatically via standard thresholding approaches. The nuclei of the WT mutant cells may be more challenging and in the worst case scenario may require manual segmentation. It appears that the authors have already the segmented cell outlines, since they include cell-based measurements in their results (lines 621-622). At any rate, if the authors would like to draw important functional conclusions from their elegant experiments, then it is the opinion of this reviewer at least, that these more careful assessments need to be undertaken.

As noted in the previous response, all our analyses are relative to Day 0/ neighboring cells and not representing absolute redox ratio of the cytosol in each cell. These analyses show statistically significant differences after expression of oncogenic mutations alone and not in wild-type littermates. While nuclear segmentation is indeed a future goal, it would require 3D automatic segmentation tools that can process the mitochondria - like signal of NAD(P)H through the layers of skin – which are not developed yet.

Data normalization presentation

It is great to see that the authors have modified their previous interpretation of the data from “However, the winner cells in each model ... rapidly recover their redox ratios, irrespective of the mutation induced.” to indicate that the differences are now flattened. However, the manner in which the data is presented in the main figure of the paper still presents the data in a manner which suggests recovery of the redox ratio for the Hras group. This remains misleading and in the opinion of this reviewer, data in the format presented in supplemental figures 2 and 4 should replace the format of the data shown in figures 3 and 4.

Redox differential between mutant and wild-type cells is a major point of the manuscript and the dynamic change in this redox differential is an important difference between BcatGOF and Hras^{G12V} cells that we observe. These differences are also flattened when redox changes are prevented by the administration of Metformin. Hence, the differences between wild-type neighbors and mutant cells have been presented in the main figure and the changes in relation to Day 0 as supporting

data. To ensure that the reader can clearly follow the goal that the supporting figure provides meaningful context to the main figure, the figure legends for both graphs explicitly refer to each other.

Metformin result interpretation

Given the magnitude of the effects of metformin, the limitations with the proper calibration and nuclear exclusion from redox estimates, the results remain unconvincing. Inclusion of experiments from more mice would make results more solid. Given the impact of metformin on variation and the size of the effect the authors observe in the non-metformin treatment groups, it should be possible to calculate the number of independent samples (i.e. mice) that need to be sampled. If such analysis indicates that 3 mice suffice, and the images are analyzed in a more robust manner, then the lack of significance reported would certainly be more robust.

As explained in the previous response - we showed statistical tests that accounted for the spread of data in the metformin model, and still showed lack of significant differences from 3 mice.

Reviewer 5

In this work from Greco et al., both in vivo optical ratios and isotope tracing are used to discern changes in single cell redox ratios with respect to gain of function mutations in b-catenin and mutant Hras. The work encompasses both in vitro and in vivo experiments, highlighting the possibility of cell to cell communication as the rationale for differential changes in redox state. I would only offer minor suggestions with respect to the remaining concerns of Reviewer #4:

1. The authors show that tumors formed in vivo from BcatGOF and HrasG12V rely on differential metabolism. Clearly glucose uptake and metabolism drives the redox phenotype. I would encourage the authors to comment on glucose uptake between the 2 models and if what is being observed is dependent on the amount of glucose carbons that the cell takes up?

We do not observe significant differences in glucose uptake as observed through 2-NBDG labelling, a fluorescent glucose analog, in vivo, (Rebuttal Figure 6) to explain the differences in redox ratio between neighboring WT and mutant cells. Importantly, through $^{13}\text{C}_6$ -glucose labelled mass spectrometry we find that fractional contribution of glucose to pyruvate is not different between the mutant epidermis and wild-type. The differences in glucose metabolism between mutant and wild-type cells start downstream of pyruvate. Abrogating these differences via metformin prevents downstream redox changes and aberrant phenotypes caused due to expression of βcatGOF and Hras^{G12V} mutations in the epidermal stem cells.

While it is difficult to measure rates of glucose uptake with the same accuracy as fractional contributions of glucose, taken together, the changes in glucose metabolism and their modulation by metformin show that that differential use of glucose for mitochondrial oxidation drives the changes in optical redox ratio and downstream aberrant phenotypes.

2. I find the author's response with respect to the system calibration adequate for this study. They are not measuring absolute values and they provide reasonable metrics for stability given such a challenging experiment.

We thank the reviewer for this summary and support of our methodology and analysis.

3. I would however suggest they tone down their use of the word "winner" or eliminate altogether. I find it confusing and also unnecessary.

We have now gone through the manuscript text and toned down the usage of winner and loser and instead focused on the cell behavior leading to cell competition outcome that we demonstrate (elimination/ expansion/ proliferative advantage).

We have also used the final paragraph in discussion to describe the discrepancy in the terms "winner" and "loser" commonly used in the cell competition field, in our model -since these cell behaviors are highly context dependent and not an inherent or unalterable property of the cells.

4. The authors spend a paragraph in the discussion (ln 523-537) discussing the evidence for cell-cell communication. Given that the authors don't actually prove cell-cell communication through experimentation, meaning that they do not do

an experiment where they show one cell makes a metabolite and the other cell consumes it, I would tone this language down and consider also that the communication may also just be substrate competition.

We have now cut down paragraph on cell-cell communication and also raised alternative hypothesis like substrate competition as the reviewer has suggested.